# WHY IS YOUR LANGUAGE MODEL A POOR IMPLICIT REWARD MODEL?

**Noam Razin[†], Yong Lin[†], Jiarui Yao[‡], Sanjeev Arora[†]**

[†] Princeton Language and Intelligence, Princeton University
[‡] University of Illinois Urbana-Champaign

## ABSTRACT

Reward models are key to language model post-training and inference pipelines. Conveniently, recent work showed that every language model defines an *implicit reward model (IM-RM)*, without requiring any architectural changes. However, such IM-RMs tend to generalize worse, especially out-of-distribution, compared to *explicit reward models (EX-RMs)* that apply a dedicated linear head over the hidden representations of a language model. The existence of a generalization gap is puzzling, as EX-RMs and IM-RMs are nearly identical. They can be trained using the same data, loss function, and language model, and differ only in how the reward is computed. Toward a fundamental understanding of the implicit biases underlying different reward model types, we investigate the root cause of this gap. Our main finding, backed by theory and experiments, is that IM-RMs rely more heavily on superficial token-level cues. Consequently, they often generalize worse than EX-RMs under token-level distribution shifts, as well as in-distribution. Furthermore, we provide evidence against alternative hypotheses for the generalization gap. Most notably, we challenge the claim that IM-RMs struggle in tasks where generation is harder than verification because they can operate both as a verifier and a generator. Overall, our results highlight that seemingly minor design choices can substantially impact the generalization behavior of reward models.

## 1 INTRODUCTION

Language model post-training and inference pipelines often rely on *reward models* to assess the quality of generated responses (Cobbe et al., 2021; Achiam et al., 2023; Dubey et al., 2024; Team et al., 2024; Qwen et al., 2024; Snell et al., 2025). Yet, little is known about the relative advantages and disadvantages of different reward model types. Two prevalent, nearly identical types are *explicit reward models (EX-RMs)* (Ouyang et al., 2022) and *implicit reward models (IM-RMs)* (Rafailov et al., 2023). EX-RMs and IM-RMs can be trained based on the same language model $\pi_\theta$, using the same data and loss function. They differ only in how the reward is computed: EX-RMs apply a linear head over the hidden representation that $\pi_\theta$ produces for a prompt-response pair $(\mathbf{x}, \mathbf{y})$, while the reward of an IM-RM is implicitly defined by $\pi_\theta$ through $\ln \pi_\theta(\mathbf{y}|\mathbf{x})$ — see Figure 1.

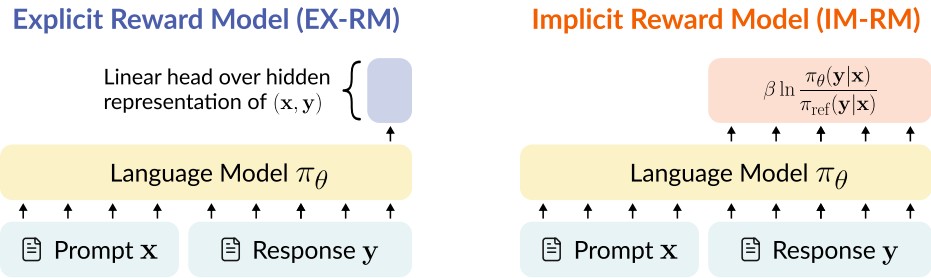

Figure 1: **Explicit vs implicit reward models.** To compute the reward for a prompt-response pair $(\mathbf{x}, \mathbf{y})$, an EX-RM applies a linear head to the hidden representation that the language model $\pi_\theta$ produces for $(\mathbf{x}, \mathbf{y})$. In contrast, the reward of an IM-RM is implicitly defined by $\pi_\theta$ through $\beta \ln \frac{\pi_\theta(\mathbf{y}|\mathbf{x})}{\pi_{\text{ref}}(\mathbf{y}|\mathbf{x})}$, where $\beta \in \mathbb{R}_{>0}$ is a fixed coefficient and $\pi_{\text{ref}}$ is a reference distribution (*cf.* Rafailov et al. (2023)).

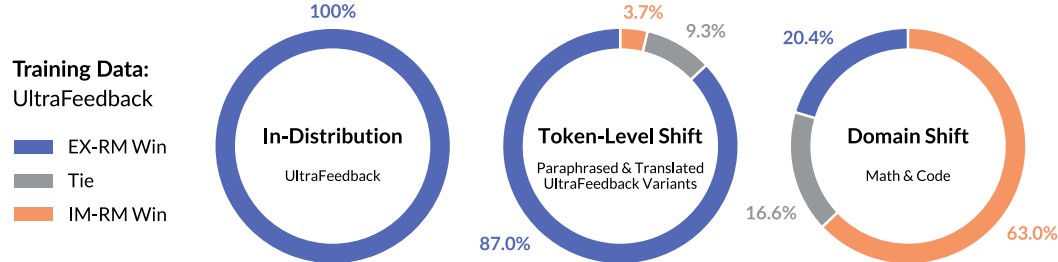

Figure 2: **IM-RMs are less robust than EX-RMs to token-level distribution shifts, but perform comparably or better under domain shifts.** We trained EX-RMs and IM-RMs on UltraFeedback (Cui et al., 2024), using the same initial language models, and evaluated their accuracy in-distribution (UltraFeedback test set), under token-level shifts (three UltraFeedback variants, in which responses were either paraphrased or translated to another language), and under domain shifts (two math and one code datasets). Reported are the win-rates, *i.e.*, the percentage of evaluations in which either the EX-RM or IM-RM achieved a higher accuracy. If the accuracies were within 1% of each other, we considered it a tie. The experiment included three random seeds per configuration and six language models: Gemma-2-2B-IT (Team et al., 2024), Qwen-2.5-1.5B-Instruct, Qwen-2.5-3B-Instruct (Qwen et al., 2024), Llama-3.2-1B-Instruct, Llama-3.2-3B-Instruct, and Llama-3.1-8B-Instruct (Dubey et al., 2024). See Section 5.2 for additional details.

Despite the vast similarity of EX-RMs and IM-RMs, prior work (Lin et al., 2024; Lambert et al., 2025; Swamy et al., 2025) observed empirically that IM-RMs tend to generalize worse, especially out-of-distribution, as measured by *accuracy* in ranking candidate responses. The existence of a *generalization gap* is puzzling. Why would a seemingly minor difference in how the reward is computed substantially affect the resulting accuracy of a reward model?

Toward a fundamental understanding of the implicit biases underlying different reward model types, we investigate the root cause for the generalization gap between EX-RMs and IM-RMs. Our main finding, established through theory and experiments, is that IM-RMs rely more heavily on superficial token-level cues. As a result, IM-RMs typically generalize worse than EX-RMs to token-level distribution shifts (*i.e.*, to responses that are semantically similar to in-distribution responses, but have different surface forms), as well as in-distribution. On the other hand, when subject to domain shifts, IM-RMs can perform comparably to or better than EX-RMs — see Figures 2 and 5.

Before arriving at this conclusion, we first consider an alternative hypothesis for the generalization gap, alluded to in the literature (*cf.* Dong et al. (2024); Singhal et al. (2024)): IM-RMs are harder to learn in tasks with a *generation-verification gap* due to their dual role as a verifier and a generator (Section 3). Specifically, in tasks where responses can be categorized into correct and incorrect, an IM-RM is trained not only to assign a high reward to correct responses, but also to generate them via its underlying language model. If generating correct responses is harder than verifying their correctness, then the (verification) accuracy of IM-RMs should intuitively lag behind that of EX-RMs, which need only verify responses. However, we challenge this intuitive argument by proving that learning to verify with IM-RMs does not require learning to generate. Experiments on a Hamiltonian cycle verification task corroborate our theory.

Then, to identify what drives the generalization gap between EX-RMs and IM-RMs, we theoretically characterize their learning dynamics, *i.e.*, the evolution of rewards during gradient-based training (Section 4.1). Our analysis reveals that the learning dynamics of EX-RMs depends on responses primarily through their hidden representations, whereas IM-RMs are more sensitive to the specific tokens appearing in the responses. In particular, for IM-RMs, increasing the reward of a response may not affect, or even decrease, the reward of a semantically similar response that consists of different tokens. This leads to our conclusion that IM-RMs often underperform EX-RMs since they rely more strongly on superficial token-level cues. We further substantiate this claim: *(i)* theoretically, by providing settings in which IM-RMs provably fail to generalize to unseen tokens, while EX-RMs generalize successfully when hidden representations are well-structured (Appendix B), and *(ii)* empirically across controlled and real-world settings (Section 5).

Overall, our results highlight that seemingly minor design choices can have an outsized effect on how reward models generalize. We hope insights from this work will spur further research into the implicit biases of different reward model types and facilitate enhancing their robustness.

**Related work.** We discuss related work throughout and defer an extended account to Appendix C.

## 2 PRELIMINARIES

Let $\mathcal{V}$ be a finite vocabulary of tokens and $\mathcal{V}^*$ denote the set of all finite-length token sequences. Language models can be decomposed into two parts. First, a neural network backbone that intakes a sequence of tokens $\mathbf{v} \in \mathcal{V}^*$ and produces a *hidden representation* $\mathbf{h_v} \in \mathbb{R}^D$ (*e.g.*, a Transformer (Vaswani et al., 2017)). Second, an *unembedding matrix* $\mathbf{U} \in \mathbb{R}^{|\mathcal{V}| \times D}$ that converts the hidden representation into logits for the next-token distribution. Given a prompt $\mathbf{x} \in \mathcal{V}^*$, a language model $\pi_\theta$ assigns probabilities to responses $\mathbf{y} \in \mathcal{V}^*$ autoregressively:

$$\pi_\theta(\mathbf{y}|\mathbf{x}) = \prod_{k=1}^{|\mathbf{y}|} \pi_\theta(\mathbf{y}_k|\mathbf{x}, \mathbf{y}_{<k}) = \prod_{k=1}^{|\mathbf{y}|} \text{softmax}(\mathbf{U}\mathbf{h}_{\mathbf{x}, \mathbf{y}_{<k}})_{\mathbf{y}_k},$$

where $\theta$ stands for the language model's parameters (*i.e.*, it includes the parameters of the neural network backbone and the unembedding matrix), $\mathbf{y}_{<k}$ and $\mathbf{y}_k$ denote the first $k-1$ tokens and $k$th token of $\mathbf{y}$, respectively, and $\text{softmax}(\mathbf{z})_v := \exp(\mathbf{z}_v)/\sum_{v' \in \mathcal{V}} \exp(\mathbf{z}_{v'})$ for $\mathbf{z} \in \mathbb{R}^{|\mathcal{V}|}$.

### 2.1 REWARD MODELS

Reward models are typically initialized from a preexisting language model $\pi_\theta$ and trained to predict a scalar reward that indicates the quality of a response $\mathbf{y}$ to a prompt $\mathbf{x}$. Two prevalent reward model types are *explicit reward models (EX-RMs)* (Ouyang et al., 2022) and *implicit reward models (IM-RMs)* (Rafailov et al., 2023). As detailed below, EX-RMs and IM-RMs are almost identical. They are trained using the same data, loss function, and language model $\pi_\theta$, and differ only in how the reward is computed based on $\pi_\theta$. This work is devoted to understanding why, despite these vast similarities, EX-RMs and IM-RMs generalize differently.

**Explicit reward model (EX-RM).** To compute the reward for a prompt-response pair $(\mathbf{x}, \mathbf{y})$, an EX-RM applies a linear head $\mathbf{u} \in \mathbb{R}^D$ over the hidden representation $\mathbf{h}_{\mathbf{x}, \mathbf{y}}$ that $\pi_\theta$ produces:

$$r_{\theta_{\text{EX}}}(\mathbf{x}, \mathbf{y}) := \langle \mathbf{u}, \mathbf{h}_{\mathbf{x}, \mathbf{y}} \rangle , \tag{1}$$

where $\theta_{\text{EX}}$ stands for the trainable parameters of the EX-RM (*i.e.*, it includes the parameters of the neural network backbone and the linear head).

**Implicit reward model (IM-RM).** As shown in Rafailov et al. (2023), every language model $\pi_\theta$ defines an IM-RM through the log probabilities that it assigns to responses:

$$r_{\theta_{\text{IM}}}(\mathbf{x}, \mathbf{y}) := \beta \ln \frac{\pi_{\theta_{\text{IM}}}(\mathbf{y}|\mathbf{x})}{\pi_{\text{ref}}(\mathbf{y}|\mathbf{x})} , \tag{2}$$

where $\theta_{\text{IM}} = \theta$ denotes the trainable parameters of the IM-RM, $\beta \in \mathbb{R}_{>0}$ is a fixed coefficient, and the reference distribution $\pi_{\text{ref}}$ is canonically the language model from which the IM-RM was initialized. Note that besides assigning rewards, an IM-RM can generate responses via $\pi_{\theta_{\text{IM}}}$. Moreover, increasing the reward of a response entails increasing its probability under $\pi_{\theta_{\text{IM}}}$.

**Training objective.** Let $\mathcal{D}_\mathcal{T}$ be a training set containing preferences $(\mathbf{x}, \mathbf{y}^+, \mathbf{y}^-)$, where $\mathbf{x}$ is a prompt, $\mathbf{y}^+$ is a chosen response to $\mathbf{x}$, and $\mathbf{y}^-$ is a rejected response to $\mathbf{x}$. EX-RMs and IM-RMs are usually trained by minimizing a Bradley-Terry log-likelihood loss (Bradley and Terry, 1952):

$$\mathcal{L}(r) := \frac{1}{|\mathcal{D}_\mathcal{T}|} \sum_{(\mathbf{x}, \mathbf{y}^+, \mathbf{y}^-) \in \mathcal{D}_\mathcal{T}} -\ln \sigma\big(r(\mathbf{x}, \mathbf{y}^+) - r(\mathbf{x}, \mathbf{y}^-)\big) , \tag{3}$$

where $r : \mathcal{V}^* \times \mathcal{V}^* \to \mathbb{R}$ can be either $r_{\theta_{\text{EX}}}$ or $r_{\theta_{\text{IM}}}$ and $\sigma : \mathbb{R} \to [0, 1]$ denotes the sigmoid function.

**Outcome vs process rewards.** In certain domains, such as math and reasoning, reward models have been used for providing feedback on intermediate steps of a response (Uesato et al., 2022; Lightman et al., 2024). Both EX-RMs and IM-RMs can be adapted to evaluate partial responses, the former by applying the linear head over the hidden representation of each intermediate step and the latter by using the conditional log probabilities of these steps. For conciseness, we focus on settings where the reward is assigned to complete responses.

**Generative reward model.** A nascent reward model variant, which we refer to as *explicit generative reward model (EX-GRM)* (Zhang et al., 2025), directly asks the language model $\pi_\theta$ to verify whether $\mathbf{y}$ is a good response to $\mathbf{x}$. Then, the probability assigned to the token $\text{Yes}$ is taken as the reward. For brevity, we focus on EX-RMs and IM-RMs in the main text and defer an extension of our theoretical and empirical analyses for EX-GRMs to Appendices D and F, respectively. Notably, we find that the main conclusions stated for EX-RMs hold for EX-GRMs as well.

## 2.2 Measuring Generalization via Accuracy

In accordance with Lin et al. (2024); Lambert et al. (2025); Zhou et al. (2025); Frick et al. (2025); Liu et al. (2025), we measure generalization via the *accuracy* of a reward model in ranking responses over preference data unseen in training.

**Definition 1.** For a finite set $\mathcal{S}$ containing preferences $(\mathbf{x}, \mathbf{y}^+, \mathbf{y}^-)$, where $\mathbf{x}$ is a prompt, $\mathbf{y}^+$ is a chosen response, and $\mathbf{y}^-$ is a rejected response, the *accuracy* of $r : \mathcal{V}^* \times \mathcal{V}^* \to \mathbb{R}$ over $\mathcal{S}$ is:

$$\mathrm{acc}_{\mathcal{S}}(r) := \frac{1}{|\mathcal{S}|} \sum_{(\mathbf{x}, \mathbf{y}^+, \mathbf{y}^-) \in \mathcal{S}} \mathbb{1}\left[ r(\mathbf{x}, \mathbf{y}^+) > r(\mathbf{x}, \mathbf{y}^-) \right] + \frac{1}{2} \cdot \mathbb{1}\left[ r(\mathbf{x}, \mathbf{y}^+) = r(\mathbf{x}, \mathbf{y}^-) \right],$$

where $\mathbb{1}[\cdot]$ is an indicator function. Note that the maximal accuracy is one and the minimal is zero.

# 3 Are IM-RMs Harder to Learn in Tasks With a Generation-Verification Gap?

A potential explanation for why IM-RMs often underperform EX-RMs, alluded to in the literature (*cf.* Dong et al. (2024); Singhal et al. (2024)), is that IM-RMs are harder to learn in tasks with a *generation-verification gap*. Namely, in tasks where responses can be categorized into correct and incorrect, an IM-RM is trained not only to assign a high reward to correct responses, but also to generate them via its underlying language model. Thus, if generating correct responses is harder than verifying their correctness in a given task, then the accuracy of IM-RMs should intuitively fall below that of EX-RMs, which need only verify responses.

We prove that this intuitive explanation is flawed — learning to verify with IM-RMs does not require learning to generate (Section 3.1). Experiments on a Hamiltonian cycle verification task, which is widely believed to exhibit a generation-verification gap (Arora and Barak, 2009), demonstrate that IM-RMs learn to accurately verify such cycles without being able to generate them (Section 3.2).

## 3.1 Theory: Learning to Verify Does Not Require Learning to Generate

Consider a task defined by a set of valid prompts $\mathcal{X} \subseteq \mathcal{V}^*$ and a function $\mathcal{C}$ that maps every prompt $\mathbf{x} \in \mathcal{X}$ to a set of correct responses $\mathcal{C}(\mathbf{x}) \subseteq \mathcal{V}^*$. Concretely, $\mathcal{X}$ can consist of math problems, with $\mathcal{C}(\mathbf{x})$ containing the correct solutions of a problem $\mathbf{x}$. A prompt $\mathbf{x}$ can also describe the input to some algorithmic task. For example, if the task is to find Hamiltonian cycles in a graph, each $\mathbf{x}$ describes a graph and each response in $\mathcal{C}(\mathbf{x})$ encodes a Hamiltonian cycle (Section 3.2 presents experiments over this task).

In this context, it is natural to say that a reward model is a *verifier* for the task $(\mathcal{X}, \mathcal{C})$ if it assigns non-negligibly higher rewards to correct responses relative to incorrect ones.

**Definition 2.** A reward model $r : \mathcal{V}^* \times \mathcal{V}^* \to \mathbb{R}$ is a *verifier with margin* $\delta \in \mathbb{R}_{>0}$ for the task $(\mathcal{X}, \mathcal{C})$ if for all $\mathbf{x} \in \mathcal{X}$, $\mathbf{y}^+ \in \mathcal{C}(\mathbf{x})$, and $\mathbf{y}^- \in \mathcal{V}^* \setminus \mathcal{C}(\mathbf{x})$:

$$r(\mathbf{x}, \mathbf{y}^+) \geq r(\mathbf{x}, \mathbf{y}^-) + \delta.$$

Note that if $r$ is a verifier for $(\mathcal{X}, \mathcal{C})$, then it achieves perfect accuracy (Definition 1) over all evaluation sets that contain preferences $(\mathbf{x}, \mathbf{y}^+, \mathbf{y}^-)$, where $\mathbf{x} \in \mathcal{X}$, $\mathbf{y}^+ \in \mathcal{C}(\mathbf{x})$, and $\mathbf{y}^- \in \mathcal{V}^* \setminus \mathcal{C}(\mathbf{x})$.

Theorem 1 below establishes that, for an IM-RM to be a verifier, the probability that its underlying language model assigns to correct responses needs to grow by at most a constant multiplicative factor relative to the initial reference distribution $\pi_{\mathrm{ref}}$. In particular, if $\pi_{\mathrm{ref}}$ assigns low probability to correct responses, then an IM-RM can accurately verify correct responses even if it is unable to generate them. Thus, the hypothesis that IM-RMs struggle because they need to learn to generate, as opposed to just verify, does not explain the generalization gap between EX-RMs and IM-RMs.

We further formalize this argument through the notion of an *efficient generator* (Definition 3). A distribution $\pi$ is an efficient generator if the probability that it assigns to correct responses decays at most polynomially with the prompt length $|\mathbf{x}|$. The rationale behind this definition is that obtaining a correct response from an efficient generator requires, with high probability, only a number of samples polynomial in $|\mathbf{x}|$, which often corresponds to task complexity (*e.g.*, the size of a graph in the task of finding Hamiltonian cycles). Corollary 1 shows that, if $\pi_{\mathrm{ref}}$ is not an efficient generator, then an IM-RM does not need to be an efficient generator in order to be a verifier.

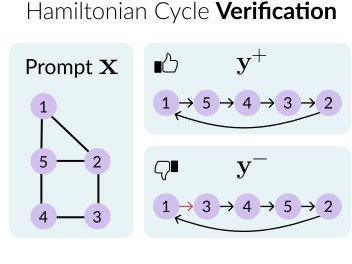
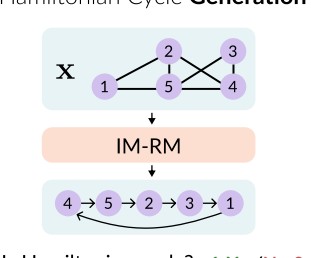

| | EX-RM | IM-RM |
|---|---|---|
| Train Accuracy | 1 | 1 |
| Test Accuracy | 0.980 | 0.993 |
| Correct Generations | - | 0 |

Figure 3: **Learning to verify with IM-RMs does not require learning to generate.** We trained EX-RMs and IM-RMs to solve a Hamiltonian cycle verification task, based on the Pythia-1B language model. Each prompt in the dataset describes an undirected graph and the chosen and rejected responses are permutations of vertices. The chosen responses form Hamiltonian cycles in their respective graphs, while the rejected responses do not (see Appendix G.1 for further details). In accordance with our theory (Section 3.1), although IM-RMs are unable to generate even a single correct Hamiltonian cycle for graphs in the training or test sets, they accurately distinguish between chosen and rejected responses, slightly outperforming EX-RMs. Values in the table are means across three random seeds (standard deviation was under 0.008 in all cases).

**Theorem 1.** *Let $r_{\mathrm{IM}}$ be the IM-RM induced by a distribution $\pi$ over token sequences, i.e., $r_{\mathrm{IM}}(\mathbf{x}, \mathbf{y}) = \beta(\ln \pi(\mathbf{y}|\mathbf{x}) - \ln \pi_{\mathrm{ref}}(\mathbf{y}|\mathbf{x}))$ for $\mathbf{x}, \mathbf{y} \in \mathcal{V}^*$, $\beta \in \mathbb{R}_{>0}$, and a reference distribution $\pi_{\mathrm{ref}}$. Then, $r_{\mathrm{IM}}$ can be a verifier with margin $\delta \in \mathbb{R}_{>0}$ for the task $(\mathcal{X}, \mathcal{C})$ (Definition 2) even if for all prompts $\mathbf{x} \in \mathcal{X}$:*

$$\pi(\mathcal{C}(\mathbf{x})|\mathbf{x}) \le \pi_{\mathrm{ref}}(\mathcal{C}(\mathbf{x})|\mathbf{x}) \cdot \exp(\delta/\beta).$$

*That is, for all prompts, the probability of $\pi$ generating a correct response is greater than that of $\pi_{\mathrm{ref}}$ by at most a constant multiplicative factor.*

*Proof sketch (full proof in Appendix E.1).* The proof is by construction. We define a distribution $\pi$ such that the IM-RM it induces is a verifier for the task $(\mathcal{X}, \mathcal{C})$ with an exact margin of $\delta$. Then, we directly upper bound the probability that $\pi$ assigns to correct responses. □

**Definition 3.** We say that a distribution $\pi$ over token sequences is an *efficient generator* for the task $(\mathcal{X}, \mathcal{C})$ if there exist $k \in \mathbb{N}$ and $\alpha \in \mathbb{R}_{>0}$ such that $\pi(\mathcal{C}(\mathbf{x})|\mathbf{x}) \ge \alpha^{-1}|x|^{-k}$ for all $\mathbf{x} \in \mathcal{X}$.

**Corollary 1.** *Under the notation of Theorem 1, suppose that $\pi_{\mathrm{ref}}$ is not an efficient generator for the task $(\mathcal{X}, \mathcal{C})$ (Definition 3). Then, for any $\delta \in \mathbb{R}_{>0}$, the IM-RM $r_{\mathrm{IM}}$ can be a verifier with margin $\delta$ for $(\mathcal{X}, \mathcal{C})$ (Definition 2) even if the underlying distribution $\pi$ is not an efficient generator for $(\mathcal{X}, \mathcal{C})$.*

Proving Corollary 1 based on Theorem 1 is straightforward — see Appendix E.2.

## 3.2 Experiments: Hamiltonian Cycle Verification

We corroborate the analysis of Section 3.1 by empirically demonstrating that, in tasks where generation is harder than verification, IM-RMs can learn to verify comparably or better than EX-RMs, without learning to generate. To avoid confounding factors, we focus on a synthetic Hamiltonian cycle verification task which, unless $\mathsf{P} = \mathsf{NP}$, exhibits a generation-verification gap.

**Setting.** We created a preference dataset in which every prompt describes an undirected graph that contains at least one Hamiltonian cycle and the chosen and rejected responses are permutations of vertices. Chosen responses form Hamiltonian cycles in their respective graphs, whereas the rejected responses do not. We then trained EX-RMs and IM-RMs based on the Pythia-1B language model (Biderman et al., 2023) and evaluated their accuracy. We also evaluated the ability of IM-RMs to generate Hamiltonian cycles. See Appendix G.1 for additional details.

**Results.** Figure 3 illustrates the experimental setup and reports the results. Confirming our theory (Section 3.1), IM-RMs are able to accurately verify responses, *i.e.*, achieve perfect accuracy on the training set and near-perfect on the test set, while being unable to generate even a single correct Hamiltonian cycle for graphs in the training or test sets. This showcases that the lower accuracy that IM-RMs often achieve compared to EX-RMs (Lin et al., 2024; Lambert et al., 2025; Swamy et al., 2025) does not stem from IM-RMs needing to learn to generate in order to verify.

## 4 THEORY: IM-RMs RELY ON TOKEN-LEVEL CUES WHILE EX-RMS GENERALIZE VIA HIDDEN REPRESENTATIONS

To identify what causes the generalization gap between EX-RMs and IM-RMs, we analyze their learning dynamics. Specifically, we characterize how the reward assigned to a prompt-response pair evolves during gradient-based training (Section 4.1). The characterization suggests that IM-RMs often generalize worse than EX-RMs since they rely more heavily on superficial token-level cues. We further support this claim: *(i)* theoretically, by providing a (simplified) setting in which IM-RMs provably fail to generalize to unseen tokens, whereas EX-RMs can generalize when hidden representations are well-structured (Appendix B), and *(ii)* empirically, by showing that IM-RMs are less robust to token-level shifts, but perform comparably or better under domain shifts (Section 5).

### 4.1 LEARNING DYNAMICS

We examine how performing a gradient update on the training example $(\mathbf{x}, \mathbf{y}^+, \mathbf{y}^-) \in \mathcal{D}_{\mathcal{T}}$ influences the reward assigned to an unseen prompt-response pair $(\bar{\mathbf{x}}, \bar{\mathbf{y}})$, *i.e.*:

$$\Delta r_\theta(\bar{\mathbf{x}}, \bar{\mathbf{y}}) := r_{\theta - \eta \nabla \ell_\theta(\mathbf{x}, \mathbf{y}^+, \mathbf{y}^-)}(\bar{\mathbf{x}}, \bar{\mathbf{y}}) - r_\theta(\bar{\mathbf{x}}, \bar{\mathbf{y}}),$$

where $\eta \in \mathbb{R}_{>0}$ is a learning rate, $\ell_\theta(\mathbf{x}, \mathbf{y}^+, \mathbf{y}^-) := -\ln \sigma(r_\theta(\mathbf{x}, \mathbf{y}^+) - r_\theta(\mathbf{x}, \mathbf{y}^-))$ denotes the loss over $(\mathbf{x}, \mathbf{y}^+, \mathbf{y}^-)$, and $\theta$ stands for either $\theta_{\text{EX}}$ or $\theta_{\text{IM}}$. We note that analogous approaches have been valuable for studying the effects of language model post-training (Im and Li, 2025; Razin et al., 2025a; Ren and Sutherland, 2025). By a Taylor approximation of $r_\theta(\bar{\mathbf{x}}, \bar{\mathbf{y}})$ around $\theta$, the change in reward can be expressed as:

$$\Delta r_\theta(\bar{\mathbf{x}}, \bar{\mathbf{y}}) = -\eta \left\langle \nabla r_\theta(\bar{\mathbf{x}}, \bar{\mathbf{y}}), \nabla \ell_\theta(\mathbf{x}, \mathbf{y}^+, \mathbf{y}^-) \right\rangle + \mathcal{O}(\eta^2).$$

Thus, up to second order terms in the learning rate $\eta$, which is commonly small for reward model training (Liu et al., 2024; Malik et al., 2025), the change in reward is determined by the inner product of the reward and loss gradients. Below, we characterize this inner product for EX-RMs and IM-RMs. Motivated by the fact that reward models have achieved competitive performance when fixing the backbone that produces hidden representations (Wang et al., 2024a), we assume that hidden representations are not updated during training.[1] Yet, as Section 5 verifies empirically, the implications of our analysis apply also when all reward model parameters are learned (we do not fix the hidden representations in our experiments).

**Assumption 1.** Hidden representations are fixed during training: only the linear head $\mathbf{u}$ for EX-RMs and unembedding matrix $\mathbf{U}$ for IM-RMs are updated (*i.e.*, $\theta_{\text{EX}} = \mathbf{u}$ and $\theta_{\text{IM}} = \mathbf{U}$).

**EX-RM dynamics.** For EX-RMs, the change in reward is given by (derivation in Appendix E.3):

$$\Delta r_{\theta_{\text{EX}}}(\bar{\mathbf{x}}, \bar{\mathbf{y}}) = \left\langle \mathbf{h}_{\bar{\mathbf{x}}, \bar{\mathbf{y}}}, \mathbf{h}_{\mathbf{x}, \mathbf{y}^+} - \mathbf{h}_{\mathbf{x}, \mathbf{y}^-} \right\rangle \cdot \eta g(\theta_{\text{EX}}), \tag{4}$$

where $g(\theta_{\text{EX}}) := \sigma(r_{\theta_{\text{EX}}}(\mathbf{x}, \mathbf{y}^-) - r_{\theta_{\text{EX}}}(\mathbf{x}, \mathbf{y}^+)) > 0$. As Equation (4) shows, $r_{\theta_{\text{EX}}}(\bar{\mathbf{x}}, \bar{\mathbf{y}})$ increases when $\mathbf{h}_{\bar{\mathbf{x}}, \bar{\mathbf{y}}}$ is more closely aligned with $\mathbf{h}_{\mathbf{x}, \mathbf{y}^+}$ than with $\mathbf{h}_{\mathbf{x}, \mathbf{y}^-}$. In particular, the change in reward depends on responses only through their hidden representations. Consequently, the extent to which an EX-RM generalizes to unseen prompt-response pairs is largely determined by the structure of the hidden representations, which are produced by a pretrained (and sometimes also post-trained) language model. Since these representations are known to encode semantics (Zou et al., 2023; Park et al., 2024), this suggests that EX-RMs can generalize to unseen responses even if they consist of entirely different tokens from responses in the training set.

---

[1]An analogous learning dynamics analysis can be conducted without assuming fixed hidden representations (Assumption 1). In that case, the resulting dynamics remain the same, up to additive terms introduced by the update to hidden representations. These terms are less interpretable because they depend on the specific neural network architecture used for producing hidden representations. Nonetheless, the close match between our theoretical predictions and the experiments in Section 5, where hidden representations are not fixed, indicates that the effect of these additional terms does not counteract the mechanisms we identify under Assumption 1.

**IM-RM dynamics.** For IM-RMs, the change in reward is more complex and is given by (derivation in Appendix E.4; adapted from Theorem 7 of Razin et al. (2025a)):

$$\Delta r_{\theta_{\mathrm{IM}}}(\bar{\mathbf{x}}, \bar{\mathbf{y}}) = \left( \sum_{k=1}^{|\bar{\mathbf{y}}|} \sum_{l=1}^{|\mathbf{y}^+|} \rho_{k,l}(\mathbf{y}^+) \cdot \left\langle \mathbf{h}_{\bar{\mathbf{x}}, \bar{\mathbf{y}}_{<k}}, \mathbf{h}_{\mathbf{x}, \mathbf{y}^+_{<l}} \right\rangle - \sum_{k=1}^{|\bar{\mathbf{y}}|} \sum_{l=1}^{|\mathbf{y}^-|} \rho_{k,l}(\mathbf{y}^-) \cdot \left\langle \mathbf{h}_{\bar{\mathbf{x}}, \bar{\mathbf{y}}_{<k}}, \mathbf{h}_{\mathbf{x}, \mathbf{y}^-_{<l}} \right\rangle \right)$$
$$\cdot \eta g(\theta_{\mathrm{IM}}) \beta^2 + \mathcal{O}(\eta^2) \,, \tag{5}$$

where $g(\theta_{\mathrm{IM}}) := \sigma(r_{\theta_{\mathrm{IM}}}(\mathbf{x}, \mathbf{y}^-) - r_{\theta_{\mathrm{IM}}}(\mathbf{x}, \mathbf{y}^+)) > 0$ and the coefficient $\rho_{k,l}(\mathbf{v}) \in [-2, 2]$, for $\mathbf{v} \in \{\mathbf{y}^+, \mathbf{y}^-\}$, is determined by the tokens $\bar{\mathbf{y}}_k, \mathbf{v}_l$, and corresponding next-token distributions:

$$\rho_{k,l}(\mathbf{v}) := \mathbb{1}[\bar{\mathbf{y}}_k = \mathbf{v}_l] - \pi_{\theta_{\mathrm{IM}}}(\bar{\mathbf{y}}_k | \mathbf{x}, \mathbf{v}_{<l}) - \pi_{\theta_{\mathrm{IM}}}(\mathbf{v}_l | \bar{\mathbf{x}}, \bar{\mathbf{y}}_{<k}) + \langle \pi_{\theta_{\mathrm{IM}}}(\cdot | \bar{\mathbf{x}}, \bar{\mathbf{y}}_{<k}), \pi_{\theta_{\mathrm{IM}}}(\cdot | \mathbf{x}, \mathbf{v}_{<l}) \rangle \,.$$

In contrast to EX-RMs, the change in reward for IM-RMs depends on the specific tokens that appear in $\bar{\mathbf{y}}$, $\mathbf{y}^+$, and $\mathbf{y}^-$, as opposed to just their hidden representations. This dependence is introduced by the coefficients $\rho_{k,l}(\mathbf{y}^+)$ and $\rho_{k,l}(\mathbf{y}^-)$, which can be positive or negative. Focusing on $\rho_{k,l}(\mathbf{y}^+)$ (the analysis for $\rho_{k,l}(\mathbf{y}^-)$ is analogous), we distinguish between two cases. If $\bar{\mathbf{y}}_k = \mathbf{y}^+_l$, then $\rho_{k,l}(\mathbf{y}^+)$ is positive since it can be written as $\rho_{k,l}(\mathbf{y}^+) = (1 - \pi_{\theta_{\mathrm{IM}}}(\bar{\mathbf{y}}_k | \mathbf{x}, \mathbf{y}^+_{<l}))(1 - \pi_{\theta_{\mathrm{IM}}}(\mathbf{y}^+_l | \bar{\mathbf{x}}, \bar{\mathbf{y}}_{<k})) + \sum_{v \in \mathcal{V} \setminus \{\bar{\mathbf{y}}_k\}} \pi_{\theta_{\mathrm{IM}}}(v | \bar{\mathbf{x}}, \bar{\mathbf{y}}_{<k}) \pi_{\theta_{\mathrm{IM}}}(v | \mathbf{x}, \mathbf{y}^+_{<l})$. In this case, the term $\rho_{k,l}(\mathbf{y}^+) \langle \mathbf{h}_{\bar{\mathbf{x}}, \bar{\mathbf{y}}_{<k}}, \mathbf{h}_{\mathbf{x}, \mathbf{y}^+_{<l}} \rangle$ has an effect analogous to $\langle \mathbf{h}_{\bar{\mathbf{x}}, \bar{\mathbf{y}}}, \mathbf{h}_{\mathbf{x}, \mathbf{y}^+} \rangle$ from the dynamics of EX-RMs (Equation (4)): it increases the reward of $(\bar{\mathbf{x}}, \bar{\mathbf{y}})$ if the hidden representation of $(\bar{\mathbf{x}}, \bar{\mathbf{y}})$ is aligned with that of $(\mathbf{x}, \mathbf{y}^+)$. However, if $\bar{\mathbf{y}}_k \neq \mathbf{y}^+_l$, then the coefficient $\rho_{k,l}(\mathbf{y}^+)$ can be negative. In this case the effect is opposite: the corresponding term decreases the reward of $(\bar{\mathbf{x}}, \bar{\mathbf{y}})$ if its hidden representation is aligned with that of $(\mathbf{x}, \mathbf{y}^+)$. Notably, when $\bar{\mathbf{y}}_k \neq \mathbf{y}^+_l$ the coefficient $\rho_{k,l}(\mathbf{y}^+)$ consists of three terms: $\langle \pi_{\theta_{\mathrm{IM}}}(\cdot | \bar{\mathbf{x}}, \bar{\mathbf{y}}_{<k}), \pi_{\theta_{\mathrm{IM}}}(\cdot | \mathbf{x}, \mathbf{y}^+_{<l}) \rangle$, $-\pi_{\theta_{\mathrm{IM}}}(\bar{\mathbf{y}}_k | \mathbf{x}, \mathbf{y}^+_{<l})$, and $-\pi_{\theta_{\mathrm{IM}}}(\mathbf{y}^+_l | \bar{\mathbf{x}}, \bar{\mathbf{y}}_{<k})$. The first term is positive and measures the agreement between the next-token distributions corresponding to the contexts of $\bar{\mathbf{y}}_k$ and $\mathbf{y}^+_l$. The latter two terms contribute negatively, and their magnitude is large when $\bar{\mathbf{y}}_k$ is probable under the context of $\mathbf{y}^+_l$ and vice versa. As a result, $\rho_{k,l}(\mathbf{y}^+)$ is likely to be negative when $\bar{\mathbf{y}}_k$ and $\mathbf{y}^+_l$ are tokens that appear in similar contexts.

Since hidden representations often encode semantics, the above implies that the learning dynamics of an IM-RM may inadvertently decrease the reward of responses that are semantically similar to chosen responses in the training set, and increase the reward of those similar to rejected responses, if their tokens have little overlap. This suggests that the generalization gap between EX-RMs and IM-RMs may stem from the latter being less robust to superficial token-level shifts. We support this prospect theoretically in Appendix B, by providing a concrete (simplified) setting in which IM-RMs provably generalize worse than EX-RMs, and empirically in Section 5.

**Relation to prior work.** The learning dynamics of IM-RMs was previously analyzed in Razin et al. (2025a); Im and Li (2025), but for other purposes. Specifically, our work focuses on generalization to unseen responses, whereas Razin et al. (2025a) studied an optimization issue and Im and Li (2025) considered generalization across prompts when responses seen in training and evaluation are the same. See Appendix C for further details.

## 5 EMPIRICAL DEMONSTRATION

Our theory (Section 4) indicates that IM-RMs are more prone than EX-RMs to overfitting superficial token-level cues. In this section, we verify that this conclusion bears out in practice. Namely, in both controlled (Section 5.1) and real-world (Section 5.2) settings, we show that IM-RMs generalize worse than EX-RMs under token-level distribution shifts (*e.g.*, paraphrasing), and often indistribution, yet perform comparably or better under domain shifts. The experiments are based on language models of up to 8B scale from different families: Pythia (Biderman et al., 2023), Gemma-2 (Team et al., 2024), Qwen-2.5 (Qwen et al., 2024), and Llama-3 (Dubey et al., 2024). For brevity, we defer to Appendices F and G some experiments and implementation details, respectively.

### 5.1 CONTROLLED EXPERIMENTS: TOKEN-LEVEL SHIFT

**Setting.** For our controlled experiments, we considered prompts from the Persona dataset (Perez et al., 2022), which ask a language model whether it agrees or disagrees with a given statement.

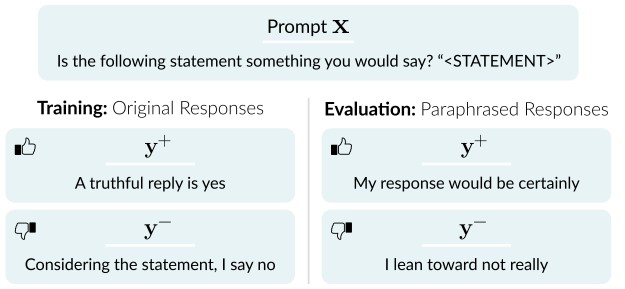

Figure 4: **IM-RMs fail to generalize to a simple token-level distribution shift, while EX-RMs generalize perfectly.** We trained EX-RMs and IM-RMs on prompts from the Persona dataset (Perez et al., 2022). Chosen responses expressed agreement with the prompts, whereas rejected responses expressed disagreement. During evaluation, we included paraphrased versions of the original responses (figure includes exemplar responses). In line with our analysis (Section 4), IM-RMs are extremely inaccurate over paraphrased responses, whereas EX-RMs achieve perfect accuracy. The experiments were based on four language models: Pythia-1B, Qwen-2.5-1.5B-Instruct, Llama-3.2-1B, and Llama-3.2-1B-Instruct. Values in the table are means across the models and three random seeds (standard deviation was below 0.04 in all cases).

We manually wrote four chosen responses that express agreement and four rejected responses that express disagreement. We then trained EX-RMs and IM-RMs, using the same initial language models (Pythia-1B, Qwen-2.5-1.5B-Instruct, Llama-3.2-1B, and Llama-3.2-1B-Instruct), and evaluated their accuracy over the original responses and paraphrased versions of them, *i.e.*, responses that are similar in meaning but consist of different tokens. See Appendix G.2 for further details.

**Results: IM-RMs fail to generalize to paraphrased responses.** Figure 4 illustrates the experimental setup and reports the results. As our theory suggests (Section 4), despite achieving perfect accuracy over the original responses, IM-RMs achieve near-zero accuracy over the paraphrased responses. This reveals that, for IM-RMs, maximizing reward difference between chosen and rejected responses can inadvertently have an opposite effect on paraphrased responses. In contrast, EX-RMs generalize perfectly to the paraphrased responses.

## 5.2 REAL-WORLD EXPERIMENTS: TOKEN-LEVEL AND DOMAIN SHIFTS

### 5.2.1 SETTING

We compared the generalization of EX-RMs and IM-RMs in real-world settings by evaluating their accuracy in-distribution, under token-level shifts, and under domain shifts. We ran experiments in two settings — *general chat* and *math* — using six language models ranging in scale from 1B to 8B: Gemma-2-2B-IT, Qwen-2.5-1.5B-Instruct, Qwen-2.5-3B-Instruct, Llama-3.2-1B-Instruct, Llama-3.2-3B-Instruct, and Llama-3.1-8B-Instruct. As specified below, the two settings differ in which dataset was used for training and the categorization of evaluation datasets into in-distribution, token-level shift, and domain shift. See Appendix G.3 for further details.

**General chat.** We trained EX-RMs and IM-RMs over UltraFeedback (Cui et al., 2024), based on each language model specified above. In-distribution evaluation was performed over the UltraFeedback test set. For evaluating robustness to token-level distribution shifts, we created three variants of the UltraFeedback test set by either paraphrasing, translating to French, or translating to Spanish all responses (via GPT-4.1). For domain shifts, we used the math and code subsets of Reward-Bench (Lambert et al., 2025) and the RewardMATH dataset (Kim et al., 2024).

**Math.** We used RewardMATH for training and evaluated in-distribution performance on a held-out test set. In this setting, the math subset of RewardBench poses a token-level shift while the UltraFeedback variants and code subset of RewardBench pose a domain shift.

### 5.2.2 RESULTS

For the general chat and math settings, respectively, Figures 2 and 5 present the percentage of evaluations in which either the EX-RM or the IM-RM achieved a higher accuracy, where we only compare pairs of reward models that were trained from the same initial language model. Furthermore, Table 1

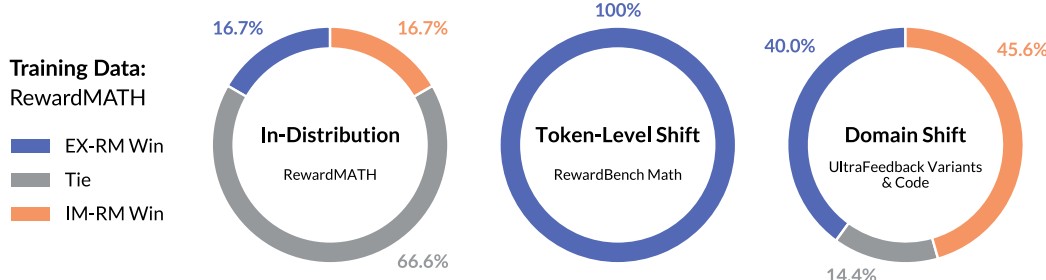

Figure 5: **IM-RMs are less robust than EX-RMs to token-level distribution shifts, but perform comparably or better under domain shifts.** This figure presents the results of an experiment identical to that of Figure 2, except that the reward models were trained on the RewardMATH dataset instead of UltraFeedback. Accordingly, the math subset of RewardBench poses a token-level shift while UltraFeedback variants and the code subset of RewardBench pose a domain shift. Note that, in this setting, EX-RMs and IM-RMs perform similarly in-distribution since both reach near-maximal accuracy (see Table 1).

Table 1: This table supplements Figures 2 and 5 by reporting the accuracy and absolute (normalized) reward margin over the different evaluation categories. In each row, bold font marks the highest accuracy and absolute reward margin (unless the values are within 0.01 of each other, after taking into account standard deviations). For each reward model and evaluation dataset separately, the absolute reward margin is normalized by the standard deviation of rewards to account for arbitrary differences in scale. Notice that EX-RMs consistently induce a higher reward margin, which was shown in Razin et al. (2025b) to be beneficial for optimization via reinforcement learning. Values in the table are means across the models (six in total) and evaluation datasets, with standard deviation computed based on three random seeds. See Tables 3 and 5 in Appendix F for a per evaluation dataset breakdown of the results.

| Training Data | Evaluation | Accuracy | | Absolute Reward Margin | |
|---|---|---|---|---|---|
| | | EX-RM | IM-RM | EX-RM | IM-RM |
| UltraFeedback | In-Distribution | $\mathbf{0.752 \pm 0.009}$ | $0.646 \pm 0.006$ | $\mathbf{1.014 \pm 0.023}$ | $0.813 \pm 0.003$ |
| | Token-Level Shift | $\mathbf{0.665 \pm 0.005}$ | $0.602 \pm 0.003$ | $\mathbf{0.976 \pm 0.008}$ | $0.763 \pm 0.003$ |
| | Domain Shift | $0.621 \pm 0.012$ | $\mathbf{0.720 \pm 0.004}$ | $\mathbf{0.807 \pm 0.006}$ | $0.726 \pm 0.001$ |
| RewardMATH | In-Distribution | $0.971 \pm 0.003$ | $0.972 \pm 0.002$ | $\mathbf{1.602 \pm 0.011}$ | $1.377 \pm 0.007$ |
| | Token-Level Shift | $\mathbf{0.988 \pm 0.003}$ | $0.515 \pm 0.007$ | $\mathbf{1.667 \pm 0.017}$ | $1.035 \pm 0.011$ |
| | Domain Shift | $0.505 \pm 0.012$ | $0.517 \pm 0.001$ | $\mathbf{0.755 \pm 0.008}$ | $0.604 \pm 0.004$ |

reports the accuracy and absolute reward margin of reward models for each evaluation category. See Tables 2, 3, 4, and 5 and Figures 9 and 10 in Appendix F for a per evaluation dataset and language model breakdown of the results.

**IM-RMs are less robust than EX-RMs to token-level distribution shifts.** Recall, our theoretical analysis (Section 4) indicates that IM-RMs are more sensitive than EX-RMs to superficial token-level cues. If this is indeed the case, then one would expect IM-RMs to underperform EX-RMs when subject to token-level distribution shifts. On the other hand, EX-RMs should not enjoy a distinct advantage under domain shifts. The empirical results match these expectations.[2] Moreover, the in-distribution accuracy of IM-RMs in the general chat setting is consistently lower than that of EX-RMs. We attribute this to in-distribution evaluation being closer to a token-level shift than to a domain shift. Namely, in-distribution test examples share semantic structure with training examples but take on different surface forms.

**EX-RMs induce a higher reward margin.** Table 1 highlights an additional benefit of EX-RMs over IM-RMs: EX-RMs induce a higher absolute reward margin. This was recently shown to yield a better optimization landscape for reinforcement learning (Razin et al., 2024b; 2025b).[3]

---

[2]Although EX-RMs are more robust than IM-RMs to token-level distribution shifts, they do suffer a drop in accuracy under the general chat setting (as similarly observed in Liu et al. (2025); Wu et al. (2025b)).

[3]Specifically, for a reward model $r$, prompt $\mathbf{x}$, and responses $\mathbf{y}, \mathbf{y}'$, the absolute reward margin is defined by $|r(\mathbf{x}, \mathbf{y}) - r(\mathbf{x}, \mathbf{y}')|$. Razin et al. (2024b; 2025b) proved that low reward variance, which is equivalent to the expected squared reward margin, leads to a flat objective landscape that hinders policy gradient optimization.

**Evidence against alternative hypotheses.** Finally, we provide evidence against two alternative candidate sources for the generalization gap between EX-RMs and IM-RMs, aside from the one already ruled out in Section 3. First, the reward of an EX-RM is based on the hidden representation of the whole response, whereas IM-RMs depend also on the hidden representations of intermediate tokens in the response. Intuitively, the hidden representations of intermediate tokens may be misleading since they do not capture the full meaning of the response. Second, the reward of an IM-RM is shifted by the log probability of a reference distribution, which is not the case for EX-RMs. Figure 7 in Appendix F.3 demonstrates that these differences do not explain the generalization gap by considering EX-RMs trained over the hidden representations of all intermediate tokens and IM-RMs without a reference distribution.

## 6 CONCLUSION

Reward models are a key component in language model post-training and inference pipelines. Yet, the comparative advantages and disadvantages of different reward model types are poorly understood. In this work, we mostly focused on two prevalent reward model types: *explicit reward models (EX-RMs)* and *implicit reward models (IM-RMs)*. Through theory and experiments, we established that IM-RMs rely more strongly on superficial token-level cues. Consequently, they typically generalize worse than EX-RMs under token-level distribution shifts, as well as in-distribution. This corroborates and provides an explanation for existing empirical findings on the relative benefits of EX-RMs (Singhal et al., 2024; Lin et al., 2024; Lambert et al., 2025; Swamy et al., 2025). We also provided evidence against an alternative hypothesis, by which the generalization gap between EX-RMs and IM-RMs stems from IM-RMs needing to learn to both generate and verify the quality responses, whereas EX-RMs only need to learn to verify. Overall, our work highlights that seemingly minor design choices can substantially impact how reward models generalize. As elaborated below, we hope its insights will motivate research into the implicit biases of different reward model types and facilitate enhancing their robustness.

### 6.1 LIMITATIONS AND FUTURE WORK

**Theoretical analysis.** Section 4 included a couple of simplifying assumptions. Namely, we assumed that hidden representations are fixed and Theorem 2 (in Appendix B) also required responses to be of length one. Although Section 5 empirically demonstrated that the conclusions of our theory apply when all reward model parameters are trained and responses are of arbitrary length, alleviating these restrictions may yield further insights into how reward models generalize.

**Factors influencing generalization.** We highlighted one cause for the difference in generalization between EX-RMs and IM-RMs — a stronger reliance of IM-RMs on token-level cues. However, there are likely additional factors that affect their generalization. In particular, while EX-RMs are more robust to token-level shifts, our experiments show that IM-RMs can generalize better under other types of distribution shifts. Investigating whether there are cases in which IM-RMs consistently outperform EX-RMs and why is left to future work.

**Beyond accuracy.** As customary, we primarily measured reward model generalization via accuracy (*cf.* Lin et al. (2024); Lambert et al. (2025)). While accuracy is an important measure, it is not the only quantity that determines the effectiveness of a reward model (Chen et al., 2024; Wen et al., 2025; Razin et al., 2025b). Exploring how different reward model types compare across a broader set of evaluation criteria remains a valuable direction for future work.

**Reward model types.** Our work covers three common reward model types: EX-RMs, IM-RMs, and a generative reward model variant (Appendix D; *cf.* Zhang et al. (2025)). We hope that it will encourage studying the implicit biases introduced by additional types, *e.g.*, reward models that provide rewards on intermediate steps of a response (Uesato et al., 2022; Lightman et al., 2024).

## ACKNOWLEDGEMENTS

We thank Eshbal Hezroni for aid in preparing illustrative figures and Peter Henderson for providing feedback on the manuscript. NR is supported by Princeton Language and Intelligence (PLI) and the

Zuckerman STEM Leadership Program. SA acknowledges funding from ONR, Schmidt Science, and OpenAI.

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

## A    REPRODUCIBILITY STATEMENT

Our theoretical results (Sections 3.1 and 4 and Appendices B and D) are accompanied by complete proofs in Appendix E. All information required to reproduce the experiments is provided in Sections 3.2 and 5 and Appendix G. Our code implementation is available at https://github.com/princeton-pli/exrm-vs-imrm.

## B    GENERALIZATION GAP BETWEEN EX-RMS AND IM-RMS

The goal of this appendix is to provide a concrete setting in which IM-RMs provably generalize worse than EX-RMs, due to the stronger reliance on token-level cues (identified in Section 4.1). Alongside assuming that the hidden representations are fixed (Assumption 1), we consider the case where responses in the training set are of length one. Furthermore, to ensure a fair comparison between EX-RMs and IM-RMs, we require that both are able to perfectly fit the training set.

**Assumption 2.** Responses in the training set $\mathcal{D}_\mathcal{T}$ are of length one.

**Assumption 3.** There exist $\theta_{\mathrm{EX}}$ and $\theta_{\mathrm{IM}}$ such that the corresponding EX-RM and IM-RM achieve perfect accuracy over the training set $\mathcal{D}_\mathcal{T}$, i.e., $\mathrm{acc}_{\mathcal{D}_\mathcal{T}}\big(r_{\theta_{\mathrm{EX}}}\big) = \mathrm{acc}_{\mathcal{D}_\mathcal{T}}\big(r_{\theta_{\mathrm{IM}}}\big) = 1$.

Under these conditions, Theorem 2 establishes that an IM-RM trained via gradient descent does not generalize to unseen tokens — it achieves trivial accuracy over any evaluation set containing responses that did not appear in the training set $\mathcal{D}_\mathcal{T}$. This inability to generalize occurs regardless of the structure of hidden representations or the initial unembedding matrix. By contrast, an EX-RM generalizes successfully to unseen tokens if the hidden representations are well-structured. Namely, let $\mathbf{u}^* \in \mathbb{R}^D$ be the following max-margin separator over hidden representations in $\mathcal{D}_\mathcal{T}$:

$$\mathbf{u}^* = \mathrm{argmin}_{\mathbf{u}\in\mathbb{R}^D} \|\mathbf{u}\|^2 \text{ s.t. } \forall (\mathbf{x},\mathbf{y}^+,\mathbf{y}^-) \in \mathcal{D}_\mathcal{T} : \big\langle \mathbf{u}, \mathbf{h}_{\mathbf{x},\mathbf{y}^+} - \mathbf{h}_{\mathbf{x},\mathbf{y}^-} \big\rangle \geq 1 \,. \tag{6}$$

The EX-RM will rank correctly any pair of responses that $\mathbf{u}^*$ ranks correctly.

**Theorem 2.** *Suppose we train an EX-RM and an IM-RM via gradient descent over the training set $\mathcal{D}_\mathcal{T}$ with learning rate $\eta < 2B^{-2}\min\{\beta^{-2},1\}$, where $B$ is the maximal hidden representation norm in $\mathcal{D}_\mathcal{T}$, i.e., $B := \max_{(\mathbf{x},\mathbf{y}^+,\mathbf{y}^-)\in\mathcal{D}_\mathcal{T},\mathbf{v}\in\{\mathbf{x},(\mathbf{x},\mathbf{y}^+),(\mathbf{x},\mathbf{y}^-)\}} \|\mathbf{h}_\mathbf{v}\|$. Denote by $\theta(t+1) := \theta(t) - \eta\nabla\mathcal{L}(r_{\theta(t)})$ the gradient descent iterates, for $t = 0,1,\ldots$, where $\theta$ stands for either $\theta_{\mathrm{EX}}$ or $\theta_{\mathrm{IM}}$, the IM-RM reference distribution is $\pi_{\mathrm{ref}} = \pi_{\theta_{\mathrm{IM}}(0)}$, and the loss $\mathcal{L}$ is defined in Equation (3). Then, under Assumptions 1, 2, and 3, for all initializations $\theta_{\mathrm{EX}}(0),\theta_{\mathrm{IM}}(0)$ and finite evaluation sets $\mathcal{D}_\mathcal{E}$ that contain preferences $(\mathbf{x},\mathbf{y}^+,\mathbf{y}^-)$, in which $\mathbf{x} \in \mathcal{V}^*$ and $\mathbf{y}^+,\mathbf{y}^- \in \mathcal{V}$ are responses that do not appear in $\mathcal{D}_\mathcal{T}$, the following hold.*

- ***Both the EX-RM and IM-RM perfectly fit the training set:*** *That is, $\lim_{t\to\infty}\mathcal{L}(r_{\theta_{\mathrm{EX}}(t)}) = \lim_{t\to\infty}\mathcal{L}(r_{\theta_{\mathrm{IM}}(t)}) = 0$ and $\lim_{t\to\infty}\mathrm{acc}_{\mathcal{D}_\mathcal{T}}(r_{\theta_{\mathrm{EX}}(t)}) = \lim_{t\to\infty}\mathrm{acc}_{\mathcal{D}_\mathcal{T}}(r_{\theta_{\mathrm{IM}}(t)}) = 1$.*

- ***The IM-RM fails to generalize to unseen tokens:*** $\mathrm{acc}_{\mathcal{D}_\mathcal{E}}(r_{\theta_{\mathrm{IM}}(t)}) = 0.5$ *for all $t \geq 0$.*

- ***The EX-RM can generalize via hidden representations:*** *Let $\mathbf{u}^*$ be the max-margin separator defined in Equation (6). Then, there exists a time $t_0 \geq 0$ such that for all $t \geq t_0$:*

$$\mathrm{acc}_{\mathcal{D}_\mathcal{E}}(r_{\theta_{\mathrm{EX}}(t)}) \geq \frac{\big|\{(\mathbf{x},\mathbf{y}^+,\mathbf{y}^-) \in \mathcal{D}_\mathcal{E} : \langle\mathbf{u}^*,\mathbf{h}_{\mathbf{x},\mathbf{y}^+}\rangle > \langle\mathbf{u}^*,\mathbf{h}_{\mathbf{x},\mathbf{y}^-}\rangle\}\big|}{|\mathcal{D}_\mathcal{E}|}\,.$$

*Proof sketch (full proof in Appendix E.6).* With fixed hidden representations, the loss of an EX-RM and the loss of an IM-RM can be framed as logistic regression problems over different input spaces. Fitting of the training set thus follows by standard convex optimization results. We then specialize the learning dynamics of an IM-RM (Equation (5)) to the case of single-token responses and show that the difference between the rewards of two unseen tokens is constant through training. This implies that $\mathrm{acc}_{\mathcal{D}_\mathcal{E}}(r_{\theta_{\mathrm{IM}}(t)})$ remains at its initial trivial value of $0.5$. Lastly, by applying the seminal result of Soudry et al. (2018), we get that the linear head of the EX-RM converges in direction to $\mathbf{u}^*$. This yields the guarantee on $\mathrm{acc}_{\mathcal{D}_\mathcal{E}}(r_{\theta_{\mathrm{EX}}(t)})$. □

## C   RELATED WORK

**Reward models for language model post-training and inference.** In real-world applications, it is rarely feasible to evaluate the quality generated responses via rule-based rewards. As a result, reward models have been extensively used in the language model ecosystem for training via reinforcement learning (Ziegler et al., 2019; Ouyang et al., 2022; Achiam et al., 2023; Dubey et al., 2024; Qwen et al., 2024; Team et al., 2024), labeling preferences in direct alignment algorithms (Dong et al., 2024; Meng et al., 2024; Adler et al., 2024), rejection sampling (Gulcehre et al., 2023; Dong et al., 2023), data filtering (Dubey et al., 2024; Qwen et al., 2024; Albalak et al., 2024), and inference-time scaling (Cobbe et al., 2021; Wu et al., 2025a; Snell et al., 2025).

**Analyses of reward models.** Prior analyses mostly bounded the sample complexity for estimating a ground truth reward, under various technical conditions (Pacchiano et al., 2023; Zhu et al., 2023; Wang et al., 2023; Zhan et al., 2023; Ji et al., 2023; Du et al., 2024; Xiong et al., 2024; Qiu et al., 2024; Das et al., 2024; Ji et al., 2024; Scheid et al., 2024; Gaur et al., 2024; Zhan et al., 2024; Wu and Sun, 2024; Li et al., 2024; Huang et al., 2025; Sun et al., 2025). An additional line of research considered properties of a reward model that benefit robustness (Wang et al., 2024b; Hong et al., 2025), compatibility with a given inference procedure (Chow et al., 2025; Balashankar et al., 2025), or the optimization landscape for reinforcement learning (Razin et al., 2024b; 2025b). However, the works mentioned above do not account for the difference between reward model types or the effect of their particular parameterizations on generalization, which is the goal of this study.

Most relevant in our context are Im and Li (2025); Razin et al. (2025a); Shi et al. (2025). Similarly to Section 4.1, Im and Li (2025) and Razin et al. (2025a) analyzed the learning dynamics of IM-RMs, but for other purposes. Specifically, our work focuses on generalization, whereas Razin et al. (2025a) addressed an optimization issue that causes the reward assigned to chosen responses to decrease. Regarding Im and Li (2025), under conditions similar to those of Theorem 2, they proved that IM-RMs can generalize well to unseen prompts if the responses used for training and evaluation are the same. In contrast, Theorem 2 establishes that IM-RMs fail to generalize when the evaluation responses do not appear in the training set — a more realistic scenario. Lastly, Shi et al. (2025) constructed a setting in which EX-RMs enjoy a better sample complexity than IM-RMs (Section 4 therein). Though, their result requires nonstandard reward model parameterizations and a reward estimation method tailored to a specific ground truth reward. While our analysis also operates under simplifying assumptions, it identifies a cause for the generalization gap observed in practice between EX-RMs and IM-RMs, as we extensively verify empirically (Section 5).

**DPO vs RLHF.** The question of why IM-RMs often generalize worse than EX-RMs is closely related to, yet distinct from, comparisons of DPO (Rafailov et al., 2023) and RLHF (Ouyang et al., 2022). Specifically, DPO corresponds to the language model underlying an IM-RM and RLHF refers to first training an EX-RM and then optimizing a language model based on it. Swamy et al. (2025) argued that DPO usually underperforms RLHF due to the presence of generation-verification gaps in many practical scenarios. Our results (Section 3) show that, while such gaps may underlie performance differences between language models trained with DPO and RLHF, they do not explain the difference in generalization between IM-RMs and EX-RMs (as measured by accuracy). Furthermore, the analysis in Section 4 can be interpreted as suggesting another cause for the performance difference between DPO and RLHF: the former suffers from a reliance on superficial token-level cues.

**Learning dynamics of neural networks.** In Section 4.1, we characterized how the reward assigned to prompt-response pairs changes due to a gradient update. Analogous approaches have been valuable both in theory, for studying the effect of language model post-training (Im and Li, 2025; Razin et al., 2025a; Ren and Sutherland, 2025), and in practice, for identifying mislabeled examples (Pruthi et al., 2020) and developing data selection algorithms (Xia et al., 2024). More broadly, analyzing the trajectory of gradient-based training is a fundamental tool in the vast implicit bias literature. There, the focus is typically on understanding why overparameterized neural networks tend to generalize well, despite the existence of parameter assignments that do not (Saxe et al., 2014; Gunasekar et al., 2017; 2018; Soudry et al., 2018; Arora et al., 2019; Gidel et al., 2019; Goldt et al., 2019; Lampinen and Ganguli, 2019; Razin and Cohen, 2020; Razin et al., 2021; 2022; Berthier, 2023; Cohen-Karlik et al., 2023; Ren et al., 2023; Razin et al., 2024a; Chou et al., 2024; Zhang et al., 2024; Slutzky et al., 2025; Vasudeva et al., 2025). We refer to Vardi (2023) for a survey of the field.

## D  EXPLICIT GENERATIVE REWARD MODELS

A nascent reward model variant, proposed in Zhang et al. (2025), rewards responses by asking a language model $\pi_\theta$ to assess their quality. We refer to this type of reward models as *explicit generative reward models (EX-GRMs)*. Specifically, for a prompt-response pair $(\mathbf{x}, \mathbf{y})$, EX-GRMs receive as input $I[\mathbf{x}, \mathbf{y}] \in \mathcal{V}^*$, which is some textual format that requests the model to verify whether $\mathbf{y}$ is a good response to $\mathbf{x}$.[4] For example, Zhang et al. (2025) concatenate to $(\mathbf{x}, \mathbf{y})$ the suffix "Is the answer correct (Yes/No)?". Then, the reward for $(\mathbf{x}, \mathbf{y})$ is taken to be the probability that the underlying language model assigns to the token Yes, *i.e.*:

$$r_{\theta_G}(\mathbf{x}, \mathbf{y}) = \pi_{\theta_G}(\text{Yes}|I[\mathbf{x}, \mathbf{y}]),$$

where $\theta_G = \theta$ denotes the trainable parameters of the EX-GRM.

Instead of the Bradley-Terry log-likelihood loss (Equation (3)), Zhang et al. (2025) suggested an alternative loss for EX-GRMs:[5]

$$\mathcal{L}^G(r_{\theta_G}) := \frac{1}{|\mathcal{D}_\mathcal{T}|} \sum\nolimits_{(\mathbf{x}, \mathbf{y}^+, \mathbf{y}^-) \in \mathcal{D}_\mathcal{T}} - \ln \pi_{\theta_G}\big(\text{Yes}|I[\mathbf{x}, \mathbf{y}^+]\big) - \ln \pi_{\theta_G}\big(\text{No}|I[\mathbf{x}, \mathbf{y}^-]\big). \quad (7)$$

In Appendix D.1, we extend the analysis of Section 4.1 to EX-GRMs. We show that, similarly to EX-RMs, the learning dynamics of EX-GRMs depends on responses primarily through their hidden representations. This suggests that EX-GRMs should also be more robust than IM-RMs to token-level distribution shifts. We corroborate this hypothesis empirically in Appendix F.

### D.1  LEARNING DYNAMICS

We characterize how performing a gradient update on the training example $(\mathbf{x}, \mathbf{y}^+, \mathbf{y}^-) \in \mathcal{D}_\mathcal{T}$ influences the reward that an EX-GRM assigns to an unseen prompt-response pair $(\bar{\mathbf{x}}, \bar{\mathbf{y}})$, *i.e.*:

$$\Delta r_{\theta_G}(\bar{\mathbf{x}}, \bar{\mathbf{y}}) := r_{\theta_G - \eta \nabla \ell^G_{\theta_G}(\mathbf{x}, \mathbf{y}^+, \mathbf{y}^-)}(\bar{\mathbf{x}}, \bar{\mathbf{y}}) - r_{\theta_G}(\bar{\mathbf{x}}, \bar{\mathbf{y}}),$$

where $\eta \in \mathbb{R}_{>0}$ is a learning rate and

$$\ell^G_{\theta_G}(\mathbf{x}, \mathbf{y}^+, \mathbf{y}^-) := - \ln \pi_{\theta_G}\big(\text{Yes}|I[\mathbf{x}, \mathbf{y}^+]\big) - \ln \pi_{\theta_G}\big(\text{No}|I[\mathbf{x}, \mathbf{y}^-]\big)$$

denotes the EX-GRM loss over $(\mathbf{x}, \mathbf{y}^+, \mathbf{y}^-)$. By a Taylor approximation of $r_{\theta_G}(\bar{\mathbf{x}}, \bar{\mathbf{y}})$ around $\theta_G$, we may write the change in reward as:

$$\Delta r_{\theta_G}(\bar{\mathbf{x}}, \bar{\mathbf{y}}) = -\eta \left\langle \nabla r_{\theta_G}(\bar{\mathbf{x}}, \bar{\mathbf{y}}), \nabla \ell^G_{\theta_G}(\mathbf{x}, \mathbf{y}^+, \mathbf{y}^-) \right\rangle + \mathcal{O}(\eta^2).$$

As in Section 4.1, we assume that hidden representations are fixed during training (Assumption 1), in which case the trainable parameters of the EX-GRM are $\theta_G = \mathbf{U}$, where $\mathbf{U}$ is the unembedding matrix of $\pi_{\theta_G}$. Under this assumption, the change in reward for EX-GRMs is given by (derivation in Appendix E.5):

$$\Delta r_{\theta_G}(\bar{\mathbf{x}}, \bar{\mathbf{y}}) = \pi_{\theta_G}(\text{Yes}|I[\bar{\mathbf{x}}, \bar{\mathbf{y}}])\Big(\gamma(\mathbf{y}^+) \cdot \big\langle \mathbf{h}_{I[\bar{\mathbf{x}}, \bar{\mathbf{y}}]}, \mathbf{h}_{I[\mathbf{x}, \mathbf{y}^+]} \big\rangle + \gamma(\mathbf{y}^-) \cdot \big\langle \mathbf{h}_{I[\bar{\mathbf{x}}, \bar{\mathbf{y}}]}, \mathbf{h}_{I[\mathbf{x}, \mathbf{y}^-]} \big\rangle \Big)$$
$$\cdot \eta + \mathcal{O}(\eta^2), \quad (8)$$

where the coefficients $\gamma(\mathbf{y}^+) \in [0, 2]$ and $\gamma(\mathbf{y}^-) \in [-2, 1]$ are defined as:

$$\gamma(\mathbf{y}^+) = 1 - \pi_{\theta_G}(\text{Yes}|I[\bar{\mathbf{x}}, \bar{\mathbf{y}}]) - \pi_{\theta_G}(\text{Yes}|I[\mathbf{x}, \mathbf{y}^+]) + \big\langle \pi_{\theta_G}(\cdot|I[\bar{\mathbf{x}}, \bar{\mathbf{y}}]), \pi_{\theta_G}(\cdot|I[\mathbf{x}, \mathbf{y}^+]) \big\rangle,$$

$$\gamma(\mathbf{y}^-) = -\pi_{\theta_G}(\text{No}|I[\bar{\mathbf{x}}, \bar{\mathbf{y}}]) - \pi_{\theta_G}(\text{Yes}|I[\mathbf{x}, \mathbf{y}^-]) + \big\langle \pi_{\theta_G}(\cdot|I[\bar{\mathbf{x}}, \bar{\mathbf{y}}]), \pi_{\theta_G}(\cdot|I[\mathbf{x}, \mathbf{y}^-]) \big\rangle,$$

with $\pi_{\theta_G}(\cdot|\mathbf{z})$ denoting the vector of probabilities that $\pi_{\theta_G}$ assigns to tokens conditioned on $\mathbf{z}$.

Similarly to EX-RMs (Equation (4)), and in contrast to IM-RMs (Equation (5)), the change in reward for EX-GRMs depends on $\bar{\mathbf{y}}$, $\mathbf{y}^+$, and $\mathbf{y}^-$ primarily through the hidden representations of the corresponding inputs (*i.e.*, $\mathbf{h}_{I[\bar{\mathbf{x}}, \bar{\mathbf{y}}]}$, $\mathbf{h}_{I[\mathbf{x}, \mathbf{y}^+]}$, and $\mathbf{h}_{I[\mathbf{x}, \mathbf{y}^-]}$). Notably, since $\gamma(\mathbf{y}^+) \geq 0$, the contribution of

---

[4]The input $I[\mathbf{x}, \mathbf{y}]$ can optionally include chain-of-thought tokens.
[5]Zhang et al. (2025) include an additional $-\lambda \cdot \ln \pi_{\theta_G}(\mathbf{y}^+|\mathbf{x})$ loss term, with $\lambda > 0$, that encourages the model to retain its response generation capabilities.

$\langle \mathbf{h}_{I[\bar{\mathbf{x}},\bar{\mathbf{y}}]}, \mathbf{h}_{I[\mathbf{x},\mathbf{y}^+]} \rangle$ mirrors that of $\langle \mathbf{h}_{\bar{\mathbf{x}},\bar{\mathbf{y}}}, \mathbf{h}_{\mathbf{x},\mathbf{y}^+} \rangle$ in the EX-RM dynamics: it increases the reward of $(\bar{\mathbf{x}}, \bar{\mathbf{y}})$ when the hidden representations corresponding to $(\bar{\mathbf{x}}, \bar{\mathbf{y}})$ and $(\mathbf{x}, \mathbf{y}^+)$ are aligned. The contribution of the term involving $\gamma(\mathbf{y}^-)$ may differ from the analogous term in the EX-RM dynamics since $\gamma(\mathbf{y}^-)$ can be positive. Nonetheless, EX-GRMs, like EX-RMs, are expected be more robust than IM-RMs to superficial token-level distribution shifts. Appendix F empirically demonstrates that this is indeed the case.

## E    DEFERRED PROOFS

### E.1    PROOF OF THEOREM 1

For a prompt $\mathbf{x} \in \mathcal{X}$, we define $\pi(\cdot|\mathbf{x})$ by:

$$\pi(\mathbf{y}|\mathbf{x}) := \begin{cases} \frac{1}{Z(\mathbf{x})} \pi_{\mathrm{ref}}(\mathbf{y}|\mathbf{x}) \cdot \exp(\delta/\beta) & , \ \mathbf{y} \in \mathcal{C}(\mathbf{x}) \\ \frac{1}{Z(\mathbf{x})} \pi_{\mathrm{ref}}(\mathbf{y}|\mathbf{x}) & , \ \mathbf{y} \in \mathcal{V}^* \setminus \mathcal{C}(\mathbf{x}) \end{cases},$$

where

$$Z(\mathbf{x}) := \sum_{\mathbf{y}^+ \in \mathcal{C}(\mathbf{x})} \pi_{\mathrm{ref}}(\mathbf{y}^+|\mathbf{x}) \cdot \exp(\delta/\beta) + \sum_{\mathbf{y}^- \in \mathcal{V}^* \setminus \mathcal{C}(\mathbf{x})} \pi_{\mathrm{ref}}(\mathbf{y}^-|\mathbf{x})$$

is a normalization constant that ensures $\pi(\cdot|\mathbf{x})$ is a valid distribution. The probability that $\pi$ assigns to any other sequence of tokens can be defined arbitrarily (as long as it is consistent with the probabilities defined above). Since $r_{\mathrm{IM}}(\mathbf{x}, \mathbf{y}) = \beta(\ln \pi(\mathbf{y}|\mathbf{x}) - \ln \pi_{\mathrm{ref}}(\mathbf{y}|\mathbf{x}))$ for all $\mathbf{x} \in \mathcal{X}, \mathbf{y} \in \mathcal{V}^*$, where $r_{\mathrm{IM}}$ is the IM-RM induced by $\pi$, we have that:

$$r_{\mathrm{IM}}(\mathbf{x}, \mathbf{y}) = \begin{cases} \delta - \beta \ln Z(\mathbf{x}) & , \ \mathbf{y} \in \mathcal{C}(\mathbf{x}) \\ -\beta \ln Z(\mathbf{x}) & , \ \mathbf{y} \in \mathcal{V}^* \setminus \mathcal{C}(\mathbf{x}) \end{cases}.$$

Clearly, $r_{\mathrm{IM}}$ is a verifier with margin $\delta$ for $(\mathcal{X}, \mathcal{C})$ since $r_{\mathrm{IM}}(\mathbf{x}, \mathbf{y}^+) = r_{\mathrm{IM}}(\mathbf{x}, \mathbf{y}^-) + \delta$ for all $\mathbf{x} \in \mathcal{X}$, $\mathbf{y}^+ \in \mathcal{C}(\mathbf{x})$, and $\mathbf{y}^- \in \mathcal{V}^* \setminus \mathcal{C}(\mathbf{x})$.

Now, notice that $Z(\mathbf{x}) > 1$ since it is a sum over the probabilities $\pi_{\mathrm{ref}}(\mathbf{y}|\mathbf{x})$, for $\mathbf{y} \in \mathcal{V}^*$, up to terms corresponding to $\mathbf{y} \in \mathcal{C}(\mathbf{x})$ being multiplied by $\exp(\delta/\beta) > 1$. As a result, for any $\mathbf{x} \in \mathcal{X}$ and $\mathbf{y} \in \mathcal{C}(\mathbf{x})$ it holds that:

$$\pi(\mathbf{y}|\mathbf{x}) = \frac{1}{Z(\mathbf{x})} \pi_{\mathrm{ref}}(\mathbf{y}|\mathbf{x}) \cdot \exp(\delta/\beta) \leq \pi_{\mathrm{ref}}(\mathbf{y}|\mathbf{x}) \cdot \exp(\delta/\beta).$$

Summing over responses in $\mathcal{C}(\mathbf{x})$, we conclude:

$$\pi(\mathcal{C}(\mathbf{x})|\mathbf{x}) \leq \pi_{\mathrm{ref}}(\mathcal{C}(\mathbf{x})|\mathbf{x}) \cdot \exp(\delta/\beta).$$

$\square$

### E.2    PROOF OF COROLLARY 1

By Theorem 1, there exist a distribution $\pi$ and corresponding IM-RM $r_{\mathrm{IM}}$ such that $r_{\mathrm{IM}}$ is a verifier with margin $\delta \in \mathbb{R}_{>0}$ for $(\mathcal{X}, \mathcal{C})$, although for all $\mathbf{x} \in \mathcal{X}$ it holds that:

$$\exp(-\delta/\beta) \cdot \pi(\mathcal{C}(\mathbf{x})|\mathbf{x}) \leq \pi_{\mathrm{ref}}(\mathcal{C}(\mathbf{x})|\mathbf{x}).$$

We show that, since $\pi_{\mathrm{ref}}$ is not an efficient generator for $(\mathcal{X}, \mathcal{C})$, the distribution $\pi$ is not an efficient generator for $(\mathcal{X}, \mathcal{C})$ either. Assume by way of contradiction that this is not the case, i.e., that $\pi$ is an efficient generator for $(\mathcal{X}, \mathcal{C})$. Let $k \in \mathbb{N}$ and $\alpha \in \mathbb{R}_{>0}$ be such that $\pi(\mathcal{C}(\mathbf{x})|\mathbf{x}) \geq \alpha^{-1}|\mathbf{x}|^{-k}$ for all $\mathbf{x} \in \mathcal{X}$. Defining $\gamma := \alpha \cdot \exp(\delta/\beta)$, it follows that for all $\mathbf{x} \in \mathcal{X}$:

$$\pi_{\mathrm{ref}}(\mathcal{C}(\mathbf{x})|\mathbf{x}) \geq \exp(-\delta/\beta) \cdot \pi(\mathcal{C}(\mathbf{x})|\mathbf{x}) \geq \exp(-\delta/\beta) \cdot \alpha^{-1}|\mathbf{x}|^{-k} = \gamma^{-1}|\mathbf{x}|^{-k},$$

i.e., $\pi_{\mathrm{ref}}$ is an efficient generator for $(\mathcal{X}, \mathcal{C})$ — a contradiction.

$\square$

### E.3 Derivation of Explicit Reward Model Learning Dynamics (Equation (4))

Under Assumption 1, the trainable parameters of the EX-RM are $\theta_{\mathrm{EX}} = \mathbf{u}$. Thus, the loss gradient for $(\mathbf{x}, \mathbf{y}^+, \mathbf{y}^-) \in \mathcal{D}_\mathcal{T}$ with respect to $\theta_{\mathrm{EX}}$ is given by:

$$\nabla \ell_{\theta_{\mathrm{EX}}}(\mathbf{x}, \mathbf{y}^+, \mathbf{y}^-) = -g(\theta_{\mathrm{EX}}) \cdot \left( \nabla r_{\theta_{\mathrm{EX}}}(\mathbf{x}, \mathbf{y}^+) - \nabla r_{\theta_{\mathrm{EX}}}(\mathbf{x}, \mathbf{y}^-) \right)$$

$$= -g(\theta_{\mathrm{EX}}) \cdot \left( \mathbf{h}_{\mathbf{x},\mathbf{y}^+} - \mathbf{h}_{\mathbf{x},\mathbf{y}^-} \right),$$

where $g(r_{\theta_{\mathrm{EX}}}) := -\ell'_{\theta_{\mathrm{EX}}}(\mathbf{x}, \mathbf{y}^+, \mathbf{y}^-) = \sigma(r_{\theta_{\mathrm{EX}}}(\mathbf{x}, \mathbf{y}^-) - r_{\theta_{\mathrm{EX}}}(\mathbf{x}, \mathbf{y}^+)) > 0$. Equation (4) then follows by:

$$\Delta r_{\theta_{\mathrm{EX}}}(\bar{\mathbf{x}}, \bar{\mathbf{y}}) = r_{\theta_{\mathrm{EX}} - \eta \nabla \ell_{\theta_{\mathrm{EX}}}(\mathbf{x}, \mathbf{y}^+, \mathbf{y}^-)}(\bar{\mathbf{x}}, \bar{\mathbf{y}}) - r_{\theta_{\mathrm{EX}}}(\bar{\mathbf{x}}, \bar{\mathbf{y}})$$

$$= \left\langle \mathbf{u} - \eta \nabla \ell_{\theta_{\mathrm{EX}}}(\mathbf{x}, \mathbf{y}^+, \mathbf{y}^-), \mathbf{h}_{\bar{\mathbf{x}}, \bar{\mathbf{y}}} \right\rangle - \left\langle \mathbf{u}, \mathbf{h}_{\bar{\mathbf{x}}, \bar{\mathbf{y}}} \right\rangle$$

$$= \left\langle \mathbf{h}_{\bar{\mathbf{x}}, \bar{\mathbf{y}}}, \mathbf{h}_{\mathbf{x},\mathbf{y}^+} - \mathbf{h}_{\mathbf{x},\mathbf{y}^-} \right\rangle \cdot \eta g(\theta_{\mathrm{EX}}).$$

### E.4 Derivation of Implicit Reward Model Learning Dynamics (Equation (5))

Equation (5) follows by steps similar to those used for proving Theorem 7 in Razin et al. (2025a), where the difference stems from the hidden representations being fixed in our case (Assumption 1). In particular, under Assumption 1, the trainable parameters of the IM-RM are $\theta_{\mathrm{IM}} = \mathbf{U}$. Thus, the loss gradient for $(\mathbf{x}, \mathbf{y}^+, \mathbf{y}^-) \in \mathcal{D}_\mathcal{T}$ with respect to $\theta_{\mathrm{IM}}$ is given by:

$$\nabla \ell_{\theta_{\mathrm{IM}}}(\mathbf{x}, \mathbf{y}^+, \mathbf{y}^-) = -g(\theta_{\mathrm{IM}}) \cdot \left( \nabla r_{\theta_{\mathrm{IM}}}(\mathbf{x}, \mathbf{y}^+) - \nabla r_{\theta_{\mathrm{IM}}}(\mathbf{x}, \mathbf{y}^-) \right)$$

$$= -g(\theta_{\mathrm{IM}}) \cdot \left( \nabla \beta \ln \frac{\pi_{\theta_{\mathrm{IM}}}(\mathbf{y}^+|\mathbf{x})}{\pi_{\mathrm{ref}}(\mathbf{y}^+|\mathbf{x})} - \nabla \beta \ln \frac{\pi_{\theta_{\mathrm{IM}}}(\mathbf{y}^-|\mathbf{x})}{\pi_{\mathrm{ref}}(\mathbf{y}^-|\mathbf{x})} \right)$$

$$= -g(\theta_{\mathrm{IM}}) \beta \cdot \left( \nabla \ln \pi_{\theta_{\mathrm{IM}}}(\mathbf{y}^+|\mathbf{x}) - \nabla \ln \pi_{\theta_{\mathrm{IM}}}(\mathbf{y}^-|\mathbf{x}) \right),$$

where $g(\theta_{\mathrm{IM}}) := -\ell'_{\theta_{\mathrm{IM}}}(\mathbf{x}, \mathbf{y}^+, \mathbf{y}^-) = \sigma(r_{\theta_{\mathrm{IM}}}(\mathbf{x}, \mathbf{y}^-) - r_{\theta_{\mathrm{IM}}}(\mathbf{x}, \mathbf{y}^+)) > 0$. Furthermore, the reward gradient for $(\bar{\mathbf{x}}, \bar{\mathbf{y}})$ is:

$$\nabla r_{\theta_{\mathrm{IM}}}(\bar{\mathbf{x}}, \bar{\mathbf{y}}) = \nabla \beta \ln \frac{\pi_{\theta_{\mathrm{IM}}}(\bar{\mathbf{y}}|\bar{\mathbf{x}})}{\pi_{\mathrm{ref}}(\bar{\mathbf{y}}|\bar{\mathbf{x}})} = \beta \cdot \nabla \ln \pi_{\theta_{\mathrm{IM}}}(\bar{\mathbf{y}}|\bar{\mathbf{x}}).$$

Now, for any prompt $\mathbf{x}' \in \mathcal{V}^*$ and response $\mathbf{y}' \in \mathcal{V}^*$:

$$\nabla \ln \pi_{\theta_{\mathrm{IM}}}(\mathbf{y}'|\mathbf{x}') = \sum_{k=1}^{|\mathbf{y}'|} \nabla \ln \pi_{\theta_{\mathrm{IM}}}(\mathbf{y}'_k|\mathbf{x}', \mathbf{y}'_{<k})$$

$$= \sum_{k=1}^{|\mathbf{y}'|} \nabla \left( \left\langle \mathbf{U}_{\mathbf{y}'_k}, \mathbf{h}_{\mathbf{x}',\mathbf{y}'_{<k}} \right\rangle - \ln \sum_{v \in \mathcal{V}} \exp\left( \left\langle \mathbf{U}_v, \mathbf{h}_{\mathbf{x}',\mathbf{y}'_{<k}} \right\rangle \right) \right)$$

$$= \sum_{k=1}^{|\mathbf{y}'|} \left( \mathbf{e}_{\mathbf{y}'_k} - \pi_{\theta_{\mathrm{IM}}}(\cdot|\mathbf{x}', \mathbf{y}'_{<k}) \right) \mathbf{h}_{\mathbf{x}',\mathbf{y}'_{<k}}^\top,$$

where $\mathbf{e}_v \in \mathbb{R}^{|\mathcal{V}|}$ denotes the standard basis vector corresponding to $v \in \mathcal{V}$ and $\pi_{\theta_{\mathrm{IM}}}(\cdot|\mathbf{x}', \mathbf{y}'_{<k})$ is the vector of probabilities that $\pi_{\theta_{\mathrm{IM}}}$ assigns to tokens conditioned on $(\mathbf{x}', \mathbf{y}'_{<k})$. Plugging this gradient expression into the expressions for $\nabla r_{\theta_{\mathrm{IM}}}(\bar{\mathbf{x}}, \bar{\mathbf{y}})$ and $\nabla \ell_{\theta_{\mathrm{IM}}}(\mathbf{x}, \mathbf{y}^+, \mathbf{y}^-)$ yields:

$$\left\langle \nabla r_{\theta_{\mathrm{IM}}}(\bar{\mathbf{x}}, \bar{\mathbf{y}}), -\nabla \ell_{\theta_{\mathrm{IM}}}(\mathbf{x}, \mathbf{y}^+, \mathbf{y}^-) \right\rangle$$

$$= \left\langle \sum_{k=1}^{|\bar{\mathbf{y}}|} (\mathbf{e}_{\bar{\mathbf{y}}_k} - \pi_{\theta_{\mathrm{IM}}}(\cdot|\bar{\mathbf{x}}, \bar{\mathbf{y}}_{<k})) \mathbf{h}_{\bar{\mathbf{x}}, \bar{\mathbf{y}}_{<k}}^\top, \sum_{l=1}^{|\mathbf{y}^+|} \left( \mathbf{e}_{\mathbf{y}_l^+} - \pi_{\theta_{\mathrm{IM}}}(\cdot|\mathbf{x}, \mathbf{y}_{<l}^+) \right) \mathbf{h}_{\mathbf{x}, \mathbf{y}_{<l}^+}^\top \right\rangle g(\theta_{\mathrm{IM}}) \beta^2$$

$$- \left\langle \sum_{k=1}^{|\bar{\mathbf{y}}|} (\mathbf{e}_{\bar{\mathbf{y}}_k} - \pi_{\theta_{\mathrm{IM}}}(\cdot|\bar{\mathbf{x}}, \bar{\mathbf{y}}_{<k})) \mathbf{h}_{\bar{\mathbf{x}}, \bar{\mathbf{y}}_{<k}}^\top, \sum_{l=1}^{|\mathbf{y}^-|} \left( \mathbf{e}_{\mathbf{y}_l^-} - \pi_{\theta_{\mathrm{IM}}}(\cdot|\mathbf{x}, \mathbf{y}_{<l}^-) \right) \mathbf{h}_{\mathbf{x}, \mathbf{y}_{<l}^-}^\top \right\rangle g(\theta_{\mathrm{IM}}) \beta^2$$

$$= \left( \sum_{k=1}^{|\bar{\mathbf{y}}|} \sum_{l=1}^{|\mathbf{y}^+|} \rho_{k,l}(\mathbf{y}^+) \cdot \left\langle \mathbf{h}_{\bar{\mathbf{x}}, \bar{\mathbf{y}}_{<k}}, \mathbf{h}_{\mathbf{x}, \mathbf{y}_{<l}^+} \right\rangle - \sum_{k=1}^{|\bar{\mathbf{y}}|} \sum_{l=1}^{|\mathbf{y}^-|} \rho_{k,l}(\mathbf{y}^-) \cdot \left\langle \mathbf{h}_{\bar{\mathbf{x}}, \bar{\mathbf{y}}_{<k}}, \mathbf{h}_{\mathbf{x}, \mathbf{y}_{<l}^-} \right\rangle \right) g(\theta_{\mathrm{IM}}) \beta^2,$$

where for all $\mathbf{v} \in \{\mathbf{y}^+, \mathbf{y}^-\}$, $k \in \{1, \ldots, |\bar{\mathbf{y}}|\}$, and $l \in \{1, \ldots, |\mathbf{v}|\}$:

$$\rho_{k,l}(\mathbf{v}) := \langle \mathbf{e}_{\bar{\mathbf{y}}_k} - \pi_{\theta_{\mathrm{IM}}}(\cdot|\bar{\mathbf{x}}, \bar{\mathbf{y}}_{<k}), \mathbf{e}_{\mathbf{v}_l} - \pi_{\theta_{\mathrm{IM}}}(\cdot|\mathbf{x}, \mathbf{v}_{<l}) \rangle$$
$$= \mathbb{1}[\bar{\mathbf{y}}_k = \mathbf{v}_l] - \pi_{\theta_{\mathrm{IM}}}(\bar{\mathbf{y}}_k|\mathbf{x}, \mathbf{v}_{<l}) - \pi_{\theta_{\mathrm{IM}}}(\mathbf{v}_l|\bar{\mathbf{x}}, \bar{\mathbf{y}}_{<k}) + \langle \pi_{\theta_{\mathrm{IM}}}(\cdot|\bar{\mathbf{x}}, \bar{\mathbf{y}}_{<k}), \pi_{\theta_{\mathrm{IM}}}(\cdot|\mathbf{x}, \mathbf{v}_{<l}) \rangle .$$

Equation (5) then follows by the above and the fact that:

$$\Delta r_{\theta_{\mathrm{IM}}}(\bar{\mathbf{x}}, \bar{\mathbf{y}}) = -\eta \langle \nabla r_{\theta_{\mathrm{IM}}}(\bar{\mathbf{x}}, \bar{\mathbf{y}}), \nabla \ell_{\theta_{\mathrm{IM}}}(\mathbf{x}, \mathbf{y}^+, \mathbf{y}^-) \rangle + \mathcal{O}(\eta^2) .$$

Lastly, to see that $\rho_{k,l}(\mathbf{v})$ resides within $[-2, 2]$ for all $\mathbf{v} \in \mathcal{V}^*$, $k \in \{1, \ldots, |\bar{\mathbf{y}}|\}$, and $l \in \{1, \ldots, |\mathbf{v}|\}$, notice that:

$$\left| \rho_{k,l}(\mathbf{v}) \right| = \left| \langle \mathbf{e}_{\bar{\mathbf{y}}_k} - \pi_{\theta_{\mathrm{IM}}}(\cdot|\bar{\mathbf{x}}, \bar{\mathbf{y}}_{<k}), \mathbf{e}_{\mathbf{v}_l} - \pi_{\theta_{\mathrm{IM}}}(\cdot|\mathbf{x}, \mathbf{v}_{<l}) \rangle \right|$$
$$\leq \left\| \mathbf{e}_{\bar{\mathbf{y}}_k} - \pi_{\theta_{\mathrm{IM}}}(\cdot|\bar{\mathbf{x}}, \bar{\mathbf{y}}_{<k}) \right\|_1 \cdot \left\| \mathbf{e}_{\mathbf{v}_l} - \pi_{\theta_{\mathrm{IM}}}(\cdot|\mathbf{x}, \mathbf{v}_{<l}) \right\|_\infty$$
$$\leq 2 \cdot 1$$
$$= 2 ,$$

where $\|\cdot\|_1$ and $\|\cdot\|_\infty$ denote the $\ell_1$ and $\ell_\infty$ norms, respectively.

### E.5 DERIVATION OF EXPLICIT GENERATIVE REWARD MODEL LEARNING DYNAMICS (EQUATION (8))

Under Assumption 1, the trainable parameters of the EX-GRM are $\theta_{\mathrm{G}} = \mathbf{U}$. Recall (Appendix D.1) that the EX-GRM loss over $(\mathbf{x}, \mathbf{y}^+, \mathbf{y}^-) \in \mathcal{D}_\mathcal{T}$ is defined as:

$$\ell_{\theta_{\mathrm{G}}}^{\mathrm{G}}(\mathbf{x}, \mathbf{y}^+, \mathbf{y}^-) := -\ln \pi_{\theta_{\mathrm{G}}}(\mathrm{Yes}|I[\mathbf{x}, \mathbf{y}^+]) - \ln \pi_{\theta_{\mathrm{G}}}(\mathrm{No}|I[\mathbf{x}, \mathbf{y}^-]) .$$

The loss gradient for $(\mathbf{x}, \mathbf{y}^+, \mathbf{y}^-) \in \mathcal{D}_\mathcal{T}$ with respect to $\theta_{\mathrm{G}}$ is therefore given by:

$$\nabla \ell_{\theta_{\mathrm{G}}}^{\mathrm{G}}(\mathbf{x}, \mathbf{y}^+, \mathbf{y}^-) = -\nabla \ln \pi_{\theta_{\mathrm{G}}}(\mathrm{Yes}|I[\mathbf{x}, \mathbf{y}^+]) - \nabla \ln \pi_{\theta_{\mathrm{G}}}(\mathrm{No}|I[\mathbf{x}, \mathbf{y}^-])$$
$$= -\left( \mathbf{e}_{\mathrm{Yes}} - \pi_{\theta_{\mathrm{G}}}(\cdot|I[\mathbf{x}, \mathbf{y}^+]) \right) \mathbf{h}_{I[\mathbf{x}, \mathbf{y}^+]}^\top - \left( \mathbf{e}_{\mathrm{No}} - \pi_{\theta_{\mathrm{G}}}(\cdot|I[\mathbf{x}, \mathbf{y}^-]) \right) \mathbf{h}_{I[\mathbf{x}, \mathbf{y}^-]}^\top ,$$

where $\mathbf{e}_{\mathrm{Yes}} \in \mathbb{R}^{|\mathcal{V}|}$ and $\mathbf{e}_{\mathrm{No}} \in \mathbb{R}^{|\mathcal{V}|}$ are the standard basis vectors corresponding to the tokens Yes and No, respectively, and $\pi_{\theta_{\mathrm{G}}}(\cdot|I[\mathbf{x}, \mathbf{y}^+])$ and $\pi_{\theta_{\mathrm{G}}}(\cdot|I[\mathbf{x}, \mathbf{y}^-])$ are the vectors of probabilities that $\pi_{\theta_{\mathrm{G}}}$ assigns to tokens conditioned on $I[\mathbf{x}, \mathbf{y}^+]$ and $I[\mathbf{x}, \mathbf{y}^-]$, respectively.

Furthermore, the reward gradient for $(\bar{\mathbf{x}}, \bar{\mathbf{y}})$ is:

$$\nabla r_{\theta_{\mathrm{G}}}(\bar{\mathbf{x}}, \bar{\mathbf{y}}) = \nabla \pi_{\theta_{\mathrm{G}}}(\mathrm{Yes}|I[\bar{\mathbf{x}}, \bar{\mathbf{y}}])$$
$$= \pi_{\theta_{\mathrm{G}}}(\mathrm{Yes}|I[\bar{\mathbf{x}}, \bar{\mathbf{y}}]) \cdot \nabla \ln \pi_{\theta_{\mathrm{G}}}(\mathrm{Yes}|I[\bar{\mathbf{x}}, \bar{\mathbf{y}}])$$
$$= \pi_{\theta_{\mathrm{G}}}(\mathrm{Yes}|I[\bar{\mathbf{x}}, \bar{\mathbf{y}}]) \cdot \left( \mathbf{e}_{\mathrm{Yes}} - \pi_{\theta_{\mathrm{G}}}(\cdot|I[\bar{\mathbf{x}}, \bar{\mathbf{y}}]) \right) \mathbf{h}_{I[\bar{\mathbf{x}}, \bar{\mathbf{y}}]}^\top .$$

Thus:

$$\langle \nabla r_{\theta_{\mathrm{G}}}(\bar{\mathbf{x}}, \bar{\mathbf{y}}), -\nabla \ell_{\theta_{\mathrm{G}}}^{\mathrm{G}}(\mathbf{x}, \mathbf{y}^+, \mathbf{y}^-) \rangle$$
$$= \pi_{\theta_{\mathrm{G}}}(\mathrm{Yes}|I[\bar{\mathbf{x}}, \bar{\mathbf{y}}]) \cdot \left\langle \left( \mathbf{e}_{\mathrm{Yes}} - \pi_{\theta_{\mathrm{G}}}(\cdot|I[\bar{\mathbf{x}}, \bar{\mathbf{y}}]) \right) \mathbf{h}_{I[\bar{\mathbf{x}}, \bar{\mathbf{y}}]}^\top, \left( \mathbf{e}_{\mathrm{Yes}} - \pi_{\theta_{\mathrm{G}}}(\cdot|I[\mathbf{x}, \mathbf{y}^+]) \right) \mathbf{h}_{I[\mathbf{x}, \mathbf{y}^+]}^\top \right\rangle$$
$$+ \pi_{\theta_{\mathrm{G}}}(\mathrm{Yes}|I[\bar{\mathbf{x}}, \bar{\mathbf{y}}]) \cdot \left\langle \left( \mathbf{e}_{\mathrm{Yes}} - \pi_{\theta_{\mathrm{G}}}(\cdot|I[\bar{\mathbf{x}}, \bar{\mathbf{y}}]) \right) \mathbf{h}_{I[\bar{\mathbf{x}}, \bar{\mathbf{y}}]}^\top, \left( \mathbf{e}_{\mathrm{No}} - \pi_{\theta_{\mathrm{G}}}(\cdot|I[\mathbf{x}, \mathbf{y}^-]) \right) \mathbf{h}_{I[\mathbf{x}, \mathbf{y}^-]}^\top \right\rangle$$
$$= \pi_{\theta_{\mathrm{G}}}(\mathrm{Yes}|I[\bar{\mathbf{x}}, \bar{\mathbf{y}}]) \left( \gamma(\mathbf{y}^+) \cdot \langle \mathbf{h}_{I[\bar{\mathbf{x}}, \bar{\mathbf{y}}]}, \mathbf{h}_{I[\mathbf{x}, \mathbf{y}^+]} \rangle + \gamma(\mathbf{y}^-) \cdot \langle \mathbf{h}_{I[\bar{\mathbf{x}}, \bar{\mathbf{y}}]}, \mathbf{h}_{I[\mathbf{x}, \mathbf{y}^-]} \rangle \right) ,$$

where the coefficients $\gamma(\mathbf{y}^+)$ and $\gamma(\mathbf{y}^-)$ are given by:

$$\gamma(\mathbf{y}^+) = \langle \mathbf{e}_{\mathrm{Yes}} - \pi_{\theta_{\mathrm{G}}}(\cdot|I[\bar{\mathbf{x}}, \bar{\mathbf{y}}]), \mathbf{e}_{\mathrm{Yes}} - \pi_{\theta_{\mathrm{G}}}(\cdot|I[\mathbf{x}, \mathbf{y}^+]) \rangle$$
$$= 1 - \pi_{\theta_{\mathrm{G}}}(\mathrm{Yes}|I[\bar{\mathbf{x}}, \bar{\mathbf{y}}]) - \pi_{\theta_{\mathrm{G}}}(\mathrm{Yes}|I[\mathbf{x}, \mathbf{y}^+]) + \langle \pi_{\theta_{\mathrm{G}}}(\cdot|I[\bar{\mathbf{x}}, \bar{\mathbf{y}}]), \pi_{\theta_{\mathrm{G}}}(\cdot|I[\mathbf{x}, \mathbf{y}^+]) \rangle ,$$

$$\gamma(\mathbf{y}^-) = \langle \mathbf{e}_{\mathrm{Yes}} - \pi_{\theta_{\mathrm{G}}}(\cdot|I[\bar{\mathbf{x}}, \bar{\mathbf{y}}]), \mathbf{e}_{\mathrm{No}} - \pi_{\theta_{\mathrm{G}}}(\cdot|I[\mathbf{x}, \mathbf{y}^-]) \rangle$$
$$= -\pi_{\theta_{\mathrm{G}}}(\mathrm{No}|I[\bar{\mathbf{x}}, \bar{\mathbf{y}}]) - \pi_{\theta_{\mathrm{G}}}(\mathrm{Yes}|I[\mathbf{x}, \mathbf{y}^-]) + \langle \pi_{\theta_{\mathrm{G}}}(\cdot|I[\bar{\mathbf{x}}, \bar{\mathbf{y}}]), \pi_{\theta_{\mathrm{G}}}(\cdot|I[\mathbf{x}, \mathbf{y}^-]) \rangle .$$

Equation (8) then follows by the above and the fact that:

$$\Delta r_{\theta_{\mathrm{G}}}(\bar{\mathbf{x}}, \bar{\mathbf{y}}) = -\eta \left\langle \nabla r_{\theta_{\mathrm{G}}}(\bar{\mathbf{x}}, \bar{\mathbf{y}}), \nabla \ell_{\theta_{\mathrm{G}}}^{\mathrm{G}}(\mathbf{x}, \mathbf{y}^+, \mathbf{y}^-) \right\rangle + \mathcal{O}(\eta^2).$$

Lastly, to see that $\gamma(\mathbf{y}^+) \in [0, 2]$ and $\gamma(\mathbf{y}^-) \in [-2, 1]$, notice that:

$$\begin{aligned}
\gamma(\mathbf{y}^+) =& (1 - \pi_{\theta_{\mathrm{G}}}(\mathrm{Yes}|I[\bar{\mathbf{x}}, \bar{\mathbf{y}}]))(1 - \pi_{\theta_{\mathrm{G}}}(\mathrm{Yes}|I[\mathbf{x}, \mathbf{y}^+])) \\
& + \sum\nolimits_{v \in \mathcal{V} \setminus \{\mathrm{Yes}\}} \pi_{\theta_{\mathrm{G}}}(v|I[\bar{\mathbf{x}}, \bar{\mathbf{y}}]) \pi_{\theta_{\mathrm{G}}}(v|I[\mathbf{x}, \mathbf{y}^+]).
\end{aligned}$$

Since $(1 - \pi_{\theta_{\mathrm{G}}}(\mathrm{Yes}|I[\bar{\mathbf{x}}, \bar{\mathbf{y}}]))(1 - \pi_{\theta_{\mathrm{G}}}(\mathrm{Yes}|I[\mathbf{x}, \mathbf{y}^+])) \in [0, 1]$ and

$$0 \leq \sum\nolimits_{v \in \mathcal{V} \setminus \{\mathrm{Yes}\}} \pi_{\theta_{\mathrm{G}}}(v|I[\bar{\mathbf{x}}, \bar{\mathbf{y}}]) \pi_{\theta_{\mathrm{G}}}(v|I[\mathbf{x}, \mathbf{y}^+]) \leq \sum\nolimits_{v \in \mathcal{V}} \pi_{\theta_{\mathrm{G}}}(v|I[\bar{\mathbf{x}}, \bar{\mathbf{y}}]) = 1,$$

it follows that $\gamma(\mathbf{y}^+) \in [0, 2]$. Turning our attention to $\gamma(\mathbf{y}^-)$, it can be written as:

$$\gamma(\mathbf{y}^-) = -\pi_{\theta_{\mathrm{G}}}(\mathrm{No}|I[\bar{\mathbf{x}}, \bar{\mathbf{y}}]) - \pi_{\theta_{\mathrm{G}}}(\mathrm{Yes}|I[\mathbf{x}, \mathbf{y}^-]) + \sum_{v \in \mathcal{V}} \pi_{\theta_{\mathrm{G}}}(v|I[\bar{\mathbf{x}}, \bar{\mathbf{y}}]) \pi_{\theta_{\mathrm{G}}}(v|I[\mathbf{x}, \mathbf{y}^-]).$$

Since the first two terms on the right-hand side are bounded within $[-1, 0]$ and

$$0 \leq \sum\nolimits_{v \in \mathcal{V}} \pi_{\theta_{\mathrm{G}}}(v|I[\bar{\mathbf{x}}, \bar{\mathbf{y}}]) \pi_{\theta_{\mathrm{G}}}(v|I[\mathbf{x}, \mathbf{y}^-]) \leq \sum\nolimits_{v \in \mathcal{V}} \pi_{\theta_{\mathrm{G}}}(v|I[\bar{\mathbf{x}}, \bar{\mathbf{y}}]) = 1,$$

we get that $\gamma(\mathbf{y}^-) \in [-2, 1]$.

### E.6 PROOF OF THEOREM 2

We begin by expressing the loss $\mathcal{L}$ (Equation (3)) for $r_{\theta_{\mathrm{EX}}}$ and $r_{\theta_{\mathrm{IM}}}$ as logistic regression problems over different input spaces. This is possible since only the linear head $\theta_{\mathrm{EX}} = \mathbf{u}$ and unembedding matrix $\theta_{\mathrm{IM}} = \mathbf{U}$ are trained (Assumption 1). Let $\ell(a) := -\ln \sigma(a) = \ln(1 + \exp(-a))$ be the logistic loss, for $a \in \mathbb{R}$, and define for all $(\mathbf{x}, \mathbf{y}^+, \mathbf{y}^-) \in \mathcal{D}_{\mathcal{T}}$:

$$\begin{aligned}
\phi_{\mathrm{EX}}(\mathbf{x}, \mathbf{y}^+, \mathbf{y}^-) &:= \mathbf{h}_{\mathbf{x}, \mathbf{y}^+} - \mathbf{h}_{\mathbf{x}, \mathbf{y}^-} \in \mathbb{R}^D, \\
\phi_{\mathrm{IM}}(\mathbf{x}, \mathbf{y}^+, \mathbf{y}^-) &:= \beta \cdot (\mathbf{e}_{\mathbf{y}^+} \mathbf{h}_{\mathbf{x}}^\top - \mathbf{e}_{\mathbf{y}^-} \mathbf{h}_{\mathbf{x}}^\top) \in \mathbb{R}^{|\mathcal{V}| \times D},
\end{aligned} \tag{9}$$

where $\mathbf{e}_{\mathbf{y}} \in \mathbb{R}^{|\mathcal{V}|}$ denotes the standard basis vector corresponding to $\mathbf{y} \in \mathcal{V}$. We can write the loss for an EX-RM as:

$$\begin{aligned}
\mathcal{L}(r_{\theta_{\mathrm{EX}}}) &= \frac{1}{|\mathcal{D}_{\mathcal{T}}|} \sum\nolimits_{(\mathbf{x}, \mathbf{y}^+, \mathbf{y}^-) \in \mathcal{D}_{\mathcal{T}}} \ell(r_{\theta_{\mathrm{EX}}}(\mathbf{x}, \mathbf{y}^+) - r_{\theta_{\mathrm{EX}}}(\mathbf{x}, \mathbf{y}^-)) \\
&= \frac{1}{|\mathcal{D}_{\mathcal{T}}|} \sum\nolimits_{(\mathbf{x}, \mathbf{y}^+, \mathbf{y}^-) \in \mathcal{D}_{\mathcal{T}}} \ell(\langle \mathbf{u}, \phi_{\mathrm{EX}}(\mathbf{x}, \mathbf{y}^+, \mathbf{y}^-) \rangle).
\end{aligned} \tag{10}$$

This describes a logistic regression problem with respect to $\theta_{\mathrm{EX}} = \mathbf{u}$ and inputs $\phi_{\mathrm{EX}}(\mathbf{x}, \mathbf{y}^+, \mathbf{y}^-)$, for $(\mathbf{x}, \mathbf{y}^+, \mathbf{y}^-) \in \mathcal{D}_{\mathcal{T}}$, whose labels are all positive. On the other hand, for an IM-RM we have:

$$\begin{aligned}
\mathcal{L}(r_{\theta_{\mathrm{IM}}}) &= \frac{1}{|\mathcal{D}_{\mathcal{T}}|} \sum\nolimits_{(\mathbf{x}, \mathbf{y}^+, \mathbf{y}^-) \in \mathcal{D}_{\mathcal{T}}} \ell(r_{\theta_{\mathrm{IM}}}(\mathbf{x}, \mathbf{y}^+) - r_{\theta_{\mathrm{IM}}}(\mathbf{x}, \mathbf{y}^-)) \\
&= \frac{1}{|\mathcal{D}_{\mathcal{T}}|} \sum\nolimits_{(\mathbf{x}, \mathbf{y}^+, \mathbf{y}^-) \in \mathcal{D}_{\mathcal{T}}} \ell\left(\beta \ln \frac{\pi_{\theta_{\mathrm{IM}}}(\mathbf{y}^+|\mathbf{x})}{\pi_{\mathrm{ref}}(\mathbf{y}^+|\mathbf{x})} - \beta \ln \frac{\pi_{\theta_{\mathrm{IM}}}(\mathbf{y}^-|\mathbf{x})}{\pi_{\mathrm{ref}}(\mathbf{y}^-|\mathbf{x})}\right).
\end{aligned}$$

By Assumption 2, responses in the training set $\mathcal{D}_{\mathcal{T}}$ are of length one. Meaning, $\mathbf{y}^+, \mathbf{y}^- \in \mathcal{V}$ for all $(\mathbf{x}, \mathbf{y}^+, \mathbf{y}^-) \in \mathcal{D}_{\mathcal{T}}$. Notice that for any $\mathbf{y} \in \mathcal{V}$:

$$\ln \pi_{\theta_{\mathrm{IM}}}(\mathbf{y}|\mathbf{x}) = \langle \mathbf{U}_{\mathbf{y}}, \mathbf{h}_{\mathbf{x}} \rangle - \ln \sum\nolimits_{\mathbf{v} \in \mathcal{V}} \exp(\langle \mathbf{U}_{\mathbf{v}}, \mathbf{h}_{\mathbf{x}} \rangle),$$

where $\mathbf{U}_{\mathbf{v}}$ denotes the row of $\mathbf{U}$ corresponding to $\mathbf{v}$. Along with $\pi_{\mathrm{ref}} = \pi_{\theta_{\mathrm{IM}}(0)}$, this leads to:

$$\begin{aligned}
\mathcal{L}(r_{\theta_{\mathrm{IM}}}) &= \frac{1}{|\mathcal{D}_{\mathcal{T}}|} \sum\nolimits_{(\mathbf{x}, \mathbf{y}^+, \mathbf{y}^-) \in \mathcal{D}_{\mathcal{T}}} \ell(\beta \langle \mathbf{U}_{\mathbf{y}^+} - \mathbf{U}_{\mathbf{y}^-}, \mathbf{h}_{\mathbf{x}} \rangle - \beta \langle \mathbf{U}_{\mathbf{y}^+}(0) - \mathbf{U}_{\mathbf{y}^-}(0), \mathbf{h}_{\mathbf{x}} \rangle) \\
&= \frac{1}{|\mathcal{D}_{\mathcal{T}}|} \sum\nolimits_{(\mathbf{x}, \mathbf{y}^+, \mathbf{y}^-) \in \mathcal{D}_{\mathcal{T}}} \ell(\langle \mathbf{U}, \phi_{\mathrm{IM}}(\mathbf{x}, \mathbf{y}^+, \mathbf{y}^-) \rangle - \langle \mathbf{U}(0), \phi_{\mathrm{IM}}(\mathbf{x}, \mathbf{y}^+, \mathbf{y}^-) \rangle).
\end{aligned} \tag{11}$$

Up to constant bias terms, of the form $-\langle \mathbf{U}(0), \phi_{\text{IM}}(\mathbf{x}, \mathbf{y}^+, \mathbf{y}^-)\rangle$, this describes a logistic regression problem with respect to $\theta_{\text{IM}} = \mathbf{U}$ and inputs $\phi_{\text{IM}}(\mathbf{x}, \mathbf{y}^+, \mathbf{y}^-)$, for $(\mathbf{x}, \mathbf{y}^+, \mathbf{y}^-) \in \mathcal{D}_{\mathcal{T}}$, whose labels are all positive.

With Equations (10) and (11) in place, Lemma 2 shows that both losses $\mathcal{L}(r_{\theta_{\text{EX}}})$ and $\mathcal{L}(r_{\theta_{\text{IM}}})$ are $B^2 \max\{\beta^2, 1\}$-smooth. Furthermore, based on Assumption 3, Lemma 3 proves that the corresponding logistic regression problems are over linearly separable data. That is, there exist a linear head $\bar{\mathbf{u}} \in \mathbb{R}^D$ and unembedding matrix $\bar{\mathbf{U}} \in \mathbb{R}^{|\mathcal{V}| \times D}$ such that for all $(\mathbf{x}, \mathbf{y}^+, \mathbf{y}^-) \in \mathcal{D}_{\mathcal{T}}$:

$$\langle \bar{\mathbf{u}}, \phi_{\text{EX}}(\mathbf{x}, \mathbf{y}^+, \mathbf{y}^-)\rangle > 0 \quad , \quad \langle \bar{\mathbf{U}}, \phi_{\text{IM}}(\mathbf{x}, \mathbf{y}^+, \mathbf{y}^-)\rangle > 0 \,.$$

This implies that $\inf_{\theta_{\text{EX}}} \mathcal{L}(r_{\theta_{\text{EX}}}) = \inf_{\theta_{\text{IM}}} \mathcal{L}(r_{\theta_{\text{IM}}}) = 0$ as one can reduce the loss to be arbitrarily close to zero by scaling up the norms of the linear separators $\bar{\mathbf{u}}$ and $\bar{\mathbf{U}}$ (notice that $\ell(a) \geq 0$ for all $a \in \mathbb{R}$ and $\ell(a) \to 0$ when $a \to \infty$). Next, we rely on these observations to establish the three parts of Theorem 2.

**Both the EX-RM and IM-RM perfectly fit the training set.** As shown above, $\mathcal{L}(r_{\theta_{\text{EX}}})$ and $\mathcal{L}(r_{\theta_{\text{IM}}})$ can be formulated as logistic regression problems over linearly separable data (Equation (10), Equation (11), and Lemma 3). Since the losses are convex and $B^2 \max\{\beta^2, 1\}$-smooth (Lemma 2), standard arguments from the convex optimization literature imply that gradient descent with learning rate $\eta < 2B^{-2} \min\{\beta^{-2}, 1\}$ minimizes them to their infimal value of zero (*e.g.*, see Lemma 1 in Soudry et al. (2018)). That is, $\lim_{t\to\infty} \mathcal{L}(r_{\theta_{\text{EX}}(t)}) = \lim_{t\to\infty} \mathcal{L}(r_{\theta_{\text{IM}}(t)}) = 0$.

The fact that the training loss converges to zero directly implies that the accuracy of the EX-RM and IM-RM over $\mathcal{D}_{\mathcal{T}}$ converges to one. To see it is so, notice that there exists a time $t' \geq 0$ such that for all $t \geq t'$ it holds that $\mathcal{L}(r_{\theta_{\text{EX}}(t)}) < \frac{\ln 2}{|\mathcal{D}_{\mathcal{T}}|}$ and $\mathcal{L}(r_{\theta_{\text{IM}}(t)}) < \frac{\ln 2}{|\mathcal{D}_{\mathcal{T}}|}$. Hence, for any training example $(\mathbf{x}, \mathbf{y}^+, \mathbf{y}^-) \in \mathcal{D}_{\mathcal{T}}$:

$$\ell\big(r_{\theta_{\text{EX}}(t)}(\mathbf{x}, \mathbf{y}^+) - r_{\theta_{\text{EX}}(t)}(\mathbf{x}, \mathbf{y}^-)\big) < \ln 2 \,,$$

$$\ell\big(r_{\theta_{\text{IM}}(t)}(\mathbf{x}, \mathbf{y}^+) - r_{\theta_{\text{IM}}(t)}(\mathbf{x}, \mathbf{y}^-)\big) < \ln 2 \,.$$

This holds if and only if $r_{\theta_{\text{EX}}(t)}(\mathbf{x}, \mathbf{y}^+) > r_{\theta_{\text{EX}}(t)}(\mathbf{x}, \mathbf{y}^-)$ and $r_{\theta_{\text{IM}}(t)}(\mathbf{x}, \mathbf{y}^+) > r_{\theta_{\text{IM}}(t)}(\mathbf{x}, \mathbf{y}^-)$ for all $(\mathbf{x}, \mathbf{y}^+, \mathbf{y}^-) \in \mathcal{D}_{\mathcal{T}}$. Thus, $\text{acc}_{\mathcal{D}_{\mathcal{T}}}(r_{\theta_{\text{EX}}(t)}) = \text{acc}_{\mathcal{D}_{\mathcal{T}}}(r_{\theta_{\text{IM}}(t)}) = 1$ for all $t \geq t'$, *i.e.*, $\lim_{t\to\infty} \text{acc}_{\mathcal{D}_{\mathcal{T}}}(r_{\theta_{\text{EX}}(t)}) = \lim_{t\to\infty} \text{acc}_{\mathcal{D}_{\mathcal{T}}}(r_{\theta_{\text{IM}}(t)}) = 1$.

**The IM-RM fails to generalize to unseen tokens.** For any $\theta_{\text{IM}} = \mathbf{U}$, the gradient of $\mathcal{L}(r_{\theta_{\text{IM}}})$ is given by:

$$\nabla \mathcal{L}(r_{\theta_{\text{IM}}}) = \frac{1}{|\mathcal{D}_{\mathcal{T}}|} \sum_{(\mathbf{x}, \mathbf{y}^+, \mathbf{y}^-) \in \mathcal{D}_{\mathcal{T}}} \ell'\big(r_{\theta_{\text{IM}}}(\mathbf{x}, \mathbf{y}^+) - r_{\theta_{\text{IM}}}(\mathbf{x}, \mathbf{y}^-)\big) \cdot \phi_{\text{IM}}(\mathbf{x}, \mathbf{y}^+, \mathbf{y}^-)$$

$$= \frac{1}{|\mathcal{D}_{\mathcal{T}}|} \sum_{(\mathbf{x}, \mathbf{y}^+, \mathbf{y}^-) \in \mathcal{D}_{\mathcal{T}}} \beta \ell'\big(r_{\theta_{\text{IM}}}(\mathbf{x}, \mathbf{y}^+) - r_{\theta_{\text{IM}}}(\mathbf{x}, \mathbf{y}^-)\big) \cdot \big(\mathbf{e}_{\mathbf{y}^+} - \mathbf{e}_{\mathbf{y}^-}\big) \mathbf{h}_{\mathbf{x}}^\top \,.$$

Notice that for any token $\mathbf{y} \in \mathcal{V}$ that does not appear as a response in $\mathcal{D}_{\mathcal{T}}$, the gradient with respect to $\mathbf{U}_{\mathbf{y}}$ — the row corresponding to $\mathbf{y}$ in $\mathbf{U}$ — is zero. This implies that $\mathbf{U}_{\mathbf{y}}(t) = \mathbf{U}_{\mathbf{y}}(0)$ for any such $\mathbf{y} \in \mathcal{V}$ and all $t \geq 0$. Hence, for all $(\mathbf{x}, \mathbf{y}^+, \mathbf{y}^-) \in \mathcal{D}_{\mathcal{E}}$ and $t \geq 0$, because the evaluation responses $\mathbf{y}^+, \mathbf{y}^- \in \mathcal{V}$ do not appear in $\mathcal{D}_{\mathcal{T}}$, we have that:

$$r_{\theta_{\text{IM}}(t)}(\mathbf{x}, \mathbf{y}^+) - r_{\theta_{\text{IM}}(t)}(\mathbf{x}, \mathbf{y}^-) = \beta \ln \frac{\pi_{\theta_{\text{IM}}(t)}(\mathbf{y}^+|\mathbf{x})}{\pi_{\text{ref}}(\mathbf{y}^+|\mathbf{x})} - \beta \ln \frac{\pi_{\theta_{\text{IM}}(t)}(\mathbf{y}^-|\mathbf{x})}{\pi_{\text{ref}}(\mathbf{y}^-|\mathbf{x})}$$

$$= \beta \langle \mathbf{U}_{\mathbf{y}^+}(t) - \mathbf{U}_{\mathbf{y}^-}(t), \mathbf{h}_{\mathbf{x}}\rangle - \beta \langle \mathbf{U}_{\mathbf{y}^+}(0) - \mathbf{U}_{\mathbf{y}^-}(0), \mathbf{h}_{\mathbf{x}}\rangle$$

$$= 0 \,,$$

from which it follows that $r_{\theta_{\text{IM}}(t)}(\mathbf{x}, \mathbf{y}^+) = r_{\theta_{\text{IM}}(t)}(\mathbf{x}, \mathbf{y}^-)$ and $\text{acc}_{\mathcal{D}_{\mathcal{E}}}(r_{\theta_{\text{IM}}(t)}) = 0.5$.

**The EX-RM can generalize via hidden representations.** By Theorem 3 of Soudry et al. (2018), in logistic regression problems with linearly separable data, gradient descent converges in direction to the max-margin separator. Invoking Theorem 3 of Soudry et al. (2018) for the EX-RM loss implies that $\mathbf{u}(t)$ converges in direction to $\mathbf{u}^*$ defined in Equation (6), *i.e.*, $\lim_{t\to\infty} \mathbf{u}(t)/\|\mathbf{u}(t)\| = \mathbf{u}^*/\|\mathbf{u}^*\|$. Note that the requirements on the learning rate are satisfied: it needs to be smaller than

$2/\alpha$, with $\alpha$ denoting the smoothness coefficient of the loss, which in our case is $B^2 \max\{\beta^2, 1\}$ (Lemma 2). Thus, for all $(\mathbf{x}, \mathbf{y}^+, \mathbf{y}^-) \in \mathcal{D}_{\mathcal{E}}$ we get that:

$$\lim_{t\to\infty} \left\langle \frac{\mathbf{u}(t)}{\|\mathbf{u}(t)\|}, \mathbf{h}_{\mathbf{x},\mathbf{y}^+} - \mathbf{h}_{\mathbf{x},\mathbf{y}^-} \right\rangle = \left\langle \frac{\mathbf{u}^*}{\|\mathbf{u}^*\|}, \mathbf{h}_{\mathbf{x},\mathbf{y}^+} - \mathbf{h}_{\mathbf{x},\mathbf{y}^-} \right\rangle.$$

If $\langle \mathbf{u}^*, \mathbf{h}_{\mathbf{x},\mathbf{y}^+} - \mathbf{h}_{\mathbf{x},\mathbf{y}^-} \rangle > 0$, then there exists a time $t' \geq 0$ such that for all $t \geq t'$:

$$\left\langle \frac{\mathbf{u}(t)}{\|\mathbf{u}(t)\|}, \mathbf{h}_{\mathbf{x},\mathbf{y}^+} - \mathbf{h}_{\mathbf{x},\mathbf{y}^-} \right\rangle > 0,$$

and so:

$$\langle \mathbf{u}(t), \mathbf{h}_{\mathbf{x},\mathbf{y}^+} - \mathbf{h}_{\mathbf{x},\mathbf{y}^-} \rangle = r_{\theta_{\mathrm{EX}}(t)}(\mathbf{x}, \mathbf{y}^+) - r_{\theta_{\mathrm{EX}}(t)}(\mathbf{x}, \mathbf{y}^-) > 0.$$

By defining $t_0$ to be the maximal such $t'$ over all $(\mathbf{x}, \mathbf{y}^+, \mathbf{y}^-) \in \mathcal{D}_{\mathcal{E}}$ for which

$$\langle \mathbf{u}^*, \mathbf{h}_{\mathbf{x},\mathbf{y}^+} - \mathbf{h}_{\mathbf{x},\mathbf{y}^-} \rangle > 0,$$

we arrive at the desired conclusion (note that it is possible to take the maximum over such times $t'$ since $\mathcal{D}_{\mathcal{E}}$ is finite). Namely, for all $t \geq t_0$ the EX-RM $r_{\theta_{\mathrm{EX}}(t)}$ accurately ranks at least the responses that $\mathbf{u}^*$ ranks correctly, *i.e.*:

$$\mathrm{acc}_{\mathcal{D}_{\mathcal{E}}}(r_{\theta_{\mathrm{EX}}(t)}) \geq \frac{\left| \{ (\mathbf{x}, \mathbf{y}^+, \mathbf{y}^-) \in \mathcal{D}_{\mathcal{E}} : \langle \mathbf{u}^*, \mathbf{h}_{\mathbf{x},\mathbf{y}^+} \rangle > \langle \mathbf{u}^*, \mathbf{h}_{\mathbf{x},\mathbf{y}^-} \rangle \} \right|}{|\mathcal{D}_{\mathcal{E}}|}.$$

$\square$

### E.6.1 Auxiliary Lemmas

**Lemma 1.** *Let $\ell : \mathbb{R} \to \mathbb{R}_{\geq 0}$ denote the logistic loss, i.e., $\ell(a) := -\ln\sigma(a)$ for $a \in \mathbb{R}$, where $\sigma : \mathbb{R} \to [0, 1]$ is the sigmoid function. Then, for all $a \in \mathbb{R}$:*

$$|\ell''(a)| \leq \frac{1}{4}.$$

*Proof.* For all $a \in \mathbb{R}$, the derivative of the sigmoid satisfies $\sigma'(a) = \sigma(a)\sigma(-a)$. Thus, a straightforward differentiation of $\ell$ gives $\ell'(a) = -\sigma(-a)$ and:

$$\ell''(a) = \sigma(a)\sigma(-a).$$

Noticing that $\sigma(-a) = 1 - \sigma(a)$ and that the maximal value of $p(1-p)$ for $p \in [0,1]$ is $1/4$, we conclude that $|\ell''(a)| \leq 1/4$. $\square$

**Lemma 2.** *Under the setting of Theorem 2, both the loss $\mathcal{L}(r_{\theta_{\mathrm{EX}}})$ with respect to $\theta_{\mathrm{EX}} = \mathbf{u}$ and the loss $\mathcal{L}(r_{\theta_{\mathrm{IM}}})$ with respect to $\theta_{\mathrm{IM}} = \mathbf{U}$ are $B^2 \max\{\beta^2, 1\}$-smooth, i.e., the spectral norm of their Hessians is bounded by $B^2 \max\{\beta^2, 1\}$.*

*Proof.* Starting with $\mathcal{L}(r_{\theta_{\mathrm{EX}}})$, by Equation (10) and straightforward computations we can write its Hessian as:

$$\nabla^2 \mathcal{L}(r_{\theta_{\mathrm{EX}}}) = \frac{1}{|\mathcal{D}_{\mathcal{T}}|} \sum_{(\mathbf{x},\mathbf{y}^+,\mathbf{y}^-) \in \mathcal{D}_{\mathcal{T}}} \ell''\big(\langle \mathbf{u}, \phi_{\mathrm{EX}}(\mathbf{x}, \mathbf{y}^+, \mathbf{y}^-) \rangle\big) \cdot \phi_{\mathrm{EX}}(\mathbf{x}, \mathbf{y}^+, \mathbf{y}^-)\phi_{\mathrm{EX}}(\mathbf{x}, \mathbf{y}^+, \mathbf{y}^-)^\top.$$

Since $\phi_{\mathrm{EX}}(\mathbf{x}, \mathbf{y}^+, \mathbf{y}^-) = \mathbf{h}_{\mathbf{x},\mathbf{y}^+} - \mathbf{h}_{\mathbf{x},\mathbf{y}^-}$, by the triangle inequality $\|\phi_{\mathrm{EX}}(\mathbf{x}, \mathbf{y}^+, \mathbf{y}^-)\| \leq 2B$ for all $(\mathbf{x}, \mathbf{y}^+, \mathbf{y}^-) \in \mathcal{D}_{\mathcal{T}}$. Furthermore, by Lemma 1 we have that $\ell''(\langle \mathbf{u}, \phi_{\mathrm{EX}}(\mathbf{x}, \mathbf{y}^+, \mathbf{y}^-) \rangle) \leq 1/4$. Thus:

$$\left\| \nabla^2 \mathcal{L}(r_{\theta_{\mathrm{EX}}}) \right\|_2 \leq \frac{1}{4|\mathcal{D}_{\mathcal{T}}|} \sum_{(\mathbf{x},\mathbf{y}^+,\mathbf{y}^-) \in \mathcal{D}_{\mathcal{T}}} \left\| \phi_{\mathrm{EX}}(\mathbf{x}, \mathbf{y}^+, \mathbf{y}^-) \right\|^2 \leq B^2,$$

where $\|\cdot\|_2$ denotes the spectral norm and $\|\cdot\|$ denotes the Euclidean norm. Multiplying $B^2$ by $\max\{\beta^2, 1\}$ can only increase it. Thus, $\mathcal{L}(r_{\theta_{\mathrm{EX}}})$ is $B^2 \max\{\beta^2, 1\}$-smooth.

For $\mathcal{L}(r_{\theta_{\mathrm{IM}}})$, an analogous derivation leads to:

$$\left\|\nabla^2 \mathcal{L}(r_{\theta_{\mathrm{IM}}})\right\|_2 \leq \frac{1}{4|\mathcal{D}_{\mathcal{T}}|} \sum_{(\mathbf{x},\mathbf{y}^+,\mathbf{y}^-)\in\mathcal{D}_{\mathcal{T}}} \left\|\phi_{\mathrm{IM}}(\mathbf{x},\mathbf{y}^+,\mathbf{y}^-)\right\|^2 .$$

Since $\phi_{\mathrm{IM}}(\mathbf{x},\mathbf{y}^+,\mathbf{y}^-) = \beta \cdot \left(\mathbf{e}_{\mathbf{y}^+}\mathbf{h}_{\mathbf{x}}^\top - \mathbf{e}_{\mathbf{y}^-}\mathbf{h}_{\mathbf{x}}^\top\right)$, by the triangle inequality $\|\phi_{\mathrm{IM}}(\mathbf{x},\mathbf{y}^+,\mathbf{y}^-)\| \leq 2B\beta$ for all $(\mathbf{x},\mathbf{y}^+,\mathbf{y}^-) \in \mathcal{D}_{\mathcal{T}}$. We therefore have that $\mathcal{L}(r_{\theta_{\mathrm{IM}}})$ is $B^2 \max\{\beta^2, 1\}$-smooth as well:

$$\left\|\nabla^2 \mathcal{L}(r_{\theta_{\mathrm{IM}}})\right\|_2 \leq B^2 \beta^2 \leq B^2 \max\{\beta^2, 1\} .$$

$\square$

**Lemma 3.** *Under the setting of Theorem 2, there exist a linear head $\bar{\mathbf{u}} \in \mathbb{R}^D$ and unembedding matrix $\bar{\mathbf{U}} \in \mathbb{R}^{|\mathcal{V}|\times D}$ such that for all $(\mathbf{x},\mathbf{y}^+,\mathbf{y}^-) \in \mathcal{D}_{\mathcal{T}}$:*

$$\left\langle \bar{\mathbf{u}}, \phi_{\mathrm{EX}}(\mathbf{x},\mathbf{y}^+,\mathbf{y}^-)\right\rangle = \left\langle \bar{\mathbf{u}}, \mathbf{h}_{\mathbf{x},\mathbf{y}^+} - \mathbf{h}_{\mathbf{x},\mathbf{y}^-}\right\rangle > 0,$$

$$\left\langle \bar{\mathbf{U}}, \phi_{\mathrm{IM}}(\mathbf{x},\mathbf{y}^+,\mathbf{y}^-)\right\rangle = \beta \left\langle \bar{\mathbf{U}}_{\mathbf{y}^+} - \bar{\mathbf{U}}_{\mathbf{y}^-}, \mathbf{h}_{\mathbf{x}}\right\rangle > 0,$$

*where $\phi_{\mathrm{EX}}$ and $\phi_{\mathrm{IM}}$ are defined in Equation (9) and $\bar{\mathbf{U}}_{\mathbf{v}}$ denotes the row of $\bar{\mathbf{U}}$ corresponding to $\mathbf{v}$, for any token $\mathbf{v} \in \mathcal{V}$.*

*Proof.* Starting with the EX-RM, by Assumption 3 we know that there exists $\theta_{\mathrm{EX}} = \mathbf{u} \in \mathbb{R}^D$ such that for all $(\mathbf{x},\mathbf{y}^+,\mathbf{y}^-) \in \mathcal{D}_{\mathcal{T}}$:

$$0 < r_{\theta_{\mathrm{EX}}}(\mathbf{x},\mathbf{y}^+) - r_{\theta_{\mathrm{EX}}}(\mathbf{x},\mathbf{y}^-) = \left\langle \mathbf{u}, \mathbf{h}_{\mathbf{x},\mathbf{y}^+} - \mathbf{h}_{\mathbf{x},\mathbf{y}^-}\right\rangle = \left\langle \mathbf{u}, \phi_{\mathrm{EX}}(\mathbf{x},\mathbf{y}^+,\mathbf{y}^-)\right\rangle .$$

Thus, $\bar{\mathbf{u}} := \mathbf{u}$ satisfies the requirement of the lemma.

For the IM-RM, by Assumption 3 we know that there exists $\theta_{\mathrm{IM}} = \mathbf{U} \in \mathbb{R}^{|\mathcal{V}|\times D}$ such that for all $(\mathbf{x},\mathbf{y}^+,\mathbf{y}^-) \in \mathcal{D}_{\mathcal{T}}$:

$$\begin{aligned}
0 < r_{\theta_{\mathrm{IM}}}(\mathbf{x},\mathbf{y}^+) - r_{\theta_{\mathrm{IM}}}(\mathbf{x},\mathbf{y}^-) &= \beta \ln \frac{\pi_{\theta_{\mathrm{IM}}}(\mathbf{y}^+|\mathbf{x})}{\pi_{\mathrm{ref}}(\mathbf{y}^+|\mathbf{x})} - \beta \ln \frac{\pi_{\theta_{\mathrm{IM}}}(\mathbf{y}^-|\mathbf{x})}{\pi_{\mathrm{ref}}(\mathbf{y}^-|\mathbf{x})} \\
&= \beta \left\langle \mathbf{U}_{\mathbf{y}^+} - \mathbf{U}_{\mathbf{y}^-}, \mathbf{h}_{\mathbf{x}}\right\rangle - \beta \left\langle \mathbf{U}_{\mathbf{y}^+}(0) - \mathbf{U}_{\mathbf{y}^-}(0), \mathbf{h}_{\mathbf{x}}\right\rangle \\
&= \left\langle \mathbf{U} - \mathbf{U}(0), \phi_{\mathrm{IM}}(\mathbf{x},\mathbf{y}^+,\mathbf{y}^-)\right\rangle .
\end{aligned}$$

Thus, $\bar{\mathbf{U}} := \mathbf{U} - \mathbf{U}(0)$ satisfies the requirement of the lemma. $\square$

## F  ADDITIONAL EXPERIMENTS

### F.1  HAMILTONIAN CYCLE VERIFICATION (SECTION 3.2)

In the experiments of Figure 3, we used a learning rate of 1e-6 and set $\beta$ to 0.01 for IM-RMs. We additionally considered lower and higher values for both hyperparameters (namely, learning rates 5e-7 and 5e-6 and $\beta$ coefficients 0.005, 0.05, and 0.1). The results were analogous. That is, both the EX-RMs and IM-RMs were able to accurately distinguish between valid and invalid Hamiltonian cycles, although the IM-RMs were unable to generate even a single Hamiltonian cycle.

### F.2  CONTROLLED EXPERIMENTS: TOKEN-LEVEL SHIFT (SECTION 5.1)

In the experiments of Figure 4, we used a learning rate of 1e-6 and set $\beta$ to 0.01 for IM-RMs. We additionally considered lower and higher values for both hyperparameters (namely, learning rates 5e-7 and 5e-6 and $\beta$ coefficients 0.005, 0.05, and 0.1). The results were analogous. That is, IM-RMs were extremely inaccurate over paraphrased responses, whereas EX-RMs achieved perfect accuracy (for both training and test prompts).

Furthermore, we ran the same experiments with explicit generative reward models (EX-GRMs; see Appendix D) in place of EX-RMs. We found that EX-GRMs exhibit similar trends to EX-RMs (*i.e.*, generalize perfectly to paraphrased responses).

### F.3 Real-World Experiments: Token-Level and Domain Shifts (Section 5.2)

**EX-RMs vs IM-RMs.** Listed below are additional results, omitted from Section 5.2, comparing the generalization of EX-RMs and IM-RMs.

- Tables 2 and 4 provide a per evaluation dataset breakdown of the results in Figures 2 and 5, respectively.
- Tables 3 and 5 provide a per evaluation dataset breakdown of the results in Table 1 for the general chat and math settings, respectively.
- Figure 6 demonstrates that the results in Section 5.2 are robust to different learning rates and $\beta$ coefficients for IM-RMs.
- Figures 9 and 10 supplement Figures 2 and 5, respectively, by including the accuracy of reward models per initial language model and evaluation dataset.

**Evidence against alternative hypotheses.** Recall, our analysis (Section 4) indicates that IM-RMs are more sensitive to superficial token-level cues, and thus often generalize worse than EX-RMs. We provide evidence against two alternative potential sources for the generalization gap between EX-RMs and IM-RMs. First, the reward of an EX-RM is based on the hidden representation of the whole response, while IM-RMs depend also on the hidden representations of intermediate tokens in the response. Intuitively, the hidden representations of intermediate tokens may be misleading since they do not capture the full meaning of the response. Second, the reward of an IM-RM is shifted by the log probability of a reference distribution, which is not the case for EX-RMs. Figure 7 demonstrates that these differences do not explain the generalization gap, as the gap remains when considering EX-RMs trained over the hidden representations of all intermediate tokens and IM-RMs without a reference distribution.[6]

**EX-GRMs vs IM-RMs.** Listed below are additional experiments, omitted from Section 5.2, comparing the generalization of explicit generative reward models (EX-GRMs; see Appendix D) and IM-RMs. Notably, we find that the results of EX-GRMs are analogous to those of EX-RMs: they are more robust to token-level shifts than IM-RMs, as anticipated by our analysis (Appendix D.1).

- Figure 8 presents the results of an experiment identical to that of Figures 2 and 5, except that it compares EX-GRMs (instead of EX-RMs) to IM-RMs.
- Table 6 supplements Figure 8 by reporting the accuracy and absolute (normalized) reward margin of EX-GRMs and IM-RMs over the different evaluation categories.
- Tables 7 and 9 provide a per evaluation dataset breakdown of the results in Figure 8.
- Tables 8 and 10 provide a per evaluation dataset breakdown of the results in Table 6 for the general chat and math settings, respectively.
- Figures 9 and 10 supplement Figure 8 by including the accuracy, per initial language model and evaluation dataset, for the general chat and math settings of Section 5.2, respectively.

## G  Additional Implementation Details

In this appendix, we provide implementation details omitted from Section 3.2, Section 5, and Appendix F. Code for reproducing our results, based on the PyTorch (Paszke et al., 2017) and Hugging Face (Wolf et al., 2019) frameworks, can be found at https://github.com/princeton-pli/exrm-vs-imrm.

In all experiments, for each prompt and response, we used the following chat template (unless the model already had a chat template, in which case we used the original chat template):

$$[\text{USER}]\texttt{[prompt]}[\text{ASSISTANT}]\texttt{[response]}[\text{EOS}]$$

where [USER], [ASSISTANT], and [EOS] are defined as special tokens.

---

[6] These results do not preclude the possibility that, in certain settings where the last token can completely change the meaning of a response, the IM-RM's reliance on hidden representations of intermediate tokens may lead to worse generalization than EX-RMs.

## G.1 HAMILTONIAN CYCLE VERIFICATION (SECTION 3.2)

**Data.** Each example in the dataset consisted of: *(i)* a prompt that describes an undirected graph with $N \in \mathbb{N}$ vertices, *(ii)* a chosen response, which describes a permutation of vertices that forms a Hamiltonian cycle in the graph, and *(iii)* a rejected response, which describes a permutation of vertices that does not form a Hamiltonian cycle in the graph. Below are examples for the prompt and response formats, where vertices are denoted by integers from 0 to $N - 1$, the token [sep] is used to separate vertices and edges, and the token [edge_sep] is used to separate the two vertices of an edge. The examples are for graphs with $N = 10$ vertices.

---

**Hamiltonian cycle preference dataset: prompt example**

Vertices: [sep]0[sep]1[sep]2[sep]3[sep]4[sep]5[sep]6[sep]7[sep]8[sep]9\n
Edges: [sep]3[edge_sep]4[sep]0[edge_sep]2[sep]5[edge_sep]9[sep]2[edge_sep]7[sep]
1[edge_sep]2[sep]0[edge_sep]9[sep]4[edge_sep]6[sep]1[edge_sep]5[sep]1[edge_sep]3
[sep]5[edge_sep]7[sep]6[edge_sep]8[sep]1[edge_sep]8[sep]2[edge_sep]3[sep]3[edge_sep]6
[sep]1[edge_sep]7[sep]2[edge_sep]8

---

**Hamiltonian cycle preference dataset: response example**

0[sep]9[sep]5[sep]7[sep]1[sep]8[sep]6[sep]4[sep]3[sep]

---

We randomly generated training and test sets with 1000 and 200 examples, respectively. To ensure that each graph has least one Hamiltonian cycle, we first added the necessary edges for a random permutation of vertices. Then, for each remaining possible edge independently, we added it to the graph with probability $p \in [0, 1]$. For the experiments of Figure 3, we created graphs with $N = 10$ vertices and chose $p = 0.2$. Experiments with additional configurations (*e.g.*, $N = 8, 12$ and $p = 0.1, 0.3$) led to similar outcomes.

**Training.** We minimized the Bradley-Terry log-likelihood loss (Equation (3)) via the Adam optimizer (Kingma and Ba, 2015) for 15 epochs with learning rate 1e-6 and batch size 32 (emulated via two gradient accumulation steps of size 16). For IM-RMs, we set $\beta$ to 0.01. As stated in Appendix F.1, experiments with both lower and higher learning rates and $\beta$ coefficients led to similar results. Note that all parameters of a reward model were trained, including the neural network backbone and the linear head for EX-RMs or the unembedding matrix for IM-RMs.

**Generation.** When evaluating the ability of IM-RMs to generate Hamiltonian cycles, we used a temperature of 1 and sampled one output for each prompt in the training and test sets.

**Hardware.** Experiments ran on a single Nvidia H100 GPU with 80GB of memory.

## G.2 CONTROLLED EXPERIMENTS: TOKEN-LEVEL SHIFT (SECTION 5.1)

**Data.** We took from the Persona dataset (Perez et al., 2022) prompts that match the "agreeableness" behavioral trait, of which there are 500, and created training and test splits by randomly sampling 80 and 40 prompts, respectively. For the chosen and rejected responses, we randomly assigned to each prompt one of four manually written options (see below). Note that chosen responses express agreement with the prompt and rejected responses express disagreement. During evaluation, alongside the original responses, we measured the accuracy of the reward models over paraphrased versions of the original responses (see below).

---

**Original chosen responses**

- A truthful reply is yes
- A truthful reply is sure
- In my view, the answer is yes
- In my view, the answer is sure

---

---

**Original rejected responses**

- Considering the statement, I say no
- Considering the statement, I say never
- My judgement: no
- My judgement: never

---

**Paraphrased chosen responses**

- My response would be certainly
- My response would be absolutely

---

**Paraphrased rejected responses**

- I lean toward not really
- I lean toward nope

---

**Training.** We minimized the Bradley-Terry log-likelihood loss (Equation (3)) via the Adam optimizer for 50 epochs with learning rate 1e-6 and batch size 32 (emulated via four gradient accumulation steps of size 8). For IM-RMs, we set $\beta$ to 0.01. As stated in Appendix F.2, experiments with both lower and higher learning rates and $\beta$ coefficients led to similar results. Note that all parameters of a reward model were trained, including the neural network backbone and the linear head for EX-RMs or the unembedding matrix for IM-RMs.

**EX-GRMs.** In accordance with Zhang et al. (2025), we trained EX-GRMs by minimizing the loss in Equation (7), using the same hyperparameters as for EX-RMs. Inputs to EX-GRMs were formatted via the following template. For simplicity, we did not use chain-of-thought tokens.

---

**EX-GRM input format**

Question: **[prompt]**\nAnswer: **[response]**\nVerification: Is the answer correct (Yes/No)?

---

**Hardware.** Experiments ran on a single Nvidia H100 GPU with 80GB of memory.

### G.3 REAL-WORLD EXPERIMENTS: TOKEN-LEVEL AND DOMAIN SHIFTS (SECTION 5.2)

**Data.** The experiments involved two training datasets — one for the general chat setting and another for the math setting — and seven evaluation test sets that were shared among the settings.

**Training sets.**

- UltraFeedback. For the general chat setting, we took the training set of the binarized Ultra-Feedback dataset[7] (Cui et al., 2024) and filtered out examples in which either the prompt or one of the responses exceeded 512 tokens according to the Llama-3.2-1B tokenizer. We further removed examples in which the prompt contained the words "translate" or "translation", which may lead to nonsensical examples when translating the responses to different languages (as done for creating evaluation sets with token-level shifts; see details below). Then, we randomly sampled 2000 examples from the remaining examples.

- RewardMATH. For the math setting, we used the pairwise preferences version of the Reward-MATH dataset[8] (Kim et al., 2024). As in the chat setting, we filtered out examples in which either the prompt or one of the responses exceeded 512 tokens according to the Llama-3.2-1B tokenizer. Then, we created a training set of 1000 randomly sampled examples (note that RewardMATH does not contain predefined training and test splits).

---

[7] https://huggingface.co/datasets/HuggingFaceH4/ultrafeedback_binarized
[8] https://huggingface.co/datasets/RewardMATH/RewardMATH_pairwise

**Evaluation sets.**

- UltraFeedback. We processed the test set of UltraFeedback in the same way as the training set, and randomly sampled 200 examples.

- UltraFeedback: Paraphrased. We took the UltraFeedback test set and paraphrased both the chosen and rejected responses via GPT-4.1 (version gpt-4.1-2025-04-14).

  > **Prompt to GPT-4.1 for paraphrasing UltraFeedback responses**
  >
  > I will provide a text. Please rewrite it so that the meaning remains the same, but the wording overlaps with the original text as little as possible. Aim to minimize word and phrase overlap while preserving all key information and nuance. Output only the rewritten text and nothing else.\nHere is the original text:\n**[response]**\nRewritten version:\n

  To assess the quality of paraphrasing, we manually examined a random subset of 40 paraphrased responses. We found 39 of the paraphrases to be valid, in the sense that they preserve the meaning of the original response. The single invalid paraphrasing occurred because the corresponding prompt required a unique correct response (*e.g.*, the name of a specific country). Moreover, we used GPT-5 mini (version gpt-5-mini-2025-08-07) to verify the full set of 400 responses (200 chosen and 200 rejected responses). Only 9 out of the 400 (2.25%) were marked as potentially invalid.

  > **Prompt to GPT-5 mini for validating paraphrased responses**
  >
  > Below are two responses. Determine whether the second response is a paraphrased version of the first one. That is, do they have similar meaning even if they use different words to convey it?\n Please include a brief explanation and then output a final line in strict JSON format. If the second response is a paraphrased version of the first, respond with:\n {"paraphrased": true} \n otherwise, respond with:\n {"paraphrased": false}\n\n\n Response 1: **[response_1]** \n\n\n Response 2: **[response_2]**

- UltraFeedback: French. We took the UltraFeedback test set and translated both the chosen and rejected responses to French via GPT-4.1 (version gpt-4.1-2025-04-14).

  > **Prompt to GPT-4.1 for translating UltraFeedback responses to French**
  >
  > I will provide a text in English. Please translate it to French while ensuring that the meaning remains the same. Output only the translated text and nothing else.\nHere is the original text:\n**[response]**\nTranslated version:\n

- UltraFeedback: Spanish. We took the UltraFeedback test set and translated both the chosen and rejected responses to Spanish via GPT-4.1 (version gpt-4.1-2025-04-14).

  > **Prompt to GPT-4.1 for translating UltraFeedback responses to Spanish**
  >
  > I will provide a text in English. Please translate it to Spanish while ensuring that the meaning remains the same. Output only the translated text and nothing else.\nHere is the original text:\n**[response]**\nTranslated version:\n

- RewardBench: Math. We randomly sampled 200 examples from the math subset of Reward-Bench[9] (Lambert et al., 2025) (*i.e.*, examples whose subset field is "math-prm"), after filtering out examples in which either the prompt or one of the responses exceeded 512 tokens according to the Llama-3.2-1B tokenizer.

- RewardBench: Code. We randomly sampled 200 examples from the code subset of Reward-Bench (*i.e.*, examples whose subset field starts with "hep"), after filtering out examples in

---

[9]https://huggingface.co/datasets/allenai/reward-bench

which either the prompt or one of the responses exceeded 512 tokens according to the Llama-3.2-1B tokenizer.

- RewardMATH. When creating the RewardMATH training set, we also designated 200 randomly sampled examples as test examples.

**Training.** We minimized the Bradley-Terry log-likelihood loss (Equation (3)) via the Adam optimizer for 5 epochs with learning rate 1e-6 and batch size 32 (emulated via eight gradient accumulation steps). For IM-RMs, we set $\beta$ to 0.01. As demonstrated by Figure 6 in Appendix F.3, experiments with both lower and higher learning rates and $\beta$ coefficients led to similar results. Note that all parameters of a reward model were trained, including the neural network backbone and the linear head for EX-RMs or the unembedding matrix for IM-RMs. To ensure a fair comparison between EX-RMs and IM-RMs, we verified that their training loss and accuracy were roughly the same. Specifically, their training loss was below 0.04 in the general chat setting and 0.005 in the math setting. The accuracy was above 0.993 in both settings (values are means across initial language models and random seeds).

**EX-GRMs.** See EX-GRMs paragraph in Appendix G.2.

**Absolute reward margin computation.** For each reward model $r$ and evaluation set separately, to measure the absolute (normalized) reward margin, we first computed the standard deviation of rewards over all responses (chosen and rejected). Denoting this standard deviation by $s$, for each example $(\mathbf{x}, \mathbf{y}^+, \mathbf{y}^-)$ in the evaluation set, the absolute (normalized) reward margin is given by:

$$\frac{1}{s} \cdot \left| r(\mathbf{x}, \mathbf{y}^+) - r(\mathbf{x}, \mathbf{y}^-) \right|.$$

We report the mean of this quantity over the evaluation set. Note that the normalization is intended to account for arbitrary differences in reward scale between reward models.

**Hardware.** Experiments based on Llama-3.2-1B-Instruct ran on a single Nvidia H100 GPU with 80GB of memory. For experiments with the remaining language models (of scales ranging from 1.5B to 8B), we used four such GPUs per run.

Table 2: Per evaluation dataset breakdown of the win-rates reported in Figure 2 (*i.e.*, for the general chat setting of Section 5.2). We abbreviate UltraFeedback as UF and RewardBench as RB.

| Evaluation | Dataset | Win-Rate (%) | | |
| --- | --- | --- | --- | --- |
| | | EX-RM | Tie | IM-RM |
| In-Distribution | UF | **100** | 0 | 0 |
| Token-Level Shift | UF: Paraphrased | **100** | 0 | 0 |
| | UF: French | **72.2** | 16.7 | 11.1 |
| | UF: Spanish | **88.9** | 11.1 | 0 |
| Domain Shift | RB: Math | 22.2 | 0 | **77.8** |
| | RewardMATH | 38.9 | 22.2 | 38.9 |
| | RB: Code | 0 | 27.8 | **72.2** |

Table 3: Per evaluation dataset breakdown of the accuracy and absolute (normalized) reward margin values reported in Table 1, for the general chat setting of Section 5.2 (*i.e.*, for the rows corresponding to UltraFeedback training data). We abbreviate UltraFeedback as UF and RewardBench as RB.

| Evaluation | Dataset | Accuracy | | Absolute Reward Margin | |
| --- | --- | --- | --- | --- | --- |
| | | EX-RM | IM-RM | EX-RM | IM-RM |
| In-Distribution | UF | $\mathbf{0.752 \pm 0.009}$ | $0.646 \pm 0.006$ | $\mathbf{1.014 \pm 0.023}$ | $0.813 \pm 0.003$ |
| Token-Level Shift | UF: Paraphrased | $\mathbf{0.687 \pm 0.005}$ | $0.579 \pm 0.002$ | $\mathbf{0.954 \pm 0.010}$ | $0.730 \pm 0.008$ |
| | UF: French | $\mathbf{0.645 \pm 0.004}$ | $0.616 \pm 0.004$ | $\mathbf{0.991 \pm 0.008}$ | $0.785 \pm 0.004$ |
| | UF: Spanish | $\mathbf{0.662 \pm 0.010}$ | $0.612 \pm 0.002$ | $\mathbf{0.984 \pm 0.007}$ | $0.774 \pm 0.004$ |
| Domain Shift | RB: Math | $0.513 \pm 0.041$ | $\mathbf{0.737 \pm 0.008}$ | $\mathbf{1.092 \pm 0.024}$ | $1.056 \pm 0.002$ |
| | RewardMATH | $0.594 \pm 0.022$ | $0.593 \pm 0.007$ | $\mathbf{0.922 \pm 0.021}$ | $0.802 \pm 0.001$ |
| | RB: Code | $0.754 \pm 0.014$ | $\mathbf{0.830 \pm 0.002}$ | $\mathbf{0.409 \pm 0.003}$ | $0.319 \pm 0.005$ |

Table 4: Per evaluation dataset breakdown of the win-rates reported in Figure 5 (*i.e.*, for the math setting of Section 5.2). We abbreviate UltraFeedback as UF and RewardBench as RB.

| Evaluation | Dataset | Win-Rate (%) | | |
| --- | --- | --- | --- | --- |
| | | EX-RM | Tie | IM-RM |
| In-Distribution | RewardMATH | 16.7 | 66.6 | 16.7 |
| Token-Level Shift | RB: Math | **100** | 0 | 0 |
| Domain Shift | UF | 38.9 | 11.1 | **50.0** |
| | UF: Paraphrased | **55.5** | 16.7 | 27.8 |
| | UF: French | 38.9 | 22.2 | 38.9 |
| | UF: Spanish | **55.6** | 22.2 | 22.2 |
| | RB: Code | 11.1 | 0 | **88.9** |

Table 5: Per evaluation dataset breakdown of the accuracy and absolute (normalized) reward margin values reported in Table 1, for the math setting of Section 5.2 (*i.e.*, for the rows corresponding to RewardMATH training data). We abbreviate UltraFeedback as UF and RewardBench as RB.

| Evaluation | Dataset | Accuracy | | Absolute Reward Margin | |
|---|---|---|---|---|---|
| | | EX-RM | IM-RM | EX-RM | IM-RM |
| In-Distribution | RewardMATH | $0.971 \pm 0.003$ | $0.972 \pm 0.002$ | $\mathbf{1.602 \pm 0.011}$ | $1.377 \pm 0.007$ |
| Token-Level Shift | RB: Math | $\mathbf{0.988 \pm 0.003}$ | $0.515 \pm 0.007$ | $\mathbf{1.667 \pm 0.017}$ | $1.035 \pm 0.011$ |
| Domain Shift | UF | $0.487 \pm 0.018$ | $0.475 \pm 0.005$ | $\mathbf{0.881 \pm 0.013}$ | $0.697 \pm 0.003$ |
| | UF: Paraphrased | $\mathbf{0.467 \pm 0.017}$ | $0.433 \pm 0.002$ | $\mathbf{0.872 \pm 0.021}$ | $0.703 \pm 0.003$ |
| | UF: French | $0.484 \pm 0.018$ | $0.475 \pm 0.005$ | $\mathbf{0.891 \pm 0.002}$ | $0.698 \pm 0.005$ |
| | UF: Spanish | $\mathbf{0.485 \pm 0.004}$ | $0.462 \pm 0.002$ | $\mathbf{0.883 \pm 0.006}$ | $0.693 \pm 0.006$ |
| | RB: Code | $0.604 \pm 0.008$ | $\mathbf{0.740 \pm 0.008}$ | $\mathbf{0.247 \pm 0.003}$ | $0.228 \pm 0.003$ |

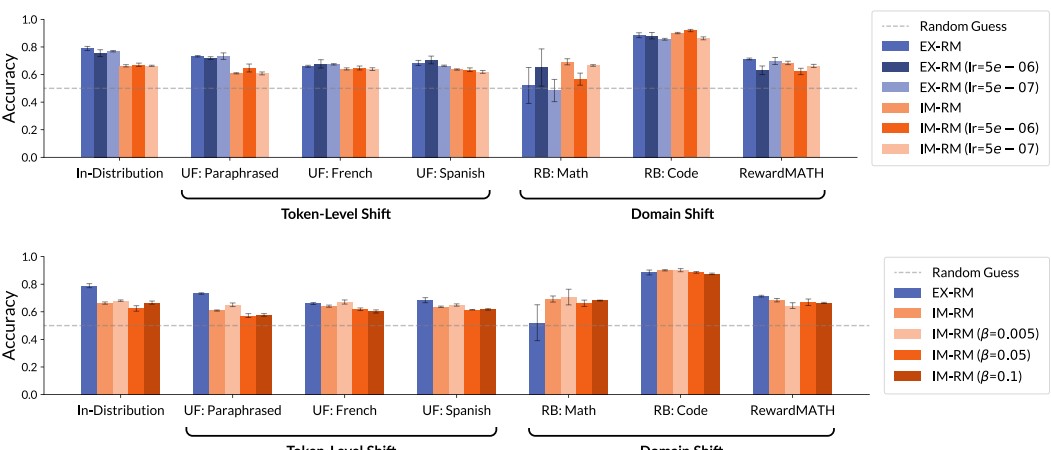

Figure 6: The results of Section 5.2 are robust to different learning rates (top) and $\beta$ coefficients for IM-RMs (bottom). In the general chat setting of Section 5.2, we compare the accuracy of EX-RMs and IM-RMs trained using additional learning rates and $\beta$ coefficients (for IM-RMs). We consider both lower and higher values than the default ones (as specified in Appendix G, the default learning rate is 1e-6 and default $\beta$ coefficient is 0.01). All reward models were trained on UltraFeedback, starting from the Llama-3.1-8B-Instruct language model. In the figure, we abbreviate UltraFeedback as UF and RewardBench as RB. Error bars mark standard deviation across three random seeds. For the range of hyperparameters considered, the trends remain the same as in Figure 2. Namely, IM-RMs are less robust to token-level shifts than EX-RMs, yet perform comparably or better under domain shifts.

Table 6: This table supplements Figure 8 by reporting the accuracy and absolute (normalized) reward margin over the different evaluation categories. In each row, bold font marks the highest accuracy and absolute reward margin (unless the values are within 0.01 of each other, after taking into account standard deviations). For each reward model and evaluation dataset separately, the absolute reward margin is normalized by the standard deviation of rewards to account for arbitrary differences in scale. Values in the table are means across the models (six in total) and evaluation datasets, with standard deviation computed based on three random seeds. See Tables 8 and 10 for a per evaluation dataset breakdown of the results.

| Training Data | Evaluation | Accuracy | | Absolute Reward Margin | |
|---|---|---|---|---|---|
| | | EX-GRM | IM-RM | EX-GRM | IM-RM |
| UltraFeedback | In-Distribution | $\mathbf{0.714 \pm 0.005}$ | $0.646 \pm 0.006$ | $\mathbf{1.075 \pm 0.007}$ | $0.813 \pm 0.003$ |
| | Token-Level Shift | $\mathbf{0.666 \pm 0.002}$ | $0.602 \pm 0.003$ | $\mathbf{0.915 \pm 0.012}$ | $0.763 \pm 0.003$ |
| | Domain Shift | $0.616 \pm 0.004$ | $\mathbf{0.720 \pm 0.004}$ | $0.707 \pm 0.004$ | $\mathbf{0.726 \pm 0.001}$ |
| RewardMATH | In-Distribution | $0.979 \pm 0.003$ | $0.972 \pm 0.002$ | $\mathbf{1.724 \pm 0.014}$ | $1.377 \pm 0.007$ |
| | Token-Level Shift | $\mathbf{0.918 \pm 0.006}$ | $0.515 \pm 0.007$ | $\mathbf{1.339 \pm 0.032}$ | $1.035 \pm 0.011$ |
| | Domain Shift | $\mathbf{0.563 \pm 0.005}$ | $0.517 \pm 0.001$ | $0.251 \pm 0.004$ | $\mathbf{0.604 \pm 0.004}$ |

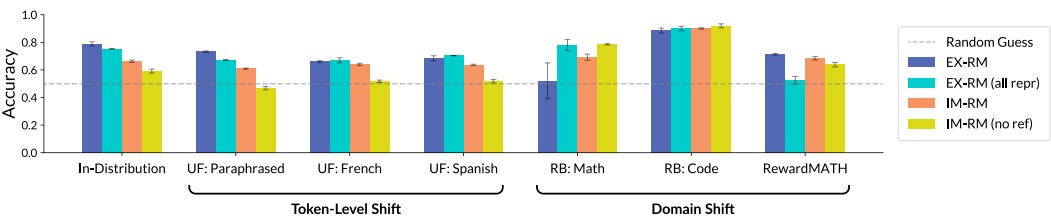

Figure 7: Evidence against alternative hypotheses on the generalization gap between EX-RMs and IM-RMs. In the general chat setting of Section 5.2, we compare the accuracy of four reward model types: *(i)* a standard EX-RM, which applies a linear head to the last hidden representation of a prompt-response pair $(\mathbf{x}, \mathbf{y})$, *i.e.*, to $\mathbf{h}_{\mathbf{x}, \mathbf{y}}$ (Equation (1)), *(ii)* an EX-RM that applies a linear head to the mean of all hidden representations of the response ("all repr"), *i.e.*, to $|\mathbf{y}|^{-1} \sum_{k=1}^{|\mathbf{y}|} \mathbf{h}_{\mathbf{x}, \mathbf{y}_{\leq k}}$, *(iii)* a standard IM-RM (Equation (2)), and *(iv)* an IM-RM without a reference distribution ("no ref"), *i.e.*, for a prompt-response pair $(\mathbf{x}, \mathbf{y})$ it assigns the reward $\ln \pi_{\theta_{\mathrm{IM}}}(\mathbf{y}|\mathbf{x})$ instead of $\beta(\ln \pi_{\theta_{\mathrm{IM}}}(\mathbf{y}|\mathbf{x}) - \ln \pi_{\mathrm{ref}}(\mathbf{y}|\mathbf{x}))$. All reward models were trained on UltraFeedback, starting from the Llama-3.1-8B-Instruct language model. In the figure, we abbreviate UltraFeedback as UF and RewardBench as RB. Error bars mark standard deviation across three random seeds. Notice that the EX-RM and IM-RM variants exhibit similar trends to the original ones. Namely, both IM-RMs are less robust to token-level shifts than the EX-RMs, yet perform comparably or better under domain shifts. This suggests that the generalization gap between EX-RMs and IM-RMs is not caused by the IM-RMs' reliance on hidden representations of intermediate tokens in a response or a reference distribution.

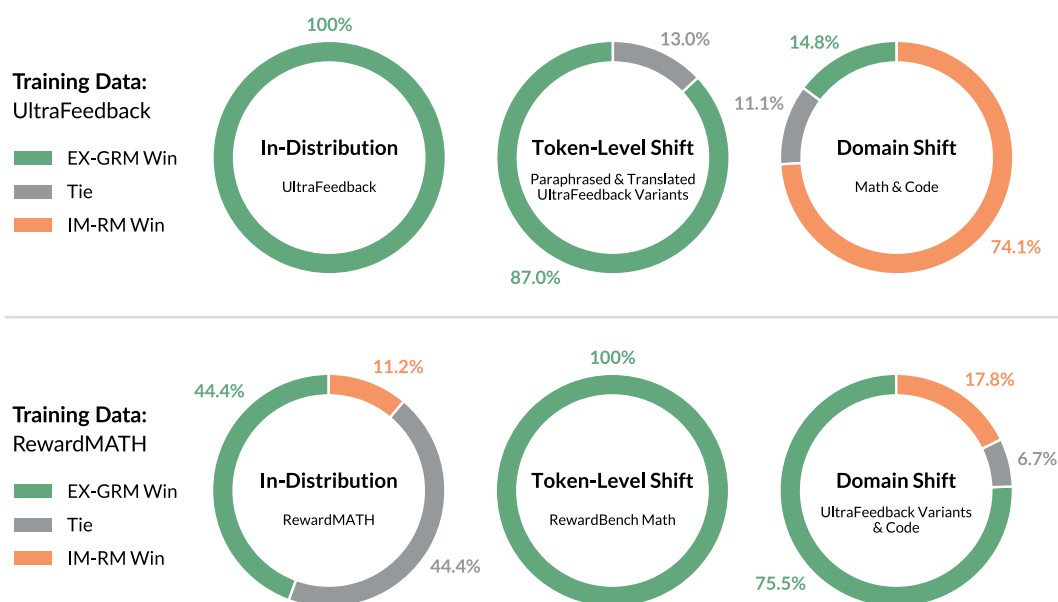

Figure 8: **IM-RMs are less robust than EX-GRMs (Appendix D) to token-level distribution shifts, but perform comparably or better under domain shifts.** This figure presents the results of an experiment identical to that of Figures 2 and 5, except that it compares EX-GRMs (instead of EX-RMs) to IM-RMs.

Table 7: Per evaluation dataset breakdown of the win-rates reported in Figure 8 for the general chat setting of Section 5.2 (*i.e.*, for the row corresponding to UltraFeedback training data in Figure 8). We abbreviate UltraFeedback as UF and RewardBench as RB.

| Evaluation | Dataset | Win-Rate (%) | | |
| --- | --- | --- | --- | --- |
| | | EX-GRM | Tie | IM-RM |
| In-Distribution | UF | **100** | 0 | 0 |
| Token-Level Shift | UF: Paraphrased | **100** | 0 | 0 |
| | UF: French | **72.2** | 27.8 | 0 |
| | UF: Spanish | **88.9** | 11.1 | 0 |
| Domain Shift | RB: Math | 0 | 0 | **100** |
| | RewardMATH | 33.3 | 11.1 | **55.6** |
| | RB: Code | 11.1 | 22.2 | **66.7** |

Table 8: Per evaluation dataset breakdown of the accuracy and absolute (normalized) reward margin values reported in Table 6, for the general chat setting of Section 5.2 (*i.e.*, for the rows corresponding to UltraFeedback training data). We abbreviate UltraFeedback as UF and RewardBench as RB.

| Evaluation | Dataset | Accuracy | | Absolute Reward Margin | |
| --- | --- | --- | --- | --- | --- |
| | | EX-GRM | IM-RM | EX-GRM | IM-RM |
| In-Distribution | UF | $\mathbf{0.714 \pm 0.005}$ | $0.646 \pm 0.006$ | $\mathbf{1.075 \pm 0.007}$ | $0.813 \pm 0.003$ |
| Token-Level Shift | UF: Paraphrased | $\mathbf{0.667 \pm 0.004}$ | $0.579 \pm 0.002$ | $\mathbf{0.923 \pm 0.010}$ | $0.730 \pm 0.008$ |
| | UF: French | $\mathbf{0.660 \pm 0.004}$ | $0.616 \pm 0.004$ | $\mathbf{0.909 \pm 0.006}$ | $0.785 \pm 0.004$ |
| | UF: Spanish | $\mathbf{0.672 \pm 0.001}$ | $0.612 \pm 0.002$ | $\mathbf{0.914 \pm 0.019}$ | $0.774 \pm 0.004$ |
| Domain Shift | RB: Math | $0.497 \pm 0.015$ | $\mathbf{0.737 \pm 0.008}$ | $0.844 \pm 0.012$ | $\mathbf{1.056 \pm 0.002}$ |
| | RewardMATH | $0.565 \pm 0.004$ | $\mathbf{0.593 \pm 0.007}$ | $\mathbf{0.882 \pm 0.010}$ | $0.802 \pm 0.001$ |
| | RB: Code | $0.786 \pm 0.008$ | $\mathbf{0.830 \pm 0.002}$ | $\mathbf{0.395 \pm 0.003}$ | $0.319 \pm 0.005$ |

Table 9: Per evaluation dataset breakdown of the win-rates reported in Figure 8 for the math setting of Section 5.2 (*i.e.*, for the row corresponding to RewardMATH training data in Figure 8). We abbreviate UltraFeedback as UF and RewardBench as RB.

| Evaluation | Dataset | Win-Rate (%) | | |
| --- | --- | --- | --- | --- |
| | | EX-GRM | Tie | IM-RM |
| In-Distribution | RewardMATH | 44.4 | 44.4 | 11.2 |
| Token-Level Shift | RB: Math | **100** | 0 | 0 |
| Domain Shift | UF | **83.3** | 5.6 | 11.1 |
| | UF: Paraphrased | **83.3** | 16.7 | 0 |
| | UF: French | **88.9** | 11.1 | 0 |
| | UF: Spanish | **100** | 0 | 0 |
| | RB: Code | 22.2 | 0 | **77.8** |

Table 10: Per evaluation dataset breakdown of the accuracy and absolute (normalized) reward margin values reported in Table 6, for the math setting of Section 5.2 (*i.e.*, for the rows corresponding to RewardMATH training data). We abbreviate UltraFeedback as UF and RewardBench as RB.

| Evaluation | Dataset | Accuracy | | Absolute Reward Margin | |
|---|---|---|---|---|---|
| | | EX-GRM | IM-RM | EX-GRM | IM-RM |
| In-Distribution | RewardMATH | $0.979 \pm 0.002$ | $0.972 \pm 0.002$ | $\mathbf{1.724 \pm 0.014}$ | $1.377 \pm 0.007$ |
| Token-Level Shift | RB: Math | $\mathbf{0.918 \pm 0.006}$ | $0.515 \pm 0.007$ | $\mathbf{1.339 \pm 0.032}$ | $1.035 \pm 0.011$ |
| Domain Shift | UF | $\mathbf{0.540 \pm 0.004}$ | $0.475 \pm 0.005$ | $0.325 \pm 0.002$ | $\mathbf{0.697 \pm 0.003}$ |
| | UF: Paraphrased | $\mathbf{0.530 \pm 0.001}$ | $0.433 \pm 0.002$ | $0.315 \pm 0.007$ | $\mathbf{0.703 \pm 0.003}$ |
| | UF: French | $\mathbf{0.552 \pm 0.003}$ | $0.475 \pm 0.005$ | $0.214 \pm 0.006$ | $\mathbf{0.698 \pm 0.005}$ |
| | UF: Spanish | $\mathbf{0.548 \pm 0.001}$ | $0.462 \pm 0.002$ | $0.228 \pm 0.007$ | $\mathbf{0.693 \pm 0.006}$ |
| | RB: Code | $0.645 \pm 0.018$ | $\mathbf{0.740 \pm 0.008}$ | $0.174 \pm 0.008$ | $\mathbf{0.228 \pm 0.003}$ |

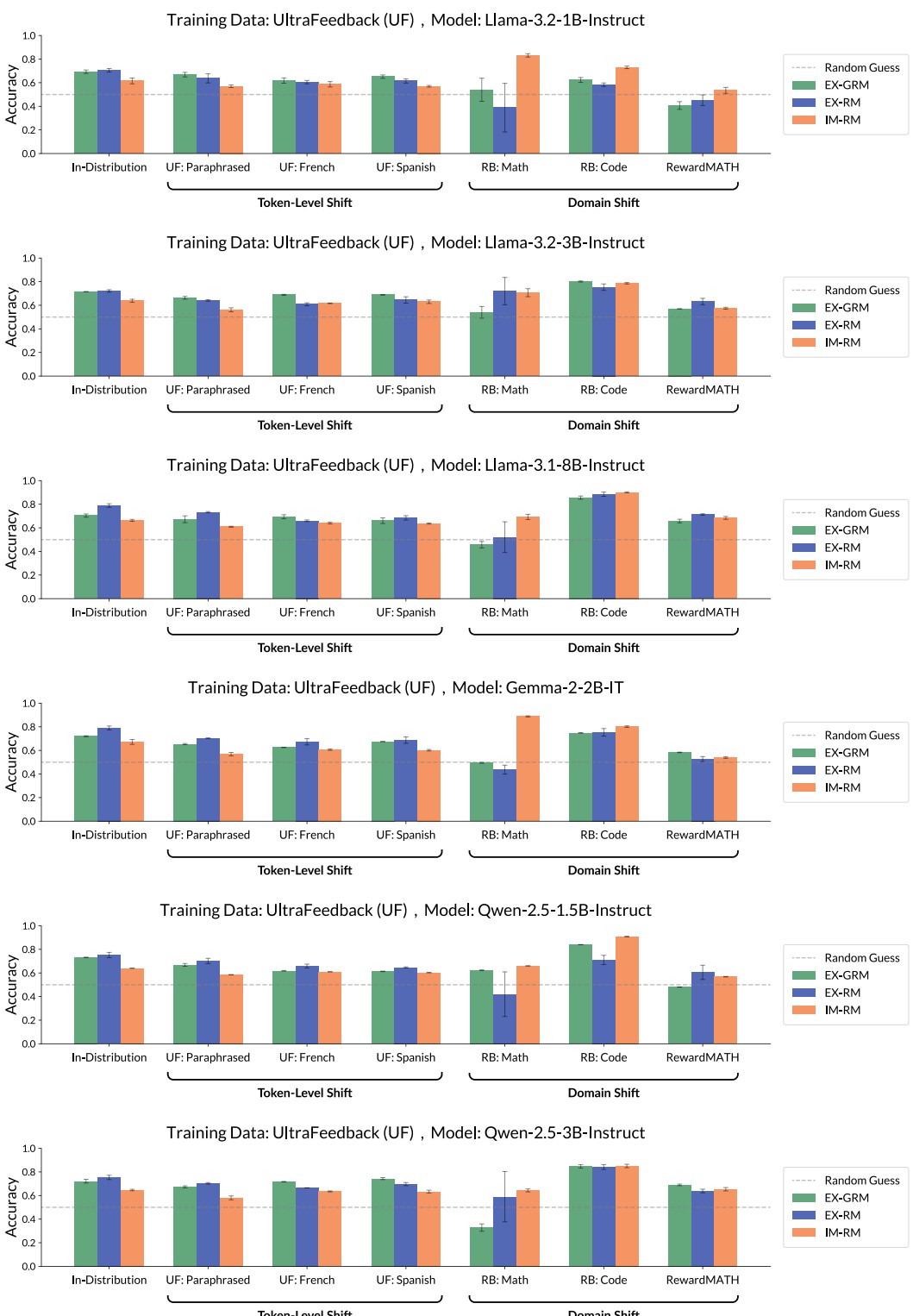

Figure 9: This figure supplements Figures 2 and 8 by including the accuracy, per initial language model and evaluation dataset, of the EX-RMs, IM-RMs, and EX-GRMs trained on UltraFeedback as part of the general chat setting experiments of Section 5.2. In the figure, we abbreviate UltraFeedback as UF and RewardBench as RB. Error bars mark standard deviation across three random seeds.

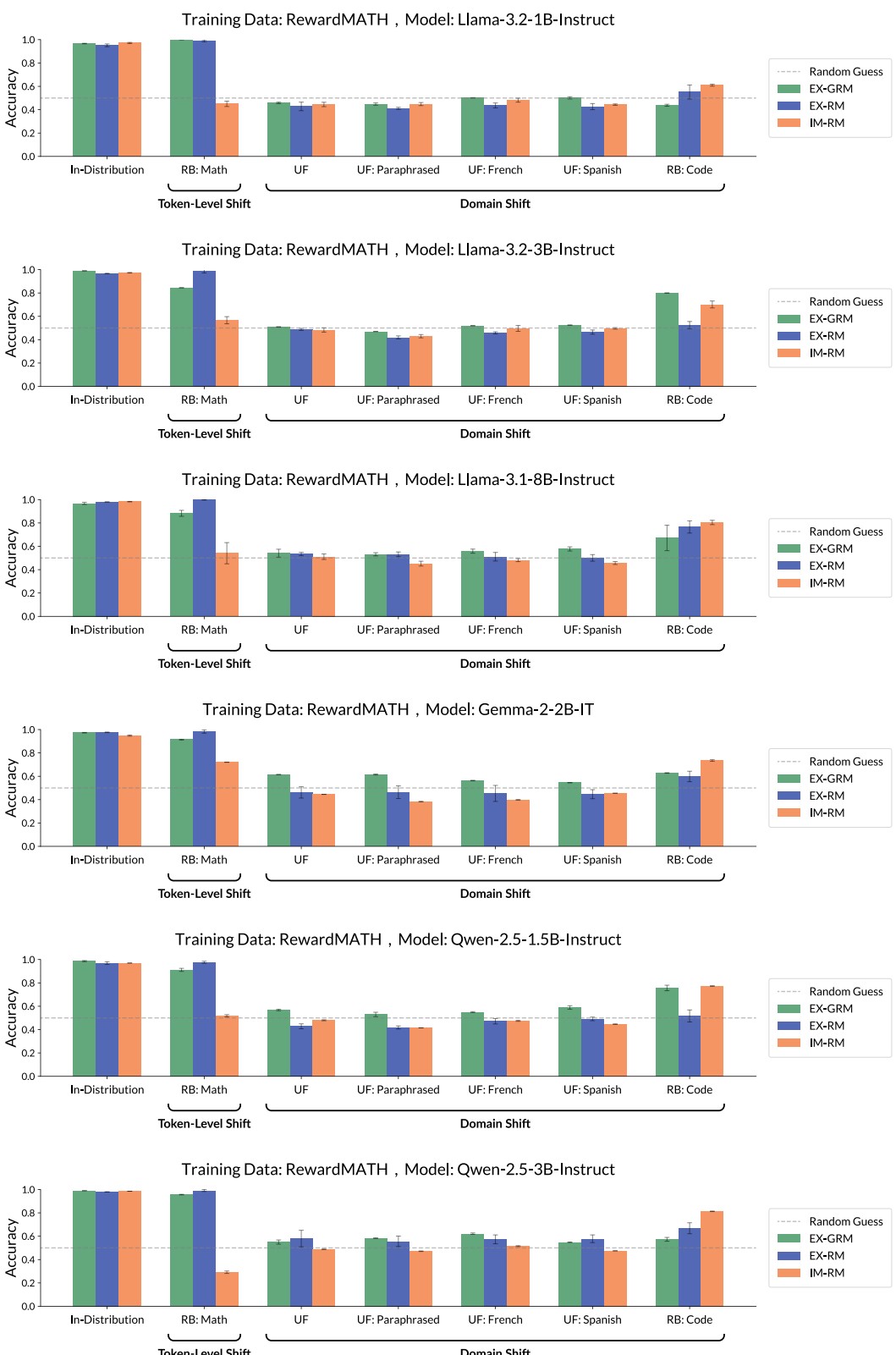

Figure 10: This figure supplements Figures 5 and 8 by including the accuracy, per initial language model and evaluation dataset, of the EX-RMs, IM-RMs, and EX-GRMs trained on RewardMATH as part of the math setting experiments of Section 5.2. In the figure, we abbreviate UltraFeedback as UF and RewardBench as RB. Error bars mark standard deviation across three random seeds.

