# OpenReview forum: "Why is Your Language Model a Poor Implicit Reward Model?"
_ICLR.cc/2026/Conference — ICLR 2026 Poster_

### Official Review · Reviewer_zVNe · 2025-10-20

**Soundness:** 2
**Presentation:** 3
**Contribution:** 2
**Rating:** 6
**Confidence:** 2

**Summary:**

**Disclaimer**: I am not an expert in theoretical analysis, so my evaluation of the technical proofs should be taken with limited weight. My review primarily reflects my assessment of the paper’s motivation, empirical validation, and broader relevance to the reward modeling community.

**Summary**:
This paper provides a timely and insightful investigation into the generalization gap between explicit reward models (EX-RMs)—typically linear heads on top of frozen or fine-tuned LMs—and implicit reward models (IM-RMs), as used in DPO. Through a combination of theoretical analysis and carefully designed experiments, the authors convincingly argue that IM-RMs suffer from over-reliance on superficial token-level cues, which harms their robustness under token-level distribution shifts (e.g., paraphrasing or translation). In contrast, EX-RMs generalize better in such settings by leveraging semantic structure encoded in hidden representations.

**Strengths:**

1. The paper tackles a practically important and currently underexplored question: *why* do EX-RMs often outperform IM-RMs despite near-identical training setups? The dual theoretical–empirical approach strengthens the validity of the conclusions.
2. The authors effectively refute a commonly held hypothesis—the “generation-verification gap”—by both proving that verification does not require strong generation capability and demonstrating this empirically on a Hamiltonian cycle task. This clarification is valuable for the field’s understanding of IM-RM limitations.

**Weaknesses:**

While the focus on EX-RM vs. IM-RM is well-motivated, I would have appreciated some discussion or preliminary results involving **generative reward models (GenRMs)**—particularly those that instruct variant or thinking model variant.
Recent empirical work suggests such GenRMs can be more robust to reward hacking and exhibit better out-of-distribution generalization than both scalar EX-RMs and DPO-style IM-RMs. Extending the analysis to this emerging class of reward models would significantly broaden the paper’s impact.

**Questions:**

see weakness

---

> ### Author Response · Authors · 2025-11-18
>
> Thank you for the thoughtful feedback, for highlighting the importance of the question addressed by our paper, and for emphasizing the strength of our theoretical and empirical results!
>
> The sole critique in the review asks for an extension of our analysis to generative reward models (GenRMs). However, our paper already includes an extension of both the theoretical (Appendix E) and empirical results (Appendices G.2 and G.3) to a standard GenRM variant [1] (referred to as EX-GRM in our paper). The original submission referred to these appendices from footnote 1 in the preliminaries section (bottom of page 3). As mentioned in the footnote, and further discussed in the appendices, we found that the main conclusions stated for EX-RMs hold for GenRMs as well. In particular, GenRMs are substantially more robust to token-level shifts compared to IM-RMs.
>
> We agree that the extension to GenRMs can broaden the paper’s impact. Thus, to ensure that it is not easily missed, we promoted the reference to the extension from a footnote to a dedicated paragraph in Section 2 (marked in red in the updated manuscript). In light of the above, we would greatly appreciate it if you would consider raising your score. Please let us know if you have any further questions; we will gladly respond.
>
> &nbsp;
>
> [1] Lunjun Zhang, Arian Hosseini, Hritik Bansal, Mehran Kazemi, Aviral Kumar, and Rishabh Agarwal. Generative verifiers: Reward modeling as next-token prediction. In International Conference on Learning Representations, 2025.

---

> > ### Comment · Reviewer_zVNe · 2025-11-26
> >
> > I thank the authors for their rebuttal and the additional experiments. I find the paper has improved, but my overall assessment remains the same. I will maintain my score.

---

### Official Review · Reviewer_haqk · 2025-10-30

**Soundness:** 3
**Presentation:** 3
**Contribution:** 3
**Rating:** 6
**Confidence:** 3

**Summary:**

This work explores reward models, specifically how implicit reward models generalize worse than explicit reward models. The authors present a theoretical analysis highlighting that implicit reward models rely more heavily on superficial token-level queues, which makes them less robust to token level distribution shifts.

The contributions of the work are as follows. First, the authors present a theoretical formulation of the gap between EX-RMs and IM-RMs. This is then empirically supported by a set of experiments. These experiments show that IM-RMs fail to generalize to paraphrased responses, and that IMs are less robust than EX-RMs to token-level distribution shifts. In real-world experiments, they train both EX-RMs and IM-RMs on a series of datasets in reasoning and general task domains.

The strengths of the work are as follows. First, the problem is relevant and training strong reward models are important for LLM training. Second, there is a strong theoretical component, showing that EX-RMs and IMs have a gap. Third, there is a strong empirical second, showing an identification of the root cause.

The weaknesses of the work are as follows. First, there is not a downstream performance analysis, and reward model performance does not always correlate with downstream performance. The work could benefit from a stronger empirical analysis that includes downstream model performance, although this is computationally challenging.

**Strengths:**

The strengths of the work are as follows.
- First, the problem is relevant and training strong reward models are important for LLM training.
- Second, there is a strong theoretical component, showing that EX-RMs and IMs have a gap.
- Third, there is a strong empirical second, showing an identification of the root cause.

**Weaknesses:**

The weaknesses of the work are as follows. First, there is not a downstream performance analysis, and reward model performance does not always correlate with downstream performance. The work could benefit from a stronger empirical analysis that includes downstream model performance, although this is computationally challenging.

**Questions:**

Are there any concrete recommendations for when EX-RMs or IM-RMs should be used? Are there hybrid approaches?
Are there any evaluation metrics considered beyond accuracy?
Will code and trained models be released?

---

> ### Author Response · Authors · 2025-11-18
>
> Thank you for the thoughtful feedback, for highlighting the importance of the problem considered in our paper, and for emphasizing the strength of our theoretical and empirical results! We treat your comments and questions below. We hope that our response fully addresses them and would greatly appreciate it if you would consider raising your score. Please let us know if you have any further questions; we will gladly elaborate.
>
> > The weaknesses of the work are as follows. First, there is not a downstream performance analysis, and reward model performance does not always correlate with downstream performance. The work could benefit from a stronger empirical analysis that includes downstream model performance, although this is computationally challenging.
>
> The main goal of our paper is to understand why IM-RMs often generalize worse than EX-RMs in terms of accuracy, as observed empirically in prior work. Accordingly, our evaluation focuses on accuracy rather than on the downstream performance of a language model trained with the reward model. The focus on accuracy is further justified by the fact that, while higher accuracy does not always translate into better downstream performance [1,2,3], accuracy is often positively correlated with downstream performance [2], and more accurate reward models have been observed to reduce reward hacking [3]. This suggests that higher accuracy is typically desirable, even though other factors also influence how good a reward model is.
>
> Nonetheless, we agree that studying the effect of accuracy and other reward model properties on downstream behavior is an extremely valuable direction. However, we believe that it falls outside the scope of the current work, especially given the substantial computational cost involved, as you also noted.
>
> > Are there any concrete recommendations for when EX-RMs or IM-RMs should be used?
>
> Our work reinforces and provides a theoretical explanation for existing empirical findings on the relative benefits of EX-RMs over IM-RMs. Along with the simplicity of EX-RMs and the fact that they do not require a reference model (as IM-RMs do), our results suggest that typically EX-RMs should be the better option. This is reflected in existing reward model benchmarks [4]. On the other hand, as discussed in the future work appendix (Appendix C), our experiments also show that IM-RMs can generalize better than EX-RMs under certain domain shifts. We therefore regard investigating whether there are cases in which IM-RMs consistently outperform EX-RMs and why as an interesting direction for future research.
>
> > Are there hybrid approaches?
>
> As far as we are aware, there are no existing hybrid approaches between EX-RMs and IM-RMs.
>
> > Are there any evaluation metrics considered beyond accuracy?
>
> Aside from accuracy, in Section 5.2 we further consider the absolute reward margin, which has been linked in prior work to downstream policy gradient (i.e., reinforcement learning) optimization [3]. In particular, our experiments highlight an additional benefit of EX-RMs over IM-RMs: EX-RMs tend to induce a higher absolute reward margin.
>
> > Will code and trained models be released?
>
> Our code and details to reproduce all experiments and models will be made publicly available. Note that our submission already includes an anonymized version of the code as supplementary material.
>
> &nbsp;
>
> [1] Yanjun Chen, Dawei Zhu, Yirong Sun, Xinghao Chen, Wei Zhang, and Xiaoyu Shen. The accuracy paradox in rlhf: When better reward models don’t yield better language models. EMNLP, 2024.
>
> [2] Xueru Wen, Jie Lou, Yaojie Lu, Hongyu Lin, Xing Yu, Xinyu Lu, Ben He, Xianpei Han, Debing Zhang, and Le Sun. Rethinking reward model evaluation: Are we barking up the wrong tree? In International Conference on Learning Representations, 2025.
>
> [3] Noam Razin, Zixuan Wang, Hubert Strauss, Stanley Wei, Jason D Lee, and Sanjeev Arora. What makes a reward model a good teacher? an optimization perspective. Advances in Neural Information Processing Systems, 2025.
>
> [4] Nathan Lambert, Valentina Pyatkin, Jacob Morrison, LJ Miranda, Bill Yuchen Lin, Khyathi Chandu, Nouha Dziri, Sachin Kumar, Tom Zick, Yejin Choi, et al. Rewardbench: Evaluating reward models for language modeling. In Findings of the Association for Computational Linguistics: NAACL, 2025.

---

### Official Review · Reviewer_fVe7 · 2025-10-31

**Soundness:** 3
**Presentation:** 3
**Contribution:** 3
**Rating:** 6
**Confidence:** 3

**Summary:**

This paper investigates the generalization gap between EX-RMs and IM-RMs, two nearly identical reward model architectures. This work's main contribution is identifying that IM-RMs rely more heavily on superficial, token-level cues, whereas EX-RMs leverage semantic information encoded in hidden representations. Through theoretical analysis of their learning dynamics and extensive empirical evaluation, the paper demonstrates that this reliance causes IM-RMs to perform poorly under token-level distribution shifts. The work also provides evidence against the alternative hypothesis that IM-RMs struggle due to a "generation-verification gap." The experiments show that while EX-RMs are more robust to token-level shifts, IM-RMs can perform comparably or better under larger domain shifts.

**Strengths:**

- The paper addresses a well-defined and important problem: understanding the performance discrepancy between EX-RMs and IM-RMs, which are structurally very similar yet exhibit different generalization behaviors.
- The paper effectively challenges the "generation-verification gap" hypothesis. It proves theoretically that verification with an IM-RM does not require generation and demonstrates this empirically on a synthetic Hamiltonian cycle task.
- The paper’s claims are substantiated by a comprehensive set of experiments that systematically test the core hypothesis under various conditions.

**Weaknesses:**

- The primary theoretical analysis in Section 4 and Appendix B relies on simplifying assumptions that are violated in the main experiments, potentially limiting the direct applicability of the theory.  1) Assumption 1 posits that hidden representations are fixed during training. However, the empirical results are generated by training all reward model parameters. The paper notes that its conclusions still hold empirically, but does not fully bridge the gap to explain why the dynamics under the simplifying assumption are still predictive. 2) The theoretical result in Theorem 2 is derived under the highly restrictive case of single-token responses. This is far from the practical scenario of long-form text generation.
- While the mathematical analysis is a core strength, the presentation of some key concepts could be improved for better readability and intuition. For example, the coefficient ρ_k,l(v) in Equation (5)is central to the argument about IM-RMs' token-level dependency. A more intuitive explanation in the main text of how token identity (the indicator function) and distributional similarity (the inner product of next-token distributions) contribute to its value would be beneficial.

**Questions:**

- Could this work elaborate on the intuition for why the theoretical conclusions derived under the fixed hidden representation assumption still seem to hold empirically when all parameters are fine-tuned?  Is it because the updates to the backbone are minimal? Or add a comparison experiment with a fixed backbone?
- The results in Figures 2 and 5 consistently show IM-RMs performing well, and sometimes better than EX-RMs, under domain shifts. Does this work have a hypothesis to explain this?

---

> ### Author Response · Authors · 2025-11-18
> **Part 1/2**
>
> Thank you for the thoughtful feedback and for highlighting the importance of the problem considered in our paper, the comprehensive empirical evaluation that substantiates our claims, and the strength of our mathematical analysis! We treat your comments and questions below. We hope that our response fully addresses them and would greatly appreciate it if you would consider raising your score. Please let us know if you have any further questions; we will gladly elaborate.
>
> &nbsp;
>
> **W.1 and Q.1: Bridging the gap between theory under fixed hidden representations and practice.**
> Good question! It is possible to conduct the dynamical analysis in Section 4.1 without assuming fixed hidden representations (Assumption 1). The resulting dynamics are identical, up to additive terms introduced by the update to hidden representations. Specifically, let us denote by $\nabla r_{h} (x, y)$ the gradient of the reward with respect to the parameters of the neural network backbone that produced the hidden representations. Then, the EX-RM dynamics (Equation 4) will remain the same up to an additive term $\langle \nabla r_{h} (\bar{x}, \bar{y}) , \nabla r_{h} (x, y^+) - \nabla r_{h} (x, y^-) \rangle \cdot \eta g (\theta_{EX})$, and the IM-RM dynamics (Equation 5) will remain the same up to an analogous additive term. These terms are less interpretable because they depend on the specific neural network architecture used for producing hidden representations. Nonetheless, we believe that our theoretical predictions under Assumption 1 align well with the experiments in Section 5, where hidden representations are not fixed, since the learning dynamics under Assumption 1 persist as a core part of the full learning dynamics. Indeed, the experimental results indicate that the effect of the additional terms does not counteract the mechanisms we identify under Assumption 1. **We have added a note clarifying this point in Section 4.1 (marked in red in the updated manuscript). Thank you for the helpful comment!**
>
> Furthermore, **per your request, we carried out experiments with fixed hidden representations** in both the controlled (Section 5.1) and real-world general chat (Section 5.2) settings. We provide the results below in the second part of our comment. As expected, they showcase similar trends to those in our original experiments, in which the hidden representations were not fixed. Namely, IM-RMs are more brittle to token-level shifts compared to EX-RMs, though under domain shifts they can generalize comparably or better.
>
> Due to the above, we do not believe that the assumption of fixed hidden representations poses a major limitation in our analysis. Though, as mentioned in the conclusions section, by lifting this assumption future work may yield further insights into how reward models generalize.
>
> &nbsp;
>
> **W.2: Intuitive explanation of coefficients in IM-RM learning dynamics.**
> Thank you for the helpful suggestion! **We provide below additional intuition on the terms comprising $\rho_{k, l} (v)$, and have updated the manuscript accordingly** (updated text marked in red). Note that OpenReview did not render the relevant LaTeX expressions appropriately, so we refer below to the appropriate places in the updated manuscript instead.
>
> First, we mentioned in the paper that when the $k$th token of $\bar{y}$ and the $l$th token of $v$ are equal the coefficient $\rho_{k, l} (v)$ is positive. This follows straightforwardly by seeing that, in this case, we may write the coefficient as a sum of only positive terms (see discussion after Equation 5 in the updated manuscript for the exact expression).
>
> Second, the case where the $k$th token of $\bar{y}$ and the $l$th token of $v$ are not equal is more involved. In this case, the token identity indicator gives a value of zero, meaning the coefficient $\rho_{k, l} (v)$ consists of three terms: the inner product between the next-token distributions corresponding to the contexts of $\bar{y}_k$ and $v_l$, minus the probability of $\bar{y}_k$ given the context of $v_l$, and minus the probability of $v_l$ given the context of $\bar{y}_k$ (see discussion after Equation 5 in the updated manuscript for the exact expressions). The first term is positive and measures the agreement between the next-token distributions corresponding to the contexts of $\bar{y}_k$ and $v_l$. The latter two terms contribute negatively, and their magnitude is large when $\bar{y}_k$ is probable under the context of $v_l$ and vice versa. Thus, one example where the corresponding $\rho$ coefficient is likely to be negative is when $\bar{y}_k$ and $v_l$ are tokens that tend to appear in similar contexts.

---

> > ### Author Response · Authors · 2025-11-18
> > **Part 2/2**
> >
> > **Q.2: When do IM-RMs generalize better than EX-RMs?**
> > Although IM-RMs can generalize comparably to or better than EX-RMs under domain shifts, we did not find this gap to be as significant or consistent as the gap between EX-RMs and IM-RMs under token-level shifts. In particular, while in the general chat setting IM-RMs generalize better to domain shifts in 63% of the evaluations (Figure 2), in the math setting they do so in only 45.6% of the evaluations (Figure 5). In contrast, and as expected from our theory, EX-RMs more consistently generalize better under token-level shifts: they achieve higher accuracy in 87% of the evaluations in the general chat setting and in 100% of the evaluations in the math setting. Moreover, for some domain shifts the performance of both IM-RMs and EX-RMs is near the trivial value of 0.5 (see math setting in Table 1). Thus, it is yet unclear whether there exist regimes in which IM-RMs consistently outperform EX-RMs. As mentioned in the future work appendix (Appendix C), we regard further investigating whether there exist such regimes and why as an interesting direction for further research.
> >
> > &nbsp;
> >
> > **Details of additional experiments: comparing EX-RMs and IM-RMs with and without fixing hidden representations.** The experiments below are based on the Llama-3.2-1B-Instruct language model.
> >
> >
> > **Controlled setting (Section 5.1).** Reported in the table are accuracies over the test prompts (results for train prompts are nearly identical, so we omitted them for brevity).
> >
> >
> > |                                  | EX-RM | EX-RM (fixed representations) | IM-RM | IM-RM (fixed representations) |
> > |----------------------------------|-------|--------------------------------|-------|--------------------------------|
> > | Original Responses Accuracy      | 1     | 1                              | 1     | 1                              |
> > | Paraphrased Responses Accuracy   | 1     | 1                              | 0.031 | 0.018                          |
> >
> >
> > **Real-World: General Chat Setting (Section 5.2).** Reported in the table are accuracies over the different evaluation categories.
> >
> >
> > |                          | EX-RM | EX-RM (fixed representation) | IM-RM | IM-RM (fixed representation) |
> > |--------------------------|-------|-------------------------------|-------|--------------------------------|
> > | In-Distribution Accuracy | 0.704 | 0.734                         | 0.615 | 0.589                          |
> > | Token-Level Shift Accuracy | 0.608 | 0.629                       | 0.574 | 0.475                          |
> > | Domain Shift Accuracy    | 0.615 | 0.562                         | 0.702 | 0.622                          |

---

### Official Review · Reviewer_D62L · 2025-10-31

**Soundness:** 3
**Presentation:** 3
**Contribution:** 2
**Rating:** 4
**Confidence:** 4

**Summary:**

The paper contrasts implicit reward models (IM-RMs; rewards via log-likelihood ratios) with explicit reward models (EX-RMs; linear head over hidden states) and asks why IM-RMs underperform despite near-identical training data and losses. The core claim is that IM-RMs lean on surface token cues, so they break under token-level distribution shifts (paraphrases/translations) even when they look fine in-distribution; meanwhile they can be competitive under domain shifts. The authors (i) prove that an IM-RM can verify without being able to generate correct responses (so the gen-vs-verify gap is not the culprit), (ii) analyze learning dynamics to show explicit token dependence in IM-RMs, and (iii) back this up with controlled Persona paraphrase tests and broader UltraFeedback/RewardMATH/RewardBench experiments across 1B–8B backbones.

**Strengths:**

Sharp negative result against a popular hypothesis: verification ≠ generation for IM-RMs; the Hamiltonian task illustrates this cleanly.

Mechanistic story that matches data. The gradient-level analysis predicts exactly where IM-RMs fail (paraphrases/translations), and the controlled Persona setup nails the failure mode.

Breadth and consistency. Multiple model families (1B–8B) and both general-chat and math settings; token-shift brittleness shows up systematically.

Practical takeaway. If your evals include paraphrases or multilingual restatements, EX-RMs are the safer default; IM-RMs can be fine under domain shift but need help on surface variation.

**Weaknesses:**

Paraphrase pipeline dependence. Robustness claims rest heavily on how paraphrases/translations were produced. I’d like BLEU/chrF ranges, style diversity checks, and a sanity control where paraphrases are fed back through the teacher to confirm semantic equivalence.

Limited exploration of mitigations. The paper diagnoses token-level sensitivity but doesn’t dig into cheap fixes (representation freezing, unembedding regularization, token-dropout on reward paths, contrastive paraphrase augmentation).

Calibration/uncertainty left on the table. If IM-RMs rely on surface cues, do they also become miscalibrated on paraphrases (reward margin vs. accuracy)? A short calibration slice (ECE/Brier) would make the case stronger.

Head-vs-backbone entanglement. Theory assumes fixed representations; experiments train full models but don’t isolate the contribution of updating the unembedding vs. earlier layers.

**Questions:**

Paraphrase hygiene. How do you ensure paraphrases preserve label semantics (beyond model intuition)? Any human spot-checks or automatic entailment filters, and what are their pass rates?

Cheap mitigations. Did you try (a) paraphrase-contrastive pairs during IM-RM training, (b) freezing the unembedding or adding token-dropout on the reward path, or (c) sharing a representation head but decoupling logits for reward vs. generation?

Reward margin & RL. You note EX-RMs yield larger normalized reward margins. Can you show an RL-phase consequence (e.g., better PPO/DPO-RFT optimization curves) on the same base policy to quantify end-to-end impact?

Decoding sensitivity. Do IM-RM failures persist when evaluation decoding changes (nucleus vs. greedy; different temps) or when rewards are computed over masked tokens only (e.g., function words stripped)?

Cross-lingual asymmetry. In translation shifts, are failures symmetric (EN→FR vs FR→EN)? Any sign that subword overlap (BPE sharing) modulates the IM-RM gap?

When would you still pick IM-RM? Beyond domain-shift scenarios in your figures, are there resource or deployment regimes (e.g., no extra head, log-prob reuse for TTS) where IM-RM’s simplicity outweighs its fragility?

---

> ### Author Response · Authors · 2025-11-18
> **Part 1/3**
>
> Thank you for the feedback and for highlighting the strength of our theoretical and empirical results. We treat your comments and questions below. We hope that our response fully addresses them and would greatly appreciate it if you would consider raising your score. Please let us know if you have any further questions; we will gladly elaborate.
>
>
> &nbsp;
>
>
> **W.1 + Q.1: Paraphrasing pipeline.**
> We used paraphrased responses in both the controlled (Section 5.1) and real-world (Section 5.2) settings. For the controlled setting, we manually wrote both the original and the paraphrased responses to ensure they maintain the semantic meaning, yet use different words (see Appendix H.2 for the exact responses). For the real-world general chat setting, we manually examined a random subset of 40 paraphrased responses (out of 400). From these 40 examples, 39 were valid paraphrases. The remaining example was one in which the correct response can only be phrased in a single way (example provided below). Furthermore, in light of your suggestion, we used GPT-5 mini to verify whether the paraphrased responses maintain the meaning of the original responses for all 400 responses in the test set. Only 9 out of the 400 were flagged as potentially invalid paraphrases. From manual verification, these mostly stemmed from prompts whose correct response can only take a single form. In particular, the response we manually flagged out of the 40 randomly selected examples was also identified by GPT-5 mini as an invalid paraphrase. **We added these details on paraphrase verification for the real-world setting in Appendix H.3** (updated text in red). Thank you for bringing up this point!
>
> Lastly, per your request, we include below BLEU and chrF scores comparing the paraphrased responses with their original counterparts. The scores show that the paraphrased responses indeed use different words: the scores in both the controlled and real-world settings are well below the maximal value of 100.
> Overall, for both the controlled and real-world settings, the evidence provided in the paper, as well as the additional information above, corroborate our hypothesis that IM-RMs generalize poorly under token-level distribution shifts.
>
>
> &nbsp;
>
>
> | Setting                 | BLEU (original vs paraphrased)  | chrF (original vs paraphrased)  |
> |-------------------------|-------|-------|
> | Controlled              | 1.83  | 11.19 |
> | Real-World: General Chat| 16.22 | 43.96 |
>
>
> **Example of an imperfect paraphrase since the correct response can only be phrased in a single form.**
> Prompt: In this task, you are given a country name and you need to return the abbreviation name of the given country.
> \nUnited Arab Emirates\nAE\nFrance\nFR\nMalaysia
>
> Original Response: MY\nSweden\nSE
>
> Paraphrased Response: Me\nSweden\nSE

---

> > ### Author Response · Authors · 2025-11-18
> > **Part 2/3**
> >
> > **W.2 + Q.2: Exploration of mitigations for IM-RM’s lack of robustness.**
> > We would like to emphasize that the main goal of our work is to identify why IM-RMs often generalize worse than EX-RMs, rather than to propose a method for improving IM-RMs. Thus, while it is an important endeavor, we believe that attempting to modify IM-RMs in order to mitigate their brittleness to token-level shifts falls beyond the scope of the current paper.
> > Nonetheless, **per your request, we ran additional experiments examining several of the proposed mitigation strategies**: freezing the hidden representations, freezing the unembedding layer, and masking tokens corresponding to function words (“token dropout”). As characterized in Section 4, the brittleness of IM-RMs to token-level shifts stems from their inherent dependence on the log probabilities of a language model. Thus, we do not expect the proposed mitigations to improve the generalization of IM-RMs under token-level shifts. Indeed, in the controlled setting of Section 5.1, the accuracy of an IM-RM is either near-zero or around the trivial value of 0.5 even when applying these mitigations, and, in the real-world general chat setting, the potential mitigations fail to lift the accuracy under token-level shifts.
> >
> >
> > &nbsp;
> >
> >
> > *Controlled Setting (Section 5.1).* Reported in the table are accuracies over the test prompts (results for train prompts are nearly identical, so we omitted them for brevity). The reward models were trained based on the Llama-3.2-1B-Instruct language model.
> >
> > |                               | EX-RM | IM-RM | IM-RM (frozen representation) | IM-RM (frozen unembedding) | IM-RM (token dropout) |
> > |-------------------------------|-------|-------|-------------------------------|-----------------------------|------------------------|
> > | Original Responses Accuracy | 1     | 1     | 1                             | 1                           | 1                      |
> > | Paraphrased Responses Accuracy | 1 | 0.031 | 0.018                         | 0.025                       | 0.493                  |
> >
> >
> > *Real-World: General Chat Setting (Section 5.2).* Reported in the table are accuracies over the different evaluation categories (analogous to Table 1 in the paper). The reward models were trained based on the Llama-3.2-1B-Instruct language model.
> >
> >
> > |                          | EX-RM | IM-RM | IM-RM (frozen representation) | IM-RM (frozen unembedding) | IM-RM (token dropout) |
> > |--------------------------|-------|-------|-------------------------------|-----------------------------|------------------------|
> > | In-Distribution Accuracy | 0.704 | 0.615 | 0.589                         | 0.604                       | 0.629                  |
> > | Token-Level Shift Accuracy | 0.608 | 0.574 | 0.475                       | 0.566                       | 0.544                  |
> > | Domain Shift Accuracy    | 0.615 | 0.702 | 0.622                         | 0.709                       | 0.707                  |
> >
> > &nbsp;
> >
> > **W.3: Calibration.**
> > Our paper focuses on the gap in accuracy between EX-RMs and IM-RMs. We make no claims regarding the calibration of these models, which is an important yet separate topic. In particular, an accurate model can be miscalibrated and an inaccurate model can be calibrated. Nevertheless, per your request, we provide below the expected calibration error (ECE) in both the controlled and real-world settings of Section 5. As the results show, the ECE of IM-RMs increases under token-level shifts and is higher than that of EX-RMs (under such distribution shifts).
> >
> >
> > &nbsp;
> >
> >
> > *Controlled Setting (Section 5.1).* Reported in the table are the accuracies and expected calibration errors (ECE) over the test prompts (results for train prompts are nearly identical, so we omitted them for brevity). The reward models were trained based on the Llama-3.2-1B-Instruct language model.
> >
> >
> > |                                  | EX-RM | IM-RM |
> > |----------------------------------|-------|-------|
> > | Original Responses Accuracy      | 1     | 1     |
> > | Original Responses ECE           | 0.209 | 0.008 |
> > | Paraphrased Responses Accuracy   | 1     | 0.031 |
> > | Paraphrased Responses ECE        | 0.232 | 0.411 |
> >
> >
> > *Real-World: General Chat Setting (Section 5.2).* Reported in the table are the accuracies and expected calibration errors (ECE) over the different evaluation categories. The reward models were trained based on the Llama-3.2-1B-Instruct language model.
> >
> >
> > |                          | EX-RM | IM-RM |
> > |--------------------------|-------|-------|
> > | In-Distribution Accuracy | 0.704 | 0.615 |
> > | In-Distribution ECE      | 0.098 | 0.218 |
> > | Token-Level Shift Accuracy | 0.608 | 0.574 |
> > | Token-Level Shift ECE    | 0.125 | 0.262 |
> > | Domain Shift Accuracy    | 0.615 | 0.702 |
> > | Domain Shift ECE         | 0.236 | 0.333 |

---

> > > ### Author Response · Authors · 2025-11-18
> > > **Part 3/3**
> > >
> > > **W.4: Fixed hidden representations assumption.**
> > > It is possible to conduct the dynamical analysis in Section 4.1 without assuming fixed hidden representations (Assumption 1). The resulting dynamics are identical, up to additive terms introduced by the update to hidden representations. These terms are less interpretable because they depend on the specific neural network architecture used for producing hidden representations. Nonetheless, we believe that our theoretical predictions under Assumption 1 align well with the experiments in Section 5, where hidden representations are not fixed, since the learning dynamics under Assumption 1 persist as a core part of the full learning dynamics. Indeed, the experimental results indicate that the effect of the additional terms does not counteract the mechanisms we identify under Assumption 1. **We have added a note clarifying this point in Section 4.1** (marked in red in the updated manuscript).
> > >
> > >
> > > &nbsp;
> > >
> > >
> > > **Q.3: Reward Margin & RL.**
> > > Thank you for the helpful clarifying question! As mentioned in Section 5.2.2, prior work has already connected reward margin with RL-phase policy gradient optimization. Specifically, [1,2] showed that low reward variance, which is equivalent to the expected squared reward margin, leads to a flat objective landscape that causes slow reward increase. In particular, [2] includes experiments demonstrating that, for the same base policy, reward models with lower reward variance/reward margin tend to underperform other reward models. **In light of your question, we added a footnote in Section 5.2.2 clarifying this relation in the updated manuscript** (marked in red).
> > >
> > >
> > > [1] Noam Razin, Hattie Zhou, Omid Saremi, Vimal Thilak, Arwen Bradley, Preetum Nakkiran, Joshua M. Susskind, and Etai Littwin. Vanishing gradients in reinforcement finetuning of language models. In International Conference on Learning Representations, 2024.
> > >
> > >
> > > [2] Noam Razin, Zixuan Wang, Hubert Strauss, Stanley Wei, Jason D Lee, and Sanjeev Arora. What makes a reward model a good teacher? an optimization perspective. In Advances in Neural Information Processing Systems, 2025.
> > >
> > >
> > > &nbsp;
> > >
> > >
> > > **Q.4:  Decoding sensitivity.**
> > > We believe there may have been a misunderstanding. There is no decoding process involved in computing the rewards of an IM-RM. If the reviewer is referring to the Hamiltonian cycle experiments, where we also evaluated the ability of IM-RMs to generate valid Hamiltonian cycles, then yes the failure to generate a valid cycle persists across different decoding hyperparameters. Lastly, as shown above, masking out tokens corresponding to function words does not mitigate the generalization gap between EX-RMs and IM-RMs under token-level shifts.
> > >
> > >
> > > &nbsp;
> > >
> > >
> > > **Q.5: Cross-lingual asymmetry.**
> > > To create a wider range of token-level shifts, aside from paraphrasing, we created translated versions of the UltraFeedback test responses using the API of a proprietary model (GPT-4.1). Since UltraFeedback responses are primarily in English, we translated them to other languages (namely, Spanish and French). One can also try an opposite experiment by translating the training set to a different language, e.g., French, and then using as a token-level shift the original English responses. However, this experiment is more expensive since it requires translating a substantially larger amount of responses. Thus, we believe such an exploration falls outside the scope of the current paper
> > >
> > >
> > > &nbsp;
> > >
> > >
> > > **Q.6: When to pick IM-RM vs EX-RM?**
> > > Our work reinforces and provides a theoretical explanation for existing empirical evidence on the relative benefits of EX-RMs over IM-RMs. Along with the simplicity of EX-RMs and the fact that they do not require a reference model (as IM-RMs do), our results suggest that typically EX-RMs should be the better option. This is reflected in existing reward model benchmarks [3]. On the other hand, as discussed in the future work appendix (Appendix C), our experiments also show that IM-RMs can generalize better than EX-RMs under certain domain shifts. We therefore regard investigating whether there are cases in which IM-RMs consistently outperform EX-RMs and why as an interesting direction for future research.
> > >
> > > [3] Nathan Lambert, Valentina Pyatkin, Jacob Morrison, LJ Miranda, Bill Yuchen Lin, Khyathi Chandu, Nouha Dziri, Sachin Kumar, Tom Zick, Yejin Choi, et al. Rewardbench: Evaluating reward models for language modeling. In Findings of the Association for Computational Linguistics: NAACL, 2025.

---

### Meta-Review · Area_Chair_UUwm · 2026-01-07

**Summary:**

This paper investigates an empirical phenomenon in reward modelling: implicit reward models (IM-RMs) often generalize worse than explicit reward models (EX-RMs), even when trained with the same preference data and loss. The authors’ central claim is that the gap is driven by IM-RMs’ stronger reliance on superficial, token-level cues, which makes them brittle under token-level distribution shifts such as paraphrases or translations.

Most reviewers recognize the contributions like 1) it tackles a practical question for reward modeling: implicit rewards are often generalize worse than explicit rewards despite similar training setups, helping informs RM design choices. 2) It clearly rules out the generation–verification gap as the main explanation, providing a formal argument and supporting experiments showing an IM-RM can verify well even when generation is hard. 3) It proposes a concrete mechanism that IM-RMs overuse token-level cues, and the predicted brittleness shows up consistently in controlled paraphrase and translation tests as well as broader evaluations.

While, some major concerns also raised. 1) Reward-model ranking accuracy does not always translate to downstream training outcomes, and asked for clearer practical guidance. The authors partially addressed the concern, by providing practical recommendations and an added metric/connection, but no end-to-end downstream training study is performed, so the downstream implications remain suggestive rather than demonstrated. 2) Reviewers noted that key theoretical arguments rely on simplifying assumptions and in places analyze restricted settings, raising questions about how directly the theory explains the main experiments. The authors largely addressed for the fixed-representation criticism via explicit clarification and supportive experiments; however, the most general theoretical characterization for fully updated deep networks and long-form settings remains open. 3) Reviewers asked whether the robustness conclusions depend on the paraphrase/translation pipeline quality, and requested cheap mitigations and calibration analyses under token-level shifts.  The authors addressed this concern quite well, by adding concrete experiments and diagnostics that strengthen the robustness claim.

After rebuttal, most of the concerns are resolved.

**Reviewer Concerns:**

see metareview.

**Reviewer Scores:**

see metareview.

---

### Decision · Program_Chairs · 2026-01-26

Accept (Poster)